# Bacterial microcompartments and energy metabolism drive gut colonization by *Bilophila wadsworthia*

Lizbeth Sayavedra [1,2] ✉, Muhammad Yasir [1,2], Andrew Goldson [1,2], Arlaine Brion [1,2], Gwenaelle Le Gall [2,3], Mar Moreno-Gonzalez [1,2], Annalisa Altera [1,2], Michael D. Paxhia [1,2], Martin Warren [1,2,3,4], George M. Savva [1,2], A. Keith Turner [1,2], Naiara Beraza [1,2] & Arjan Narbad [1,2]

High-fat diets reshape gut microbiota composition and promote the expansion of *Bilophila wadsworthia*, a sulfidogenic bacterium linked to inflammation and gut barrier dysfunction. The genetic basis for its colonisation and physiological effects remain poorly understood. Here, we show that *B. wadsworthia* colonises the gut of germ-free male mice fed a high-fat diet by relying on genes involved in microcompartment formation and anaerobic energy metabolism. Using genome-wide transposon mutagenesis, metatranscriptomics and metabolomics, we identify 34 genes essential for gut colonisation, including two clusters encoding a bacterial microcompartment (BMC), and a NADH dehydrogenase (*hdrABC-flxABCD*) complex. These systems enable *B. wadsworthia* to metabolise taurine and isethionate, producing $H_2S$, acetate, and ethanol. We further demonstrate that *B. wadsworthia* can produce and consume ethanol depending on the available electron donors. While *B. wadsworthia* reached higher abundance and $H_2S$ production in the absence of the simplified microbiota, its co-colonisation with the defined microbial consortium exacerbated host effects, including increased gut permeability, slightly elevated liver ethanol concentrations, and hepatic macrophage infiltration. Our findings reveal how microbial interactions and metabolic flexibility -including using alternative energy sources such as formate- rather than $H_2S$ alone, shape *B. wadsworthia*'s impact on host physiology, with implications for understanding diet-driven microbiome–host interactions.

Animal-derived dietary fat represents a significant source of saturated fats in traditional Western diets and has a well-established association with obesity, type II diabetes, cancer and cardiovascular disease[1]. The consumption of a high-fat (HF) diet has been linked to an increase in the abundance of *Bilophila wadsworthia*, a pathobiont that thrives in the presence of taurine-conjugated bile acids[2]. *B. wadsworthia* has emerged as a bacterium of interest as it has been implicated in gut inflammation, colorectal cancer, bile acid metabolism and liver steatosis[2–4]. Despite being part of the commensal human microbiome, the factors that trigger *B. wadsworthia* to become detrimental remain largely unknown[5]. A by-product of *B. wadsworthia* respiration is $H_2S$, which has a detrimental impact on the host that is dosage dependent.

[1]Quadram Institute Bioscience, Norwich Research Park, Norwich NR4 7UQ, UK. [2]Centre for Microbial Interactions, Norwich Research Park, Norwich NR4 7UG, UK. [3]University of East Anglia, Norwich Research Park, Norwich NR4 7TJ, UK. [4]School of Biosciences, University of Kent, Giles Ln, Canterbury CT2 7NZ, UK. ✉e-mail: lizbeth.sayavedra@quadram.ac.uk

On the beneficial side, H$_2$S is an important signalling molecule in mammals and a cellular antioxidant[6]. However, high concentrations can disrupt mucus integrity of the gastrointestinal tract, cause inflammation and contribute to cancer development[7]. It is therefore crucial to understand what initiates members of the gut microbiota to produce H$_2$S.

Sulfidogenic bacteria, such as *B. wadsworthia*, have been shown to confer protection against pathogens like *Klebsiella pneumoniae* and *Citrobacter rodentium*[8,9]. However, it remains unclear how the presence of *B. wadsworthia* affects the commensal microbiome under HF dietary conditions. Thus, we sought to determine whether the commensal microbiota was significantly altered in response to *B. wadsworthia* colonisation. We hypothesised that *B. wadsworthia* would further alter the composition and function of the gut microbiota beyond changes caused by a HF diet alone, and that different gene sets would be upregulated to enable *B. wadsworthia* to colonise the gut in the presence or absence of other representative members of the microbiome.

To address these hypotheses, we employed whole-genome-based Transposon-Directed Insertional Sequencing (TraDIS). This technique uses the transposase enzyme to randomly insert Tn5 mini-transposons into the bacterial genome, creating a library of mutants with disrupted gene function or altered genes[10]. After extracting and fragmenting the DNA, the transposon-flanking regions are selectively amplified using PCR and prepared for sequencing. The resulting sequencing data reveals the locations of transposon insertions, quantifies insertion frequencies and identifies genes essential for bacterial survival or important for specific growth conditions. Germ-free mice exposed to a HF diet were gavaged with the mutant pool of *B. wadsworthia*, a simplified humanised consortium (SIHUMI), or a combination of both. Additionally, we examined the impact of *B. wadsworthia* on host health, including weight change, inflammation markers, gut permeability and liver pathology. By elucidating the molecular mechanisms underlying *B. wadsworthia* gut colonisation, its interactions with other gut microbiota and integrating multi-omics approaches (metatranscriptomics, metabolomics), our study offers the first comprehensive understanding of the role of *B. wadsworthia* in the pathogenesis of metabolic disorders associated with HF diets. This research not only sheds light on the complex dynamics of the gut microbiome but also paves the way for targeted therapeutic strategies to mitigate the adverse effects of HF diets.

## Results

### Isolation, sequencing, and construction of a mutant library from *Bilophila wadsworthia* QI0013

We isolated *B. wadsworthia* QI0013 from the faeces of a healthy human and sequenced its genome using short and long reads. The resulting genome produced a circular chromosome of 3.61 Mbp and two circular megaplasmids of 161,291 bp and 680,317 bp. The genome has four copies of the ribosomal RNA operons and a completeness of 100%. Strain QI0013 shared >96% ANI with other *B. wadsworthia* strains, including 3.1.6 and ATCC 49260, and formed a subclade with *B. wadsworthia* 4.1.30 (>98.5% ANI) (Fig. 1a). Most *B. wadsworthia* genomes and metagenome-assembled genomes (MAGs) were recovered from human samples, as previously reported from 16S rRNA amplicon sequencing[8].

Wild-type *B. wadsworthia* QI0013 was used to construct two mutant libraries with a conjugative plasmid incorporating a gene and a mini-Tn5 transposon conferring chloramphenicol resistance. As the donor strain, we used *E. coli* MFDpir, which requires diaminopimelic acid (DAP) for growth. The resulting collection of mutants, also referred to as the mutant library, had a total of 40,613 insertions. This corresponds to an average distance of one mutation every 88 bases spread among the mutant population. Only 3722 out of 4582 identified genes had transposon insertions, allowing these genes to be assayed.

### Simplified humanised consortium of microbiota modulates *B. wadsworthia* abundance

Germ-free male mice fed a milk-derived HF diet were colonised with either *B. wadsworthia* mutant libraries, eight other representative gut bacterial species (SIHUMI), or a mixture of *B. wadsworthia* and SIHUMI (Bw+SIHUMI) after 16 days of being exposed to a HF diet (Fig. 1b). Together, the SIHUMI community represents the metabolic functions of the gut of a healthy human, including the ability to ferment amino acids and produce short-chain fatty acids (*Escherichia coli*, *Anaerostipes caccae*, *Clostridium butyricum*, *Thomasclavelia ramosa*), break down polysaccharides, dietary protein and plant material (*Bacteroides thetaiotaomicron*), and consume oligosaccharides and simple sugars (*E. coli*, *Lactiplantibacillus plantarum*, *Bifidobacterium longum* and *Blautia producta*)[11,12].

To verify colonisation of the mice intestines by *B. wadsworthia*, we quantified the abundance of *B. wadsworthia* using qPCR that targeted a single-copy marker gene, *tpa*, in *B. wadsworthia*. The abundance of *B. wadsworthia* was determined in the duodenum, jejunum, and ileum (small intestine), as well as in the colon and stool. The abundance of *B. wadsworthia* was significantly lower in mice with SIHUMI+Bw compared with *B. wadsworthia* alone in the first two segments of the small intestine (Duodenum: df = 48; t = 2.21; p = 0.0317; 95% CI = [0.39, 7.35]; Jejunum: df = 48; t = 4.53; *p* < 0.0001; 95% CI = [3.56, 9.20]). The trend remained consistent for ileum, the colon and stool, although the difference was not as pronounced (Fig. 1c).

Absolute counts of bacteria can be affected by the addition of a pathobiont. Thus, we assessed the influence of *B. wadsworthia* on the SIHUMI composition by evaluating microbial abundance and composition in stool samples collected on day 27 after the initiation of the experiment (Fig. 1b, d); we achieved this using Quantitative Microbiome Profiling (QMP)[13] by combining the metagenomic taxonomic profiling, and cell quantification using flow cytometry. This approach enabled a prediction of absolute values for cell counts. Total bacterial cell counts were significantly lower in mice gavaged only with *B. wadsworthia* than in mice gavaged with either SIHUMI or SIHUMI+Bw (Fig. 1d). In agreement with our qPCR quantification, the abundance of *B. wadsworthia* was significantly higher in the absence of the SIHUMI community than in the presence of the SIHUMI community (Fig. 1c, d). This taxonomic profiling also showed that mice gavaged only with the *B. wadsworthia* mutant library still had *E. coli* present, the bacterium used as a donor for conjugation and creation of the mutant library, despite our attempts to eliminate it by removing DAP and adding chloramphenicol to the recovery media. From the SIHUMI community, *Blautia producta* (Z = 2.63, *p* = 0.025), *Thomasclavelia ramosum* (Z = 2.93, *p* = 0.015), and *Clostridium butyricum* (Z = 2.34, *p* = 0.043) were significantly reduced in the SIHUMI+Bw group compared with the SIHUMI group, as assessed by a two-sided Wilcoxon rank-sum test with Benjamini–Hochberg correction. This suggests that *B. wadsworthia* had a negative impact on these taxa, potentially due to competition for hydrogen, formate or pyruvate.

### Genes for microcompartments and energy metabolism are essential for gut colonisation

We used a genome-wide transposon insertion mutagenesis method known as TraDIS[14] to identify the fitness determinants of *B. wadsworthia* related to gut colonisation. Using this approach, we queried the 3722 genes whose mutation did not prevent growth in vitro for roles in survival and gut colonisation. Mutant frequencies from the initial gavage were compared with those from the small intestine, colon and stool. From these, 34 genes had increased fitness in all host tissues (Fig. 2a and Supplementary Data 1). These included eight genes involved in taurine metabolism through isethionate desulfonation, including genes encoding shell proteins of BMCs[15]. Microcompartments are semi-permeable organelle-like structures composed of an outer protein shell that encase a specific metabolic pathway. These

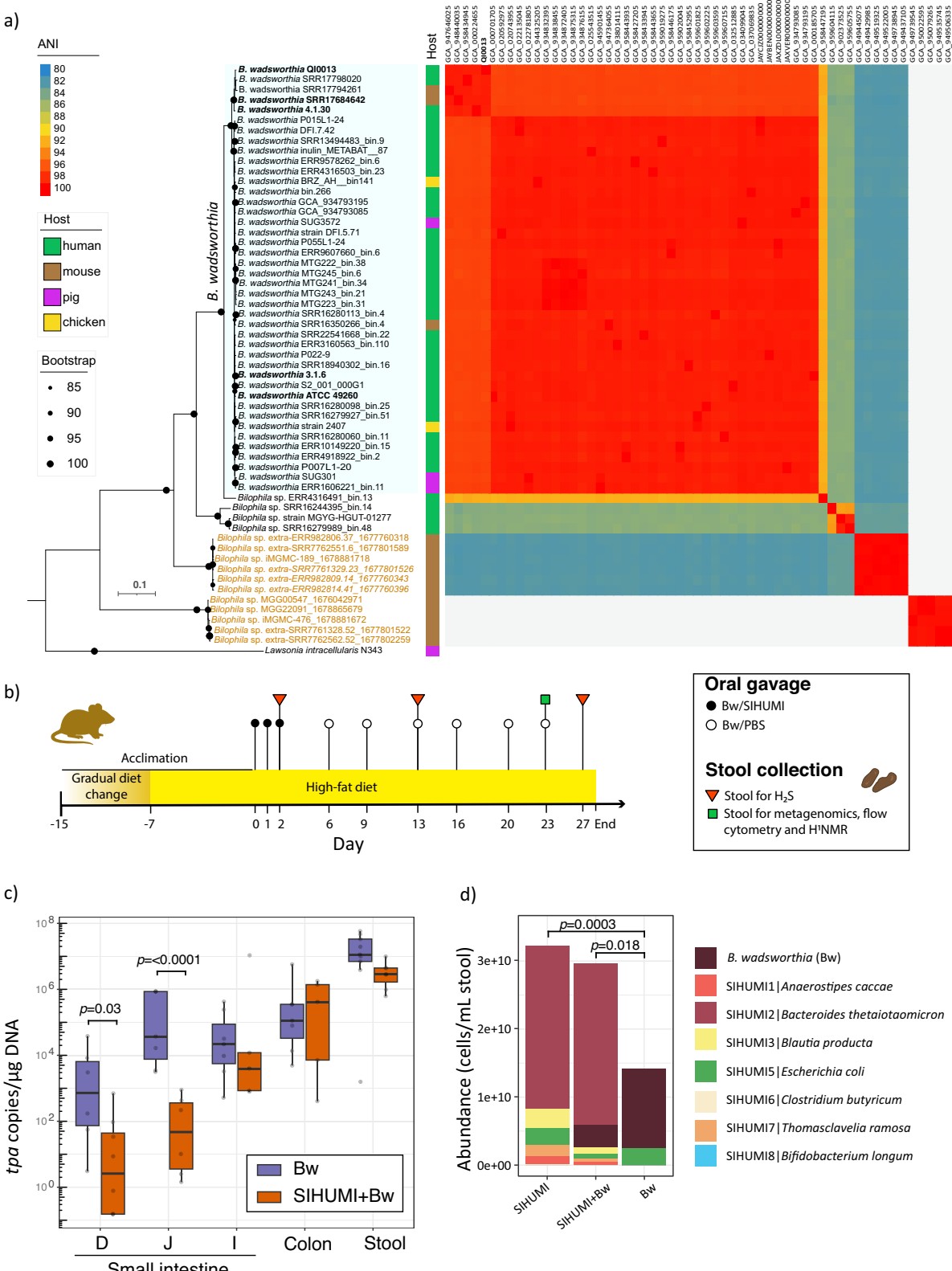

**b)**

Oral gavage
- ● Bw/SIHUMI
- ○ Bw/PBS

Stool collection
- ▼ Stool for H₂S
- ■ Stool for metagenomics, flow cytometry and H¹NMR

structures allow the bacterium to concentrate enzymes, metabolites, and cofactors for energy generation, and protect it from the accumulation of toxic intermediates, such as aldehydes[16]. The best-studied catabolic BMCs are associated separately with ethanolamine and 1,2-propanediol utilisation. Their presence give *E. coli* and *Salmonella enterica* a competitive advantage, allowing energy generation from two distinct carbon sources that are abundant in the gut[17–19]. Genes encoding shell proteins of these microcompartments were not

essential for proliferation in previous studies[20]. The genome of *B. wadsworthia* QI0013 encoded four gene clusters containing the polyhedral-body-like protein EutS, which is characteristic of BMCs. Two of these clusters encode putative ethanolamine ammonia-lyases (*eutBC*; WCP94_004157-4158 and WCP94_003786-3787), suggesting a role in ethanolamine utilisation. Indeed, ethanolamine was depleted in the mice groups that had *B. wadsworthia* (Fig. 3b). One cluster has previously been linked to isethionate utilisation[15], while the function of

**Fig. 1 | Human-derived *B. wadsworthia* successfully colonises the mouse gut. a** A maximum-likelihood phylogenomic tree of *B. wadsworthia* and its closest relatives, reconstructed using 100 single-copy genes, combined with average nucleotide identity (ANI) analysis, shows that *B. wadsworthia* QI0013 falls within the species threshold (>95% ANI). Genome accession numbers are shown above the ANI matrix. **b** Experimental design overview. Mice were acclimated to the high-fat diet for two weeks before the start of the experiment. Oral gavage treatments are shown with circles. **c** The abundance of *B. wadsworthia* was quantified along different sections of the gut (duodenum [D], jejunum [J], ileum [I], colon, and stool) using the *tpa* gene specific to *B. wadsworthia*. Stool values were measured from samples collected at week 7, one day prior to culling. Each data point represents a biological replicate, with the following sample sizes per group: duodenum (Bw: N = 6; SIHUMI +Bw: N = 6), jejunum (Bw: N = 5; SIHUMI+Bw: N = 6), ileum (Bw: N = 7; SIHUMI+Bw: N = 5), colon (Bw: N = 7; SIHUMI+Bw: N = 5), and stool (Bw: N = 9; SIHUMI+Bw: N = 7). Median values are indicated by the horizontal lines within the boxes, which span the interquartile range (IQR), while whiskers denote the 1.5× IQR. Statistical analysis was performed using a generalised linear model of the form - Group*Sample, allowing for group-specific variation, and pairwise comparisons were two-sided and adjusted using Tukey's method for multiple comparisons. **d** Average abundance of bacteria in mice stool samples, as estimated based on metagenomic taxonomic profiling and flow cytometry counts. *P*-values of two-sided Dunn-test pairwise comparisons of species abundance between groups were corrected using Benjamini-Hochberg adjustment for multiple comparisons. Source data is provided as a Source Data file.

the fourth BMC cluster remains to be discovered. Through BMC activity, potentially toxic acetaldehyde is produced as a metabolic intermediate. The isethionate cleavage microcompartment is used to metabolise the by-product of taurine respiration through isethionate, which produces sulphite and the central carbon intermediate acetate.

Genes encoding the isethionate sulphite-lyase and its activating enzyme, *islAB* (WCP94_002039-2040), which are key catabolic enzymes of the taurine and isethionate-inducible gene cluster that also includes genes for bacterial microcompartment shell proteins[15], were essential under all conditions (Fig. 2b). This aligns with previous studies on the ethanolamine and 1,2-propanediol microcompartments[19,20]. Surprisingly, two genes encoding the microcompartment shell were essential for gut colonisation, which suggests a strong selection pressure for the maintenance of this microcompartment in *B. wadsworthia*, in contrast to a previous study in *Salmonella* Typhimurium[20].

The redox balance and internal NAD$^+$ recycling from NADH within the isethionate microcompartments of *B. wadsworthia* remains elusive, despite previous suggestions implicating RnfC as an electron sink[15]. We identified a gene cluster encoding a heterodisulfide reductase linked with a flavin oxidoreductase (*hdrCBA-flxDCBA*, WCP94_001527-1532) with increased fitness in the gut (Fig. 2c). In *Nitratridesulfovibrio vulgaris* Hildenborough, this cluster facilitates NADH oxidation during growth on ethanol and sulphate, leading to acetaldehyde production; or, to a lesser extent, operates in reverse during fermentation, reducing NAD$^+$ and producing ethanol[21]. *N. vulgaris* encodes an alcohol dehydrogenase capable of reversibly oxidising ethanol, yielding NADH, which is then channelled to HdrABC for bifurcation to a ferredoxin and DsrC[21]. *B. wadsworthia* encodes an iron-containing alcohol dehydrogenase (*adh*) that also provided a fitness advantage in the gut, although it was not part of the *hdrCBA-flxDCBA* gene operon (WCP94_001710). In the absence of a NAD-dependent alcohol dehydrogenase in the operon of the isethionate microcompartment, this alcohol dehydrogenase may reduce acetaldehyde to ethanol using NADH, similar to that in *N. vulgaris*. Given the consistent essentiality and co-expression of *hdrABC-flxABCD* with the isethionate microcompartment gene cluster, we propose that this flavin-based electron bifurcation system plays a role in NADH recycling in *B. wadsworthia* (Fig. 2c).

Metatranscriptomes from mouse caecum material revealed that the isethionate microcompartment-related genes and the *hdrCBA-flxDCBA* gene cluster were amongst the most highly expressed by *B. wadsworthia*, and higher than when grown under in vitro conditions. Transmission electron microscopy images highlighted the presence of structures within the cytoplasm of the bacterial cells from caecum material from mice in the *B. wadsworthia* group, which are consistent with the size of previously reported BMCs[15]. The structures are observed to be polygonal, with straight sides and are clustered together, similar to previously reported structures for BMCs (Fig. 2d). This is in contrast to growth of *B. wadsworthia* in vitro with taurine as an electron acceptor (10 mM), where these BMC structures were not observed (Supplementary Fig. 1), indicating that the physiological induction of BMCs may require factors other than the presence of

taurine. Our findings differ from those of Burrichter et al.[15], who observed microcompartment formation without clustering in the *B. wadsworthia* strain 3.1.6 when cultured in a hydrogen-carbonate-buffered minimal media supplemented with 20 mM taurine.

Next, we compared whether the genes that had higher fitness in all tissues were shared with other *B. wadsworthia* strains. The *hdrCBA-flxDCBA* and the isethionate microcompartment gene cluster gut were conserved among representative *B. wadsworthia* strains 3.1.6, ATCC 49260, and 4.1.30 (Supplementary Data 2). Taken together, our results highlight the functional importance of bacterial microcompartments and energy conservation for gut colonisation by *B. wadsworthia* species.

## The microbiome stimulates *B. wadsworthia* transcriptome activity creating a distinct metabolic profile

More genes were up or downregulated in the SIHUMI+Bw (2617 genes) group than the *B. wadsworthia*-only group (2518 genes) when compared to the in vitro condition (Supplementary Data 3). From these, 162 were upregulated in gut conditions and provided a fitness benefit in SIHUMI+Bw group compared with 72 genes in the *B. wadsworthia* only group. Several genes that only had a role when *B. wadsworthia* was in a consortium with SIHUMI belonged to pathways involved in biosynthesis of pyrimidine and purine nucleotides, sugar nucleotides (UDP-α-galactose), the polyamine spermidine, molybdenum cofactor and amino acids including isoleucine, histidine, arginine and valine (Pathway Perturbation Score (PPS) >1, see "Methods"). This activity might reflect potential syntrophic interactions with other members of the simplified microbiome community.

The metabolite profile of faecal water from mice in the SIHUMI +Bw group showed a distinct metabolic profile with less variation between biological replicates compared to the *B. wadsworthia*-only group, as indicated by the tighter clustering in the principal component analysis (PCA) of $^1$H NMR metabolites from the same samples analysed for taxonomic profiling (Supplementary Fig. 2). As expected, tauro-conjugated bile acids were depleted in stool samples from mice in the *B. wadsworthia*-only or SIHUMI+Bw groups, likely due to the utilisation of taurine derived from the deconjugation of conjugated bile acids (Fig. 3b, Supplementary Data 4). Among the SIHUMI consortia, *B. thetaiotaomicron* has a bile salt hydrolase that can specifically deconjugate tauro-β-muricholic acid (BT_2086)[22]. By using this gene as a query, we found that *C. butyricum, L. plantarum, A. caccae, B. producta*, and *B. longum* also encoded bile salt hydrolases (BSH, EC 3.5.1.24). Based on the metatranscriptome analysis of the caecum, *B. thetaiotaomicron*, followed by *B. producta* and *C. butyricum*, exhibited the most active bile salt hydrolase expression (Supplementary Data 5). The next subsections will discuss the key results from the integrated TraDIS, transcriptomic and metabolomic analyses.

## Competition with the microbiome triggers the use of alternative electron donors in *B. wadsworthia*

*B. wadsworthia* can use a variety of electron donors that are the result of the fermentative process of bacteria that degrade complex

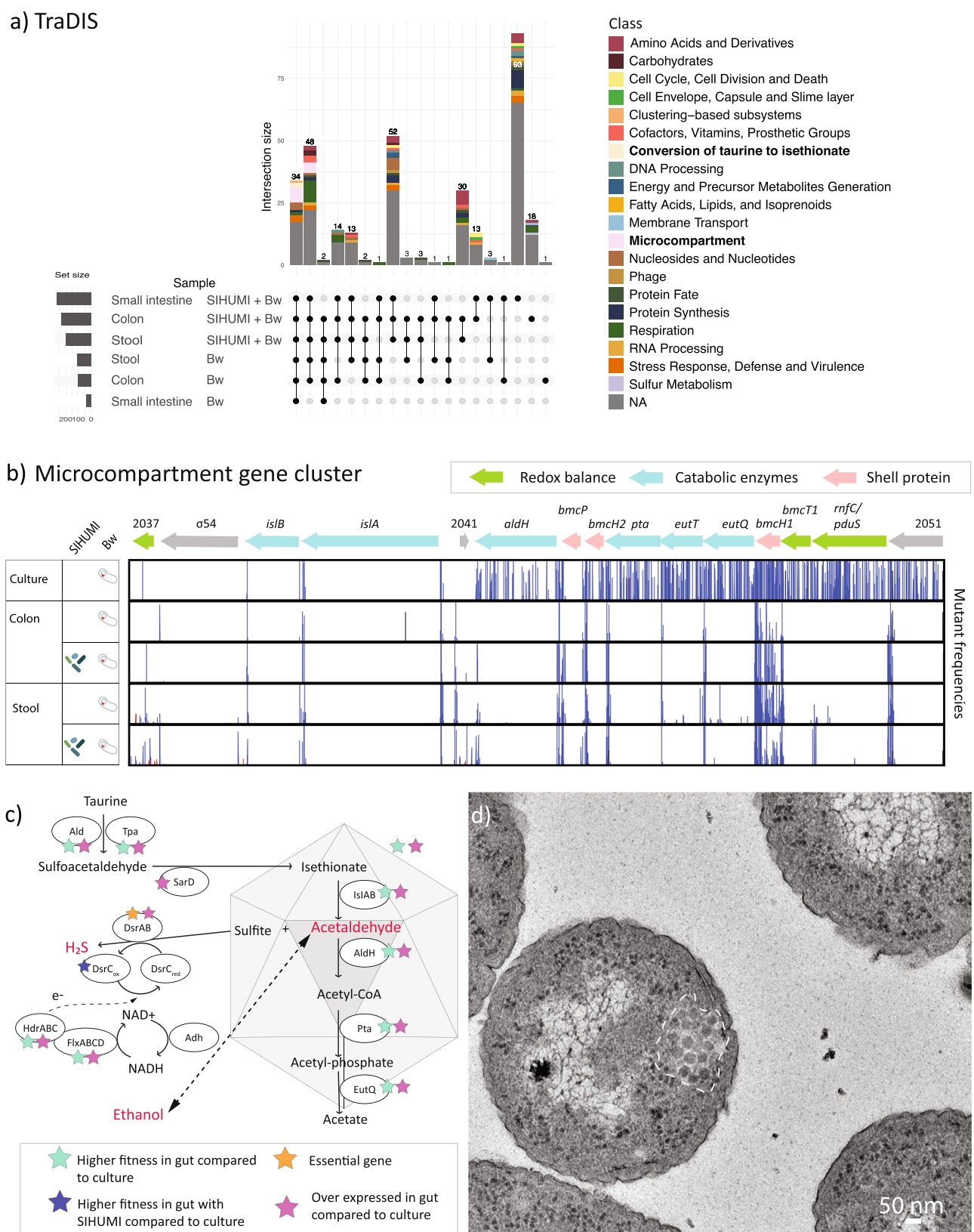

a) TraDIS

b) Microcompartment gene cluster

c)

d)

Class
- Amino Acids and Derivatives
- Carbohydrates
- Cell Cycle, Cell Division and Death
- Cell Envelope, Capsule and Slime layer
- Clustering−based subsystems
- Cofactors, Vitamins, Prosthetic Groups
- **Conversion of taurine to isethionate**
- DNA Processing
- Energy and Precursor Metabolites Generation
- Fatty Acids, Lipids, and Isoprenoids
- Membrane Transport
- **Microcompartment**
- Nucleosides and Nucleotides
- Phage
- Protein Fate
- Protein Synthesis
- Respiration
- RNA Processing
- Stress Response, Defense and Virulence
- Sulfur Metabolism
- NA

polysaccharides, including hydrogen and the short-chain fatty acids, formate, lactate, and pyruvate[23,24]. To investigate the metabolic flexibility of *B. wadsworthia*, we examined its growth in vitro to characterise the utilisation of some of these electron donors and measure the metabolite changes in the supernatant using ¹H NMR. The tested electron donors included formate, pyruvate, ethanol, and lactate,

while hydrogen was available in the headspace under the tested atmosphere for all conditions. Formate and pyruvate were rapidly consumed, with formate supporting the highest cell density of *B. wadsworthia* (Fig. 4). Over time, a significant accumulation of ethanol was observed during growth on formate and pyruvate as electron donors. When ethanol was provided as the primary energy

**Fig. 2 | Gene essentiality and expression in the gut revealed that BMCs are key for gut colonisation. a** Upset plot showing the classification of genes with increased fitness among animal tissues with the gavage of *B. wadsworthia* or SIHUMI+Bw compared to the original in vitro culture. **b** Isethionate cleavage gene cluster and frequency of mutants recovered. The height of red bars indicates mutation frequencies based on reads in the forward strand, and blue bars indicate reads on the reverse strand. Gene locus tags start with WCP94_00. **c** Microcompartments and the *hdrCBA-flxDCBA* gene operon were essential for gut colonisation and were overexpressed in the gut compared to culture conditions.

Alcohol dehydrogenases (Adh) are reversible enzymes, some of which could be active outside the microcompartment structure. The dotted arrow indicates the proposed mechanism for NAD$^+$ recycling from NADH. **d** TEM of *B. wadsworthia* inside the mice's caecum highlighting the microcompartment cluster. Transmission electron microscopy was performed on caecal content from a single mouse in the *B. wadsworthia*-only group. Two grids were imaged per sample, and ten images were captured. Structures consistent with bacterial microcompartments were observed in the majority of *B. wadsworthia* cells (20 out of 35 cells imaged).

source, acetaldehyde accumulated progressively in the supernatant, suggesting the induction of ethanol metabolism. The absence of intracellular retention suggests that acetaldehyde was not sequestered within microcompartments, and therefore was not the result of taurine desulfonation. Unexpectedly, lactate slowed the growth of *B. wadsworthia*, and during growth on pyruvate, lactate concentrations significantly increased over time. Acetate, a short-chain fatty acid (SCFA), accumulated with the growth of *B. wadsworthia*.

The analyses of mutant fitness and transcriptomic responses of *B. wadsworthia* in the gut compared to in vitro conditions revealed significant differences in genes required for the utilisation of hydrogen, formate, and lactate, which are further discussed below. Moreover, the $^1$H NMR metabolome of stool samples revealed subtle differences in the abundance of some of the electron donors of *B. wadsworthia* when together with the SIHUMI consortia. There was some evidence for formate and lactate concentrations being higher in the *B. wadsworthia*-only group compared to the other two groups (see Fig. 3a, b and Supplementary Data 4). Thus, we hypothesised that in the presence of the SIHUMI consortia, *B. wadsworthia* diversifies its use of energy sources in response to competition and the availability of other electron donors resulting from the fermentative metabolism of the members of the SIHUMI consortia[25], for example, for hydrogen.

Hydrogen is a high-energy reductant generated by colonic fermenters efficiently utilised by *B. wadsworthia*[23] and other gut bacteria such as the acetogen *Blautia producta*. *B. wadsworthia* encodes a soluble [FeFe] hydrogenase (WCP94_003875) and four [NiFe] uptake hydrogenases (WCP94_003605, WCP94_003816, WCP94_001421, WCP94_001823), highlighting hydrogen's role as an important energy source. Although hydrogen has been predicted to contribute to *B. wadsworthia* virulence, as it does for other gut pathogens[23], none of the hydrogenases had an increased fitness benefit during gut colonisation (Supplementary Data 1). However, the [NiFe] hydrogenase genes *hyaAB* (WCP94_003604-3605) were essential in the initial pool of the mutant library, which precluded their direct testing in the gut environment. Interestingly, transcriptomic data showed reduced expression of this [NiFe] hydrogenase *hyaABC* in the SIHUMI+Bw group compared with the *B. wadsworthia* alone group (Supplementary Data 3), supporting that *B. wadsworthia* might broaden its electron donor usage when co-inhabiting the gut with the SIHUMI community.

Formate and hydrogen play a crucial role in electron transfer between bacterial species, with formate intracellular cycling being used for energy conservation[26]. To use formate, bacteria rely on formate dehydrogenases, enzymes that catalyse the reversible reaction transforming formate to $CO_2$[26]. Pathway analysis of gene expression perturbations in *B. wadsworthia* suggested a significant disruption in formate respiration activity, being among the top 10 perturbed pathways based on the pathway perturbation score calculated by Pathway Tools when comparing the transcriptome of in vitro vs. in vivo conditions[27]. Three formate dehydrogenases showed significant changes in expression: *fdhAB* (WCP94_000777-778), *fdhABC₃* (WCP94_003998-4000), and *fdoI* (WCP94_000049). In the gut, *fdhAB* was under-expressed, while *fdhABC₃* and *fdoI* were overexpressed compared to in vitro conditions (see Supplementary Data 3).

The formate dehydrogenases *fdhAB* and *fdhABC₃* are the main *fdh* enzymes present when *Nitratidesulfovibrio vulgaris* is grown with

formate, lactate and hydrogen; *fdhAB* is upregulated during growth with hydrogen, while *fdhABC₃* is upregulated during growth with formate[26]. When the main electron acceptor of *N. vulgaris* is depleted (sulphate), *fdhAB* can work in reverse reducing $CO_2$ to formate, providing a proton and electron sink, leading to formate accumulation[26].

Given that the lack of microbial competition allowed *B. wadsworthia* to proliferate (Fig. 1c), it is likely that the source of its electron acceptor (e.g., taurine-conjugated bile acids) was depleted early during colonisation. Notably, *fdoI* was expressed 7.6x more in the *B. wadsworthia*-only group compared to the SIHUMI+Bw group. FdoI is used by *E. coli* for formate respiration[28]. While *fdhABC₃* and *fdoI* were essential in the input mutant pool and could not be tested for fitness differences, *fdhAB* showed a higher fitness in the *B. wadsworthia*-only group (Supplementary Data 1). Growth assays showed that formate supports *B. wadsworthia* growth at a higher rate compared to other electron donors (Fig. 4). In the *B. wadsworthia* group, formate was present and potentially available as an energy source (Fig. 3b). Taken together, these results suggest that in the absence of microbial competition, formate played an important role in supporting *B. wadsworthia* expansion in the gut.

Lactate, a metabolic by-product of *L. plantarum*, *Bacteroides* spp. and *Bifidobacterium* spp., can be used as an electron donor with lactate dehydrogenases by several members of the microbiome. Lactate is typically found in low concentrations in healthy human faecal samples (<5 mmol L$^{-1}$)[29], but high concentrations have been linked with short bowel syndrome and ulcerative colitis[30,31]. Based on the transcriptome, *B. wadsworthia*, *B. producta*, *B. thetaiotaomicron*, and *A. caccae* actively expressed L-lactate dehydrogenases (Supplementary Data 5). Lactate dehydrogenase genes (WCP94_002036-2035) had higher fitness in all the tissues of the SIHUMI+Bw group compared to the input mutant pool, as well as in the small intestine of the *B. wadsworthia*-only group. Although lactate dehydrogenase was not differentially expressed, an L-lactate permease (WCP94_002306) was more highly expressed in the gut. Lactate did not play a major role in promoting *B. wadsworthia*'s growth (Fig. 4), and, at a concentration of 53 mM, even inhibited its growth. The higher fitness of the lactate dehydrogenase in the SIHUMI+Bw groups might have helped mitigate the growth-limiting effects of lactate.

## When *B. wadsworthia* is in the presence of SIHUMI, metabolites beneficial to the host are reduced

The gut microbiota can convert lactate to the SCFAs butyrate, propionate or acetate. SCFA are produced mainly during fermentation by gut bacteria and their presence is often considered a sign of bowel health. For example, SCFA decrease in abundance in the stool of patients with ulcerative colitis or irritable bowel syndrome[32]. Stool samples from mice in the SIHUMI+Bw group had a lower abundance of acetate, propionate, and butyrate, as well as the branched-chain fatty acid isovalerate than the other two groups (Fig. 3b, Supplementary Data 4). The reduced levels of SCFA may have resulted from the competition for lactate between the SIHUMI consortia and *B. wadsworthia*, which actively expressed lactate dehydrogenases. Although *B. wadsworthia* may not have directly utilised lactate as an energy source, its lactate dehydrogenase activity could have contributed to lactate depletion. Additionally, the decreased abundance of the SCFA-producing bacteria *B. producta* and *C. butyricum* may have further influenced SCFA levels (Supplementary Data 5, Fig. 1d).

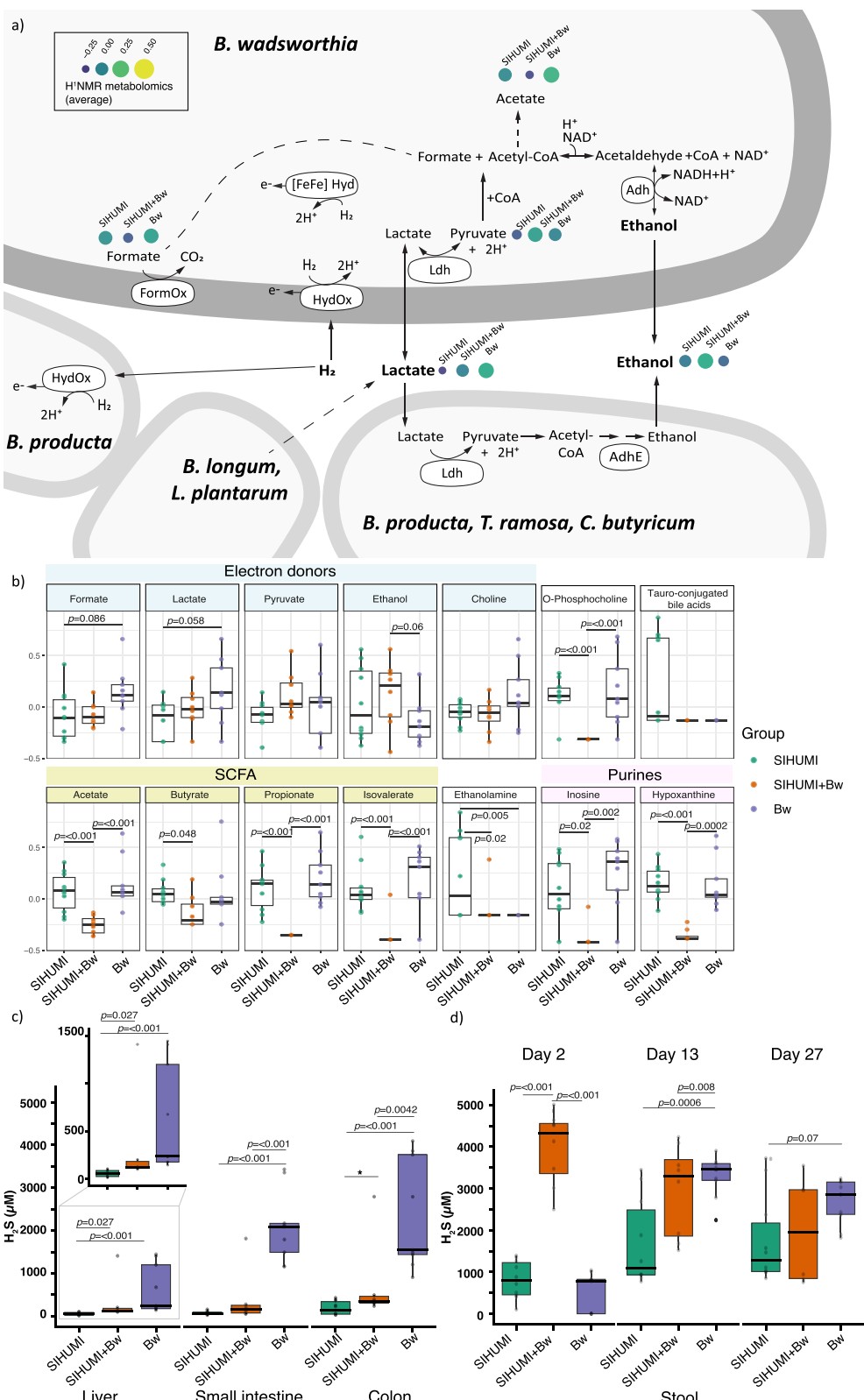

In stool, ethanol concentrations were slightly higher in the SIHUMI+Bw group (4.3 ± 3.1 mM), compared to the *B. wadsworthia*-only group (1.7 ± 1.7 mM), and the SIHUMI group (3.6 ± 3.5 mM, $p > 0.06$) (Fig. 3b). This pattern of altered ethanol levels in stool led us to investigate the ethanol concentrations in the liver. The levels were also slightly higher in the SIHUMI+Bw group (222 ± 11 μM) compared to the *B. wadsworthia*-only group (205 ± 18 μM) and the SIHUMI group (209 ± 13 μM), as revealed by [1]H NMR analysis (Fig. 5c).

These findings demonstrate that the interaction between *B. wadsworthia* and the microbiome disrupts local gut metabolite levels and leads to systemic changes in ethanol metabolism. Ethanol can be used as an electron donor by *B. wadsworthia*, which would explain the lower abundance of ethanol in the *B. wadsworthia*-only group. However, ethanol could also be produced during fermentative metabolism (Fig. 4), similar to the sulphate-reducing bacterium *N. vulgaris*[21,24]. The Flx-Hdr complex, previously shown to be involved in

**Fig. 3 | H₂S and other metabolites in mice tissue samples. a** Competition for resources might be driving *B. wadsworthia*, and other members of the SIHUMI consortia such as *B. producta, T. ramosa* and *C. butyricum* to ferment lactate to ethanol. *b. wadsworthia* can use multiple energy sources, some of which are also used by other members of the gut microbiome. The metabolite concentration detected by ¹H NMR is shown for key metabolites, transformed using square root transformation and range scaling (mean-centred and divided by the range of each variable) using Metaboanalyst. For space reasons, only the electron donors discussed in this study are shown. HydOx: hydrogen oxidation. Hyd: Hydrogenase. [FeFe] hyd: [FeFe] hydrogenase. FormOx: formate oxidation. Ldh: Lactate dehydrogenase. Adh: Alcohol dehydrogenase. AdhE *from B. producta* was on the > 3rd quartile of genes expressed in caecum. **b** Stool metabolites detected with ¹H NMR

discussed in this study, normalised using square root transformation and range scaling. **c** H₂S quantified from liver, small intestine and colon as quantified using an H₂S microsensor. **d** H₂S quantified from stool samples across time determined using the methylene blue assay. **b–d** Data represent biological replicates derived from individual mice: SIHUMI (N = 10), Bw (N = 9), and SIHUMI+Bw (N = 8 at days 2 and 13; N = 6 at day 27 due to mortality). Boxplots show the interquartile range (IQR) and the median; whiskers represent min and max values within 1.5 times the IQR. P-values were determined using two-sided generalised linear model with the square root transformed and range scaling (**b**), log10 transformed data when comparing different tissues (**c**), or the raw values (**d**) followed by pairwise comparisons with Tukey adjustment. Source data is provided as a Source Data file.

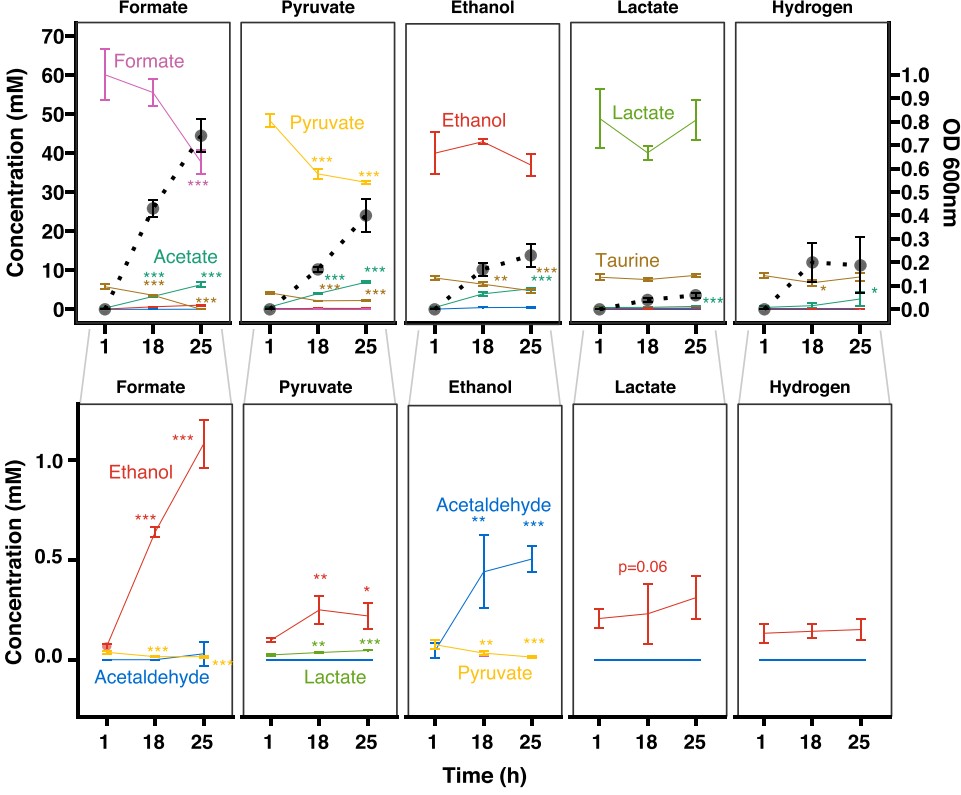

**Fig. 4 | *B. wadsworthia* can produce and consume ethanol.** Cultures were grown with taurine as an electron acceptor and various electron donors: formate (60 mM), pyruvate (60 mM), ethanol (40 mM), lactate (53 mM) or only hydrogen supplied in the gas phase (5% H₂). Acetaldehyde levels increased when *B. wadsworthia* was grown with ethanol as an electron donor. Solid lines represent the concentration of metabolites determined using ¹H NMR; dotted lines indicate OD₆₀₀. Data are

presented as mean values ± SD from N = 4 independent cultures. A one-way ANOVA was conducted in R for each incubation setup (e.g. the provided electron donor) and for each metabolite of interest with time as the predictor variable. To test for differences between each time point and the reference time (Time = 1 h), we applied Dunnett's test via the glht function. P < 0.05 (*), P < 0.01 (**), P < 0.001 (***). Source data and exact P-values are provided as a Source Data file.

ethanol production in *N. vulgaris* during pyruvate fermentation[21], may play a similar role in *B. wadsworthia*. Indeed, pyruvate was slightly increased in the SIHUMI+Bw compared to the SIHUMI group (Fig. 3b and Supplementary Data 4). Moreover, the *B. wadsworthia* alcohol dehydrogenase (*adh*) gene WCP94_001710 showed increased fitness in the small intestine of the SIHUMI+Bw group (Supplementary Data 1), suggesting that at least part of the *B. wadsworthia* population may shift towards fermentative metabolism when in competition with the SIHUMI consortia. Ethanol could then be converted to toxic acetaldehyde not only by an alcohol dehydrogenase of *B. wadsworthia*, but also by those encoded by different members of the gut microbiota[33]. Acetaldehyde is known to disrupt tight junctions, thereby compromising the intestinal cell barrier and affecting the host[33].

Transcriptomic analysis revealed significant perturbations in the ethanol metabolic pathways encoded by *B. wadsworthia*, with distinct

changes observed in the expression of alcohol dehydrogenases under gut conditions compared with in vitro conditions (in the top 5 most perturbed pathways, based on Pathways Tools). Of the nine reversible alcohol dehydrogenases encoded by *B. wadsworthia*, the aldehyde/alcohol dehydrogenase *adhE* (WCP94_001941) was 2.7x more expressed in the SIHUMI+Bw group compared to the *B. wadsworthia*-only group. AdhE is a bifunctional enzyme that converts acetyl-CoA to ethanol in two steps, playing a key role in fermentation across a range of organisms including *Klebsiella pneumoniae, E. coli, and Blautia schinkii*[34,35]. The *adh*, whose essentiality in vivo suggests a functional link to the isethionate microcompartment, was not differentially expressed in the caecum. However, its higher fitness in the small intestine of the SIHUMI+Bw group compared to other tissues suggests that lower microbiome biomass and host signalling may modulate its expression. Thus, we hypothesise that at least part of the *B. wadsworthia* population shifts to fermentation when together with the

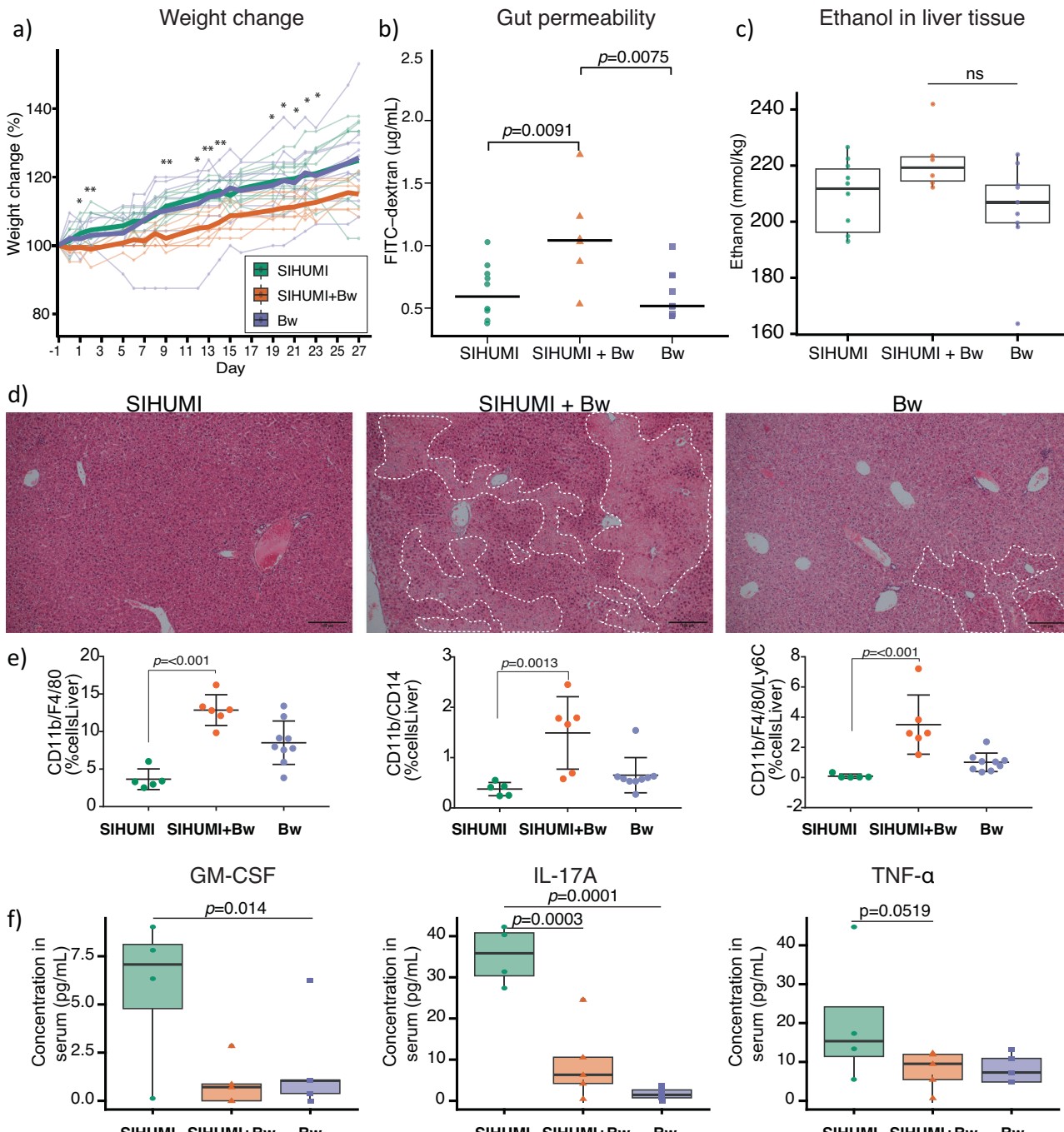

**Fig. 5 | Impact of *B. wadsworthia* on host health. a** Weight change in mice following high-fat diet acclimation. Weight change was calculated relative to body weight one day before initial gavage. The solid line represents the average weight change per treatment group. $P < 0.05$ (*), $P < 0.01$ (**), $P < 0.001$ (***) for SIHUMI vs SIHUMI+Bw. **b** Gut permeability determined using FITC-dextran was compared using a one-way ANOVA followed by two-sided pairwise comparisons of treatment groups using emmeans with Tukey's correction for multiple comparisons. Boxplots represent biological replicates from individual mice: SIHUMI (N = 10), Bw (N = 9), SIHUMI+Bw (N = 6). The solid line represents the median. **c** Ethanol quantified from liver tissue. A linear model was used to assess the difference between groups with the package lmer. SIHUMI+Bw (N = 6), SIHUMI (N = 10), Bw (N = 9). Two mice from the SIHUMI+Bw group were not included in the analysis as they died on day 40. Boxplots show the interquartile range (IQR) and the median; whiskers represent the minimum and maximum values within 1.5 times the IQR. **d** Liver histology stained with H&E of representative mice. All sections were stained on the same day. White

dashed area: areas of necrosis. Scale bar: 100 μm. **e** Macrophages in the liver were detected by flow cytometry with the gating strategy shown in Supplementary Fig. 5. Bars represent the mean of biological replicates (SIHUMI+Bw, N = 6; SIHUMI, N = 5; Bw, N = 9); error bars indicate the standard deviation (SD). Comparisons between groups were assessed using a Kruskal–Wallis test followed by two-sided pairwise comparisons with Dunn's test. *P*-values were adjusted for multiple comparisons using the Benjamini–Hochberg method. **f** Pro-inflammatory cytokines quantified from serum. Boxplots show the IQR and the median of the concentration; whiskers represent min and max values within 1.5 times the IQR. Dots show the average of two technical replicates (SIHUMI+Bw, N = 5; SIHUMI, N = 4; Bw, N = 5 biological replicates). The median is shown with a solid line. Generalised linear models were used (**a** & **f**), including baseline weight as a covariate using glmmTMB. Two-sided pairwise comparisons were calculated using emmeans with Tukey's correction for multiple comparisons. Source data and exact *p*-values for the comparisons shown in Fig. 5a are provided as a Source Data file. NS, not significant.

SIHUMI consortia, while by itself, it could be using ethanol as an additional energy source (Fig. 3a).

However, *B. wadsworthia* is likely not the only source of ethanol. Within the SIHUMI microbial consortia, *Blautia producta* expressed *adhE* at high levels in the caecum, followed by *T. ramosa*, *C. butyricum*, and *L. plantarum* (Supplementary Data 5). Additionally, *Clostridium* and *Bacteroides* species are known to produce ethanol[36]. The combined ethanol production of the members of the SIHUMI consortia, along with the fermentative activity of *B. wadsworthia* likely explains the slightly increased ethanol concentration in the livers of the SIHUMI+Bw group (see Fig. 3a for presumptive metabolite interaction).

Phosphocholine was significantly lower in the SIHUMI+Bw group compared to the other two groups (Fig. 3b). The secretion of phosphocholine in hepatocytes is primarily regulated by the bile acids cholic acid and deoxycholic acid[37]. Phosphocholine is a precursor of phosphatidylcholine, the main phospholipid found in bile. Phosphatidylcholine can be then hydrolysed by some members of the gut microbiome, including *E. coli*, to choline[38]. Choline is important in humans, contributing to cell membrane function, methyl transfer, neurotransmission and liver health, by preventing accumulation of lipids in the liver[39]. Some members of the gut microbiome can convert choline to trimethylamine (TMA), which is then converted to the disease-associated trimethylamine-N-oxide (TMAO) in the liver[38]. It has been suggested that TMAO increases the risk of fatty liver disease by decreasing the bile acid pool size[40].

*B. wadsworthia* has the genetic potential to metabolise choline to produce TMA via the action of the choline-TMA lyase *cutC* and its activating enzyme *cutD* (WCP94_000823-824)[41,42]. Additionally, *B. wadsworthia* encodes a TMA methyltransferase (WCP94_000459), which catalyses the conversion of TMA to dimethylamine (DMA), potentially reducing the available TMA for TMAO conversion in the liver[41]. Although *cutCD* did not confer a fitness advantage in the gut, both *cutC* and the TMA methyltransferase were more highly expressed in the *B. wadsworthia*-only group, which is potentially a response to the higher levels of phosphocholine observed in the colon of the mice in this group and the presence of choline (Fig. 3b).

Inosine, a purine nucleoside, and its precursor hypoxanthine were significantly decreased in the SIHUMI+Bw group compared to the other two groups (Supplementary Data 4). Inosine is known to modulate antitumor immunity and has neuroprotective and cardioprotective effects[43]. *B. wadsworthia* displayed increased fitness and higher expression of genes related to purine and pyrimidine synthesis when present with the SIHUMI consortia compared to *B. wadsworthia* itself. This suggests that *B. wadsworthia* may have contributed to the depletion of inosine and hypoxanthine in the presence of the SIHUMI consortia. However, since inosine can also be synthesised by some human microbiome members, including *B. longum*[44], the reduced abundance of inosine in the SIHUMI+Bw group might also result from shifts in the abundance and gene expression of the SIHUMI consortia due to the presence of *B. wadsworthia*.

### Presence of *B. wadsworthia* increases H$_2$S production in the gut and liver

High concentrations of H$_2$S, in the mM range, can inhibit the terminal oxidase responsible for respiratory change in the mitochondria, impairing oxygen respiration in host cells[45,46]. The gut-liver axis facilitates bidirectional metabolite exchange between the gut and liver. Thus, we sought to determine whether the presence of H$_2$S, a by-product of bacterial respiration by *B. wadsworthia*, affected various tissues including the small intestine, colon and liver. Analysis of tissues from mice harbouring only *B. wadsworthia* revealed elevated H$_2$S levels across all tissues evaluated compared to the SIHUMI+Bw and SIHUMI groups, including the liver, suggesting a direct influence of *B. wadsworthia*-derived H$_2$S on hepatic metabolism (Fig. 3c). The SIHUMI+Bw group also showed a significant increase of H$_2$S in the liver and

colon compared to the SIHUMI group. Subsequent analysis of H$_2$S content of mouse stool at days 2, 13, and 27 following exposure to a HF diet, revealed an intriguing dynamic (Fig. 3d). The SIHUMI+Bw group exhibited the highest H$_2$S levels at day 2 (Fig. 3d), peaking at $3975 \pm 883\,\mu M$, before declining over time. This suggests that one of the members of the SIHUMI consortia could have initially exacerbated H$_2$S production. Indeed, *B. thetaiotaomicron* can trigger *B. wadsworthia* to produce more H$_2$S when in co-culture[47]. Bile salt hydrolases expressed by *C. butyricum, L. plantarum, A. caccae, B. producta*, and *B. longum* could have contributed to the release of taurine from the bile acids. The availability of taurine for *B. wadsworthia* could have initially increased H$_2$S production in this group before numbers were decreased by competition with other bacteria (Figs. 1d and 3b)[48].

In contrast, the *B. wadsworthia*-only group displayed a higher trend in H$_2$S concentration compared to SIHUMI+Bw from day 13 onwards, reaching a maximum of $3281 \pm 502\,\mu M$. H$_2$S quantification in stool has the advantage of being a non-invasive method that enables tracking of the dynamics of H$_2$S in the gut, but our results suggest that in situ concentrations may vary more drastically than those quantified in stool.

### *B. wadsworthia* and SIHUMI interactions aggravate liver and systemic inflammation

We assessed the impact of the microbiome groups on mouse health, focusing on weight, gut permeability, liver macrophages and 13 serum inflammation cytokine markers. Mice gavaged with SIHUMI+Bw exhibited the least weight gain compared with those receiving SIHUMI alone (Fig. 5a, b). The SIHUMI+Bw group of mice also demonstrated the greatest gut permeability, as evidenced by elevated levels of FITC-d in the serum (Fig. 5c). The observed reduced weight gain aligns with previous findings that have associated weight loss in mice with endotoxin translocation across a leaky gut[49].

Hematoxylin and eosin staining of liver tissue from mice gavaged with either SIHUMI+Bw or *B. wadsworthia* showed increased parenchymal damage, evidenced by areas of necrosis where dark nuclei have been lost and entire cells are lost or destroyed (Fig. 5d, white dashed area), which were particularly profuse in the SIHUMI+Bw group. Hepatocellular death triggers the release of Damage-Associated Molecular Patterns (DAMPs) that activate the innate immune response, including macrophages. Macrophages are key to maintenance of hepatic homoeostasis but can also contribute to progression of liver diseases, including steatotic liver disease[50]. Flow cytometry analysis on isolated immune cells from livers confirmed the increased infiltration of macrophages. The livers of mice from the SIHUMI+Bw group had significantly increased infiltration of CD11b$^+$/F4/80$^+$, CD11b/CD14 and CD11b$^+$/F4/80$^+$/Ly6C$^+$, macrophages compared to the other two groups, which pointed to increased inflammation (Fig. 5e).

In assessing inflammation levels in mice, we conducted quantification of inflammatory cytokines from serum samples. Mice in the SIHUMI group had a greater abundance of pro-inflammatory IL-17A, GM-CSF and TNF-$\alpha$ compared to the two mice groups that had *B. wadsworthia* alone or with SIHUMI (Fig. 5f). These findings align with previous research indicating the capacity of H$_2$S to suppress the expression of some pro-inflammatory cytokines, including IL-17A, GM-CSF and TNF-$\alpha$, potentially conferring protection against acute gastrointestinal lesions[51–53]. However, in this context, H$_2$S may have an overall detrimental effect by hampering the macrophage repair response to liver parenchymal damage.

In addition to focusing on *B. wadsworthia*'s differential gene expression, we also analysed the host's transcriptome. Gene expression analysis of caecum transcriptome datasets revealed distinct host responses to colonisation by *B. wadsworthia* alone versus the SIHUMI+Bw consortium. In the SIHUMI+Bw group, mice exhibited upregulation of pathways associated with immune responses, circadian rhythm, RNA processing, plasma membrane processes, lipid binding, and

steroid and cholesterol metabolism (Supplementary Fig. 3 and Supplementary Data 6).

In contrast, mice colonised by *B. wadsworthia* alone showed upregulation of pathways related to nucleosome assembly, chromatin structural components, and protein refolding. Notably, the "alcoholism" pathway (mmu05034) was enriched in the *B. wadsworthia*-only group, which includes 24 genes linked to histone cluster 1 and the proto-oncogene AP-1 transcription factor subunit *fosB*, implicated in responses to addictive and compulsive behaviours[54] (Supplementary Data 6, based on functional enrichment using WebGestalt). The caecum mice cells also overexpressed the acetaldehyde dehydrogenases *aldh1a1*, *aldh1a7*, and *aldh18a1*, genes that encode enzymes for the detoxification of aldehyde substrates into carboxylic acids (Supplementary Data 6). Considering that ethanol concentration was higher in the SIHUMI+Bw compared to the *B. wadsworthia*-only group, and the differential host response, the mice with the *B. wadsworthia*-only group actively responded to ethanol and acetaldehyde, possibly preventing their accumulation, while the SIHUMI+Bw group had a more active immune response. The high levels of $H_2S$ in the gut and liver of these animals could have contributed to the activation of the "alcoholism" pathway, since $H_2S$ has been linked with cell apoptosis, DNA damage, histone modification and alterations in DNA methylation[55,56]. This highlights how the microbiome interactions with *B. wadsworthia* can influence the host in driving liver and systemic inflammation, suggesting a link between gut microbiota-produced ethanol and metabolic comorbidities.

## Discussion

This study determined the genetic mechanisms used by *B. wadsworthia* to colonise the gut under the influence of a milk-derived high-fat (HF) diet, either independently or in conjunction with representatives of the human microbiome. Previous research had established a correlation between HF diet consumption and increased abundance of *B. wadsworthia*[2,3], motivating our investigation into the essential genes for gut colonisation under these conditions.

Our findings revealed novel insights into the genetic determinants that facilitate adaptation in *B. wadsworthia* and its function within the gut environment. In the gut, *B. wadsworthia* showed increased fitness in genes involved in taurine and isethionate respiration, microcompartment assembly and energy conservation through the HdrABC-FlxABCD protein complex. The expression of these genes was higher in the gut compared with in vitro conditions, suggesting that *B. wadsworthia* thrives in the gut by utilising the organosulphur compounds taurine and isethionate. This metabolic adaptability resembles the strategy used by the gut pathogen *Salmonella*, which requires not only virulence factors to colonise the gut, but also genes that provide metabolic flexibility for energy conservation[20].

Taurine and isethionate are predominantly derived from the host's diet, with high concentrations in meat and seafood, though they are also found in algae and plants[57,58]. In contrast to *Taurinivorans muris*, the main taurine-utilising bacterium in the murine gut, *B. wadsworthia* can metabolise not only taurine but also isethionate and sulphite[8]. This broader metabolic range likely confers a competitive advantage to *B. wadsworthia* in the human gut, allowing it to occupy ecological niches where other bacteria may be limited by substrate availability.

In this study, ethanol metabolism emerged as an important aspect of *B. wadsworthia* physiology, particularly when interacting with the SIHUMI microbiome (Fig. 6). The potential involvement of *B. wadsworthia* in ethanol production is relevant given that ethanol can decrease bile acid secretion and synthesis, as well as modulate macrophage activation[59,60]. Moreover, recent studies have linked endogenous ethanol production by the gut microbiome to the pathogenesis of metabolic dysfunction-associated steatotic liver disease (MASLD) and its most severe form, metabolic dysfunction-

associated steatohepatitis (MASH)[61]. Under a milk-derived HF diet, caecal bile acid levels - including tauro-conjugated bile acids- were significantly increased in previous studies[2,62]. Devkota, et al.[2] showed that *B. wadsworthia*'s presence was associated with the depletion of these bile acids. Our analysis of the host transcriptome showed overexpression of genes related to alcohol metabolism in the presence of *B. wadsworthia* compared to SIHUM+Bw. In humans, alcohol metabolism is upregulated in obese individuals suffering from metabolic MASLD[61,63]. Moreover, ethanol concentration in stool samples from individuals with MASH has been found to be significantly higher than in control individuals ($2 \pm 2$ mM vs. $0.95 \pm 0.6$ mM)[64], highlighting the potential role of gut-derived ethanol in liver pathology. MASLD is linked to metabolic comorbidities, including obesity, type 2 diabetes, hyperlipidemia, hypertension, and metabolic syndrome[65].

In our study, using mice fed a high-fat diet, the slightly higher liver ethanol concentration in the SIHUMI+Bw group compared to *B. wadsworthia*, along with the depletion of SCFA compared to the other two groups, may have contributed to negative impacts on liver health, macrophage response, as well as the gut permeability. The depletion of inosine in the gut of this group could further amplify these harmful outcomes, as inosine has been shown to play a protective role in alcohol-induced liver injury in mice[66]. The combined effects of increased ethanol and $H_2S$ levels, along with reduced inosine, may have promoted hepatocyte cell death, thereby triggering macrophage infiltration and impairing their proinflammatory function under acute ethanol exposure in the livers of SIHUMI+Bw. However, since ethanol can be rapidly converted to acetaldehyde and subsequently to acetate, it remains unclear to what extent *B. wadsworthia* contributes to a significant ethanol accumulation. Future studies should examine how ethanol consumption and production fluctuate under different microbiome compositions and dietary conditions to better delineate its role.

### Pathobiont activation: what triggers *B wadsworthia* from a healthy human to become harmful?

*B. wadsworthia* has been associated with excessive accumulation of hepatic fat (steatosis), which is typical of MASLD[51,52]. However, what triggers *B. wadsworthia* to cause a detrimental effect on the host is unclear.

Our findings suggest that *B. wadsworthia* alone is not solely responsible for the detrimental effects observed in the host. Instead, its interaction with other members of the microbiota appears to amplify its pathogenicity, impacting the host's health. In the SIHUMI+Bw group, we observed increased gut permeability, the highest levels of infiltrated macrophages in the liver, and significant alterations in host gene expression related to adaptive immune response and lipid metabolism in the caecum. Previous experiments used the SIHUMI and a different *B. wadsworthia* strain (DSM 11045) in gnotobiotic Il-10$^{-/-}$ mice fed sulfoquinovose or taurocholate, and showed an initial drop in body weight (2–3%), followed by eventual recovery[67]. Burkhardt, et al.[67] also reported a slight increase in gut permeability in mice gavaged with SIHUMI+Bw and fed taurocholate compared to controls (-0.75 μg/mL FITC dextran). In our study, the combination of a high-fat diet with SIHUMI+Bw resulted in higher gut permeability ($1.2 \pm 0.3$ μg/mL FITC dextran), suggesting that the interaction between *B. wadsworthia* and the high-fat diet exacerbates the detrimental impact on the host. This highlights the significance of the diet-microbiome interaction in modulating the pathophysiological effects of *B. wadsworthia*.

The severe effects on the host from the SIHUMI+Bw contrast with the findings in the *Bw*-only group, despite this mouse group having the highest *B. wadsworthia* cell numbers and the highest $H_2S$ concentrations in the liver, gut, and stool at the last collection point before culling. The role of $H_2S$ in health has been suggested to be

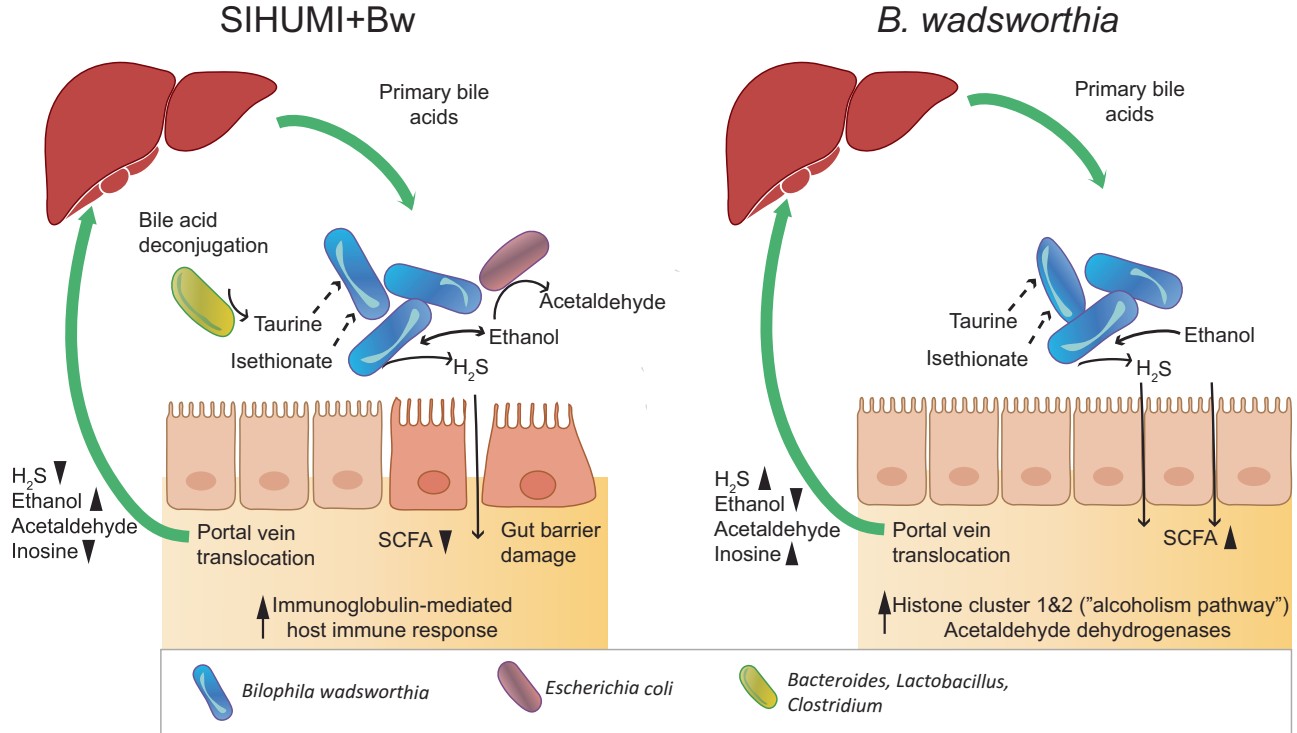

**Fig. 6 | Model of the gut-liver axis interaction of SIHUMI + Bw or *B. wadsworthia*-only.** The gut liver axis allows the transfer of some metabolites such as H$_2$S, ethanol and acetaldehyde. The liver provides primary bile acids, which can be deconjugated by bile acid hydrolases. Taurine and isethionate are respired by *Bilophila wadsworthia*, releasing H$_2$S and potentially other toxic molecules including ethanol and acetaldehyde. Ethanol can also be converted to acetaldehyde by other bacteria such as *E. coli*. Black triangles indicate whether the metabolite was in higher or lower abundance, and arrows indicate the pathways more highly expressed by the host when comparing SIHUMI+Bw versus *B. wadsworthia* alone. **SCFA**: Short-chain fatty acids.

concentration-dependent, with beneficial effects at lower doses, and detrimental at high concentrations[68]. Rey, et al.[11] showed that H$_2$S produced by *D. piger* under a high-fat diet (-0.85 mM in caecal contents) did not significantly impact gut permeability[11]. In our *B. wadsworthia*-only group, H$_2$S levels reached 2.3 ± 1.2 mM, but this still was not the primary cause of the increased permeability. However, these mice had the lowest concentration of ethanol from the groups tested, which could have prevented the exacerbated damage on the host. This difference highlights the importance of microbial interactions in exacerbating the pathogenic potential of *B. wadsworthia*.

### Impact on the microbiota

*B. wadsworthia* had a negative impact on the absolute abundance of key members of the gut microbiota. Specifically, *B. wadsworthia* led to a decrease in the abundance of the beneficial butyrate-producing bacteria *B. producta* and *C. butyricum*, which are widely recognised for their health-promoting roles[69]. *C. butyricum* has been implicated in inhibiting colorectal cancer progression and tumour growth[70]. Conversly, *B. wadsworthia* affected *T. ramosa*, a bacterium which had previously been associated with increased fat accumulation in mice under a high-fat diet[71]. A decrease in *T. ramosa* could have contributed to the observed lower body weight gain in mice receiving the SIHUMI +Bw treatment. Although *Lactiplantibacillus plantarum* was not detected in stool samples using metagenomics, transcriptomic analysis of caecal material revealed reads mapping to its reference genome (Supplementary Data 5). This suggests that this bacterium was present but below the detection limit with the metagenomic sequencing depth. High-fat diets have been linked to a decrease in the abundance of beneficial bacteria including *Lactobacillus intestinalis* and *Bifidobacterium* species, while acetate and propionate-producing bacteria such as *Clostridiales*, *Bacteroides* and *Enterobacteriales* increase in abundance[72].

These findings are in contrast to a previous study conducted in specific pathogen-free (SPF) mice, where *B. wadsworthia* was sufficient to induce systemic inflammatory responses and hepatosplenomegaly (enlarged liver) without significantly altering the overall gut microbiota composition based on 16S rRNA metataxonomics[73]. In our study, using germ-free mice, *B. wadsworthia* required the presence of additional microbial species to exert its full pathogenic effects under a high-fat diet. In contrast to Feng et al.[73], mice were gavaged with *B. wadsworthia* for three continuous days, followed by twice a week instead of a daily dose for a week. Quantitative profiling of microbiome changes was performed using a combination of flow cytometry cell counts and metagenomics, thus avoiding compositional bias from relative abundance measures[13]. Moreover, genomic differences between *B. wadsworthia* strains could also impact the outcome on the host. We used a *B. wadsworthia* strain isolated from a healthy individual, while other studies have used strains isolated from a patient with latent autoimmune diabetes[73], from a sewage plant (DSM 11045), or other patient sources, which could influence *B. wadsworthia*'s virulence.

Overall, our results suggest that i) under a HF diet that resembles a Western diet, the detrimental effects of *B. wadsworthia* on gut and liver health are exacerbated by interactions with the rest of the microbiome rather than the carriage of high levels of *B. wadsworthia* alone, and ii) *B. wadsworthia* can alter the gut microbial composition, and thus, the metabolic potential of the microbiome, which has implications for the host health. Importantly, we demonstrated that H$_2$S concentration was not the main factor driving increased gut permeability. Given that *B. wadsworthia*'s pathogenicity is amplified through the interaction with the microbiome, targeted interventions that modify the gut microbial composition, such as prebiotics, probiotics, and diet modifications, could potentially counteract the impact of *B. wadsworthia* when people are exposed to a high-fat

Western diet. For example, a combination of inosine and probiotics including *Lacticaseibacillus rhamnosus*[3], *B. producta*, *C. butyricum* and *T. ramosa* could potentially ameliorate the impact of *B. wadsworthia* on the host health. However, future studies are required to evaluate the impact of *B. wadsworthia* under different defined microbial consortia, representative of the human microbiome from different geographic locations and under different diets.

## Methods
### Ethics statement
All procedures complied with relevant ethical regulations for the use of human participants and animals in research. Human faecal sample collection was approved by the Quadram Institute Bioscience Human Research Governance Committee (IFR01/2015) and the London-Westminster Research Ethics Committee (15/LO/2169). Stool samples were donated from the QIB Colon Model, registered at ClinicalTrials.gov (NCT02653001) and conducted in accordance with the Declaration of Helsinki. Written informed consent was obtained from all participants prior to sample donation. Germ-free animal experiments were conducted under UK Home Office project licence NB70/8929 and approved by the Animal Welfare and Ethical Review Body at Quadram Institute Bioscience. All procedures followed the guidelines of the National Academy of Sciences (National Institutes of Health publication 86-23, revised 1985) and were performed within the provisions of the Animals (Scientific Procedures) Act 1986.

### Isolation, whole genome sequencing and phylogenomics of *Bilophila wadsworthia* QI0013
*Bilophila wadsworthia* QI0013 (hereafter *B. wadsworthia*) was isolated from the stool of a healthy human donor aged between 25 and 54 years. Within 3 h of collection, a 10 g sample of stool was mixed with 100 mL anaerobic PBS in a Stomacher 400 (Seward) for 30 sec and the slurry was placed in an anaerobic workstation (Don Whitley, UK). The slurry was diluted-to-extinction in modified Postgate C medium that contained, per L of distilled water: 6 g sodium lactate; 4.5 g $Na_2SO_4$, 1 g $NH_4Cl$, 1 g yeast extract, 0.5 g $KH_2PO_4$, 0.3 g sodium citrate, 0.06 g $MgSO_4.7H2O$, 0.004 g $FeSO_4.7H2O$, 4 ml resazurin (0.02% W/V), 0.04 g $CaCl_2.2H2O$, 0.5 g L-Cysteine hydrochloride and 10 mM taurine. The highest dilution that still showed a black precipitate was used to inoculate anaerobe basal broth (ABB; ThermoFisher Scientific, UK) supplemented with 10 mM taurine and sub-cultured until a pure culture was obtained.

DNA was extracted from an overnight culture grown in ABB supplemented with 10 mM taurine using the GenElute Bacterial Genomic DNA kit (NA2100, Sigma-Aldrich, United Kingdom) according to the manufacturer's instructions. DNA was sequenced at the Earlham Institute, Norwich, UK. A DNA library was constructed using a modified version of Illumina's Nextera protocol known as Low Input Transposase Enabled libraries (LITE) (Beier and Gross, 2006). Paired-end 150 bp reads were generated using the Illumina HiSeq 4000 platform. Reads were quality trimmed with bbduk (v.38.06) (sourceforge.net/projects/bbmap) using a minimum quality of three. For long-read sequencing, DNA was sent to Novogene (China) and sequenced using PacBio. A hybrid assembly was reconstructed using Unicycler (v.0.4.8)[74], scaffolded using SLR[75], and polished with Pilon (v.1.23)[76]. Genome completeness was estimated with CheckM (v.1)[77] and the genome was annotated with BV-BRC[78]. Annotation was complemented using METABOLIC (v.4)[79]. A maximum-likelihood phylogenomic tree was constructed using BV-BRC with RaxML (v.8.2.12), based on the alignment of 100 single-copy genes using MAFFT[78,80,81]. The tree included *Bilophila* spp. genomes with > 90% completeness. ANI values were calculated using fastANI (v.1.3.4)[82]. Protein sequences of QI0013 were compared against the reference strains 4.1.30, ATCC 49260, and 3.1.6 BV-BRC using BV-BRC.

### Generation of *B. wadsworthia* mutant libraries
Two mutant libraries were created using conjugation as follows: First, we evaluated *B. wadsworthia* for antibiotic resistance to chloramphenicol, kanamycin and G418; it was sensitive only to chloramphenicol, with a minimum inhibitory concentration of 12.5 µg/mL chloramphenicol in ABB media supplemented with 10 mM taurine.

As a donor strain, we used *E. coli* MFD*pir*, which requires the growth medium to be supplemented with diaminopimelic acid (DAP)[83]. This strain was used as a host for plasmid pQ*tnpA*Tn5CmPv2.1; this plasmid is a suicide vector that i) incorporates the *oriR6K* replication origin which requires the *pir* gene product for replication, ii) the mobilisation origin from plasmid RP4, and iii) protein functions encoded by the MFD*pir* host chromosome. The plasmid vector confers resistance to ampicillin and includes a mini Tn5-derived transposon coding for chloramphenicol resistance and a hyperactive allele of the Tn5 transposase for transposition of the mini-Tn5 transposon[84]. For conjugation, the donor strain was grown on Luria-Bertani (LB) medium supplemented with 0.3 mM DAP, 12.5 µg/mL chloramphenicol and 200 µg/mL ampicillin, while *B. wadsworthia* was grown on anaerobe basal broth (Oxoid, UK) supplemented with 10 mM taurine overnight. All cultures were incubated anaerobically at 37 °C. An active culture of the donor strain (2 mL) was transferred to fresh pre-warmed 10 mL media supplemented with DAP (no antibiotic), while 1 mL *B. wadsworthia* culture was transferred to pre-warmed anaerobic 10 mL ABB media supplemented with taurine and 700 µM EDTA. Cultures were grown for 4.5 hrs until an $OD_{600}$ between 0.27–0.5 was reached. Donor cells were spun down at 1500 g for 5 min and *B. wadsworthia* cells at 6000 g for 3 min. Supernatants were removed for both cultures, the pellets washed and resuspended in fresh ABB media, and re-centrifuged at the same speed and the supernatant was removed. Finally, resulting cells of each culture were resuspended in 4 mL ABB supplemented with DAP and taurine. Donor cells and *B. wadsworthia* cells were mixed in a 1:1 v/v ratio under anaerobic conditions and incubated for 1 hr. Suspensions of 50 µL were spotted onto LB agar plates supplemented with 10 mM taurine and 30 mM DAP (without antibiotic) and incubated overnight. Transconjugants were then recovered by scraping most of the bacterial cells from the plate after addition of pre-warmed anaerobic PBS. The conjugant mix recovered was centrifuged at 4000 g for 5 min at 4 °C and the supernatant removed. Cells were diluted 1:10 with ABB broth and plated onto ABB plates supplemented with 10 mM taurine and chloramphenicol, without DAP to select against the *E. coli* donor cells. Colonies were recovered after 3 days by scraping the cells from the plates. This mutant library was stored in bead cryosystem protect tubes for anaerobes (Kisker, Germany). The process was repeated to increase the number of mutants recovered.

### Evaluation of the impact of *B. wadsworthia* on mice gut and liver health
This animal experiment was conducted in accordance with the Home Office Animals (Scientific Procedures) Act 1986. C57BL/6 mice, aged 5-6 weeks old, were obtained from an in-house colony at the University of East Anglia. Mice were housed in sterile isolators in the University of East Anglia animal facility and fed on standard chow (rat and mouse No. 3 breeding, Special Diets Services, UK) before experimentation. Mice were housed in an experimental isolator, with a 12-h light/dark cycle, ambient temperature maintained at 21–23 °C, and relative humidity of 45–65%. Mice were randomly assigned to cages to minimise potential confounders related to cage location, and assays were conducted in a random order. Due to space limitations in the germ-free mice facility, the animal groups and the repeat of the experiment were staggered, as it was not possible to run them simultaneously. The assessor remained blinded to the treatment status of each mouse throughout the experiment. No statistical method was used to pre-determine sample size. In week 1 of the experiment, mice were fed

increasing proportions of a high-fat diet (HF; N = 8–10 mice per group, staggered in two sets of experiments per group, two mice of SIHUMI +Bw group died on day 40). The HF diet comprised 19.5% milk fat (45.2% kcal fat, 41% kcal carbohydrate, 3% SBO, TK VM, blue, TD.200269, Envigo, US) and the increasing daily proportions were 0%, 20%, 40%, 50%, 80%, 100% w/w HF diet to standard chow. The diet was fed *ad libitum* throughout the experiment. After 16 days, mice were gavaged on three consecutive days with either a) ~$10^8$ cells *B. wadsworthia* mutant library grown overnight b) SIHUMI consortia, with each member at ~$10^8$ cells or c) SIHUMI consortia and *B. wadsworthia*. The SIHUMI consortia was comprised of *Anaerostipes caccae* DSM 14662, *Bacteroides thetaiotaomicron* DSM 2079, *Blautia producta* DSM 2950, *Lactiplantibacillus plantarum* DSM 20174, *Escherichia coli* K-12 MG1655, *Clostridium butyricum* DSM 10702, *Thomasclavelia ramosa* DSM 1402 (previously known as *Erysipelatoclostridium ramosum)* and *Bifidobacterium longum* NCC 2705[12]. The germ-free status of the mice was confirmed before the first bacterial gavage using anaerobic culturing techniques with Brain Heart Infusion (BHI) supplemented with 0.5% v/v vitamin K, 0.0005% hemin and 0.5 g/L L-cysteine hydrochloride. The following three weeks, mice were gavaged twice per week with either *B. wadsworthia* or PBS, and stool was collected for $H_2S$ quantification on days 18, 29 and 43, as well as for taxonomic profiling and metabolomics on day 39 (see Fig. 1a for timeline). Animals were culled using cardiac puncture on day 43. Serum was collected for determination of gut permeability and cytokine expression; the liver was harvested for macrophage quantification and histology; the small intestine was divided into three fragments of similar length (duodenum, jejunum, and ileum), and, along with the colon, sub-fragments were dissected and preserved for $H_2S$ quantification and TraDIS sequencing as detailed below. Mice were dissected inside an anaerobic workstation that had an atmosphere of 90% $N_2$, 5% $CO_2$ and 5% $H_2$.

## Determination of gut permeability and cytokine expression in serum

Gut permeability in the mice was evaluated using FITC-labelled dextran (4 kDa; Sigma-Aldrich) as described by Thevaranjan, et al.[85]. Briefly, mice were gavaged 150 μL of 80 mg/mL FITC-labelled dextran in PBS 4 h before they were culled and the blood harvested. Sampled blood was kept in darkness for at least 1 h and centrifuged at 1500xg for 10 min. Fluorescence intensity of an aliquot of the clear supernatant was measured at an excitation wavelength of 485 nm and an emission wavelength of 520 nm. The rest of the supernatant was stored at −20 °C for cytokine measurements.

Thirteen cytokines were quantified simultaneously from serum using the Mouse Inflammation Panel 13-plex (740446, BioLegend, UK) according to the manufacturer's instructions. The cytokines measured included IL-23, IL-1α, IFN-γ, TNF-α, CCL2 (MCP-1), IL-12p70, IL-1β, IL-10, IL-6, IL-27, IL-17A, IFN-β, and GM-CSF. Samples were processed using a BD Fortessa LSR flow cytometer.

## Quantification of $H_2S$ in mice tissues

Hydrogen sulphide was quantified in mice tissues using an $H_2S$ microsensor (SULF-50, Unisense, Denmark) according to the manufacturer's instructions with the following modifications: Fragments of small intestine, colon, or liver were placed in pre-reduced tubes with plastic beads and 400 μL PBS under anaerobic conditions. For the small intestine, three sub-fragments from the duodenum, jejunum and ileum, were taken for quantification. Tubes were weighed before and after adding the tissue for weight normalisation. The tissue was ground for 20 s at 4 m/s using a FastPrep (MP Bio) instrument. Colon tissue underwent a second grinding cycle under the same conditions.

$H_2S$ from stool samples was quantified using the methylene blue assay[86]. Stool samples were fixed with 400 μL of 5% (w/v) zinc acetate in pre-weighed tubes containing plastic beads. Samples were homogenised in two cycles at 6 m/s for 30 seconds using a FastPrep instrument. To avoid interference with the colourimetric reaction, a 1:200 dilution of each stool sample was analysed at 670 nm and compared with a standard curve of 0-40 μM $H_2S$ that had as background a 1:200 dilution of acidified mouse stool (5.825 M HCl), left overnight open to get rid of the $H_2S$[87]. $H_2S$ levels were expressed in μM/ mg wet weight.

## $B.$ wadsworthia quantification in the mice gut

To quantify *B. wadsworthia* from the gut, the small intestine (duodenum, jejunum and ileum), colon, and stool from the animals gavaged with the Bw or SIHUMI+Bw groups were extracted using the Quick-DNA™ Faecal/Soil Microbe 96 Magbead Kit (D6010, Zymo Research, UK) as per the manufacturer's instructions. DNA was quantified using the Qubit DNA HS assay kit (Thermo Fisher Scientific, UK). *B. wadsworthia* was quantified from the extracted DNA using qPCR targeting the single-copy marker gene *tpa* with primers TPA_F (5'-CGCCGGTATC-GAAATCGTGA) and TPA_R (5'-ATTCGCGGAAGGAGCGAGAG)[3]. Thermal cycling was done on a StepOnePlus Real-Time PCR machine using KiCqStart SYBR Green qPCR ReadyMix, Low ROX (Merck, UK) according to the manufacturer's instructions. Primers were used at a concentration of 100 nM and the following conditions: 95 °C for 2 min, and 40 cycles of 95 °C for 15 s and 62 °C for 30 s. The number of copies was normalised to the DNA concentration. For statistical comparisons, a generalised linear model was run with the log of the number of copies/μL DNA using the glmmTB package with ~ *Group\*Tissue, dispformula =~ Tissue*, which allowed the variance to change with tissue type[88]. Pairwise comparisons were run with the emmeans package, and *p*-values were adjusted using the Tukey method.

## Determination of genes essential to colonise the mice gut compared to culture conditions

The genomic DNA of the mice gut tissues, previously quantified for *B. wadsworthia* abundance, was used for sequencing using biotinylated primers specific to the transposon of the mutants. For comparison of the mutant frequency with the culture conditions, an overnight culture of the pooled *B. wadsworthia* mutant libraries grown on ABB supplemented with 10 mM taurine was used as reference (same conditions used for creating the bacterial gavage used for the mice). The bacterial cell pellet was harvested and extracted using the GenElute Bacterial Genomic DNA kit (NA2100, Sigma-Aldrich, UK) following the manufacturer's instructions.

For sequencing, the DNA was diluted to 11 ng/μL wherever possible and tagmented using a MuSeek DNA fragment library preparation kit (ThermoFisher, USA). Fragmented DNA was purified using AMPure XP (Beckman Coulter, USA). DNA was amplified by PCR using biotinylated primers specific to the transposon and primers for the tagmented ends of DNA (Biotin-upTn5_2; or Biotin-upTn5_3, Supplementary Data 7), in combination with the index primers N701−N706 (Supplementary Data 7). PCR products were purified using AMPure XP beads and incubated for 4 h with streptavidin beads (Dynabeads) to enable capture of DNA fragments by the transposon. A subsequent PCR step was performed using the same barcoded sequencing primers (N701−N706, Supplementary Data 7) assigned per sample, in combination with one of the indexing primers (i5S5-, Supplementary Data 7). This dual-indexing strategy enabled multiplexing and pooling of samples for sequencing. Streptavidin beads were magnetically removed from the PCR products, which were further purified and size-selected using AMPure XP beads. PCR products were quantified using Qubit 3.0 (Invitrogen, USA) and Tapestation (Agilent Technologies, USA). The PCR products were sequenced using a Next-Seq 500/550 High Output Kit v2.5 (75 cycles) on a NextSeq 500 sequencing machine (Illumina). Nucleotide sequence reads obtained were analysed using the BioTraDIS[89] software suite, which aligns sequence reads to the reference genome of *B. wadsworthia*

thereby identifying the location of transposon insertions and the number of reads that matched at each site.

Mutant frequency was compared between the culture conditions against the small intestine, colon and stool using AlbaTraDIS[90]. To be able to increase the sequencing depth of the mutants detected, the sequencing reads of the three regions of the small intestine were combined. Additionally, the sequencing reads from replicate animals were merged per gavage group (N = 6 for SIHUMI+Bw; N = 9 for Bw), and per tissue group (small intestine, colon or stool), to have two data sets per gavage and tissue groups. Mutant frequency differences between the *B. wadsworthia* in vitro culture compared to the gavage and tissue groups were expressed as log2 fold differences with an associated q-value corrected for multiple pairwise comparisons. The minimum logFC was 1, equivalent to 2X change in mutant frequency. Pathway enrichment was determined using Pathway Tools (v.26).

**$^1$H NMR metabolomics in stool, liver and in vitro *B. wadsworthia's* growth experiments.** For $^1$H NMR analysis, pre-weighed mouse stool samples at day 39 were processed as described by Le Gall, et al.[91] with the following modifications: stools were mixed with 650 μL $^1$H NMR buffer (0.26 g $NaH_2PO_4$, 1.41 g $K_2HPO_4$, 50 mg $NaN_3$ and 1 mM sodium 3-(Trimethylsilyl)-propionate-*d*4, (TSP) made up in 100 mL 100% $D_2O$) using pellet pestles. Samples were centrifuged at 17,000 g for 5 min at 4 °C and 550 μL of supernatant was used for NMR spectral acquisition. 500 mL was transferred into a 5-mm NMR tube for spectral acquisition. High-resolution $^1$H NMR spectra were recorded on a 600-MHz Bruker Avance spectrometer fitted with a 5-mm TCI proton-optimised triple resonance NMR inverse cryoprobe and a 24-slot autosampler (Bruker, Billerica, MA, USA). Sample temperature was controlled at 300 K. Each spectrum consisted of 128 scans of 32,768 complex data points with a spectral width of 20.8 ppm (acquisition time 2.62 s). The noesypr1d pre-saturation sequence was used to suppress any residual water signal with low-power selective irradiation at the water frequency during the recycle delay (D1 = 4 s) and mixing time (D8 = 0.01 s). A 90° pulse length of 11.1 ms was set for all samples. Spectra were transformed with a 0.1 Hz line broadening and zero filling, manually phased, baseline corrected, and referenced by setting the trimethylsilyl propanoic acid methyl signal to 0 ppm. Spectra were manually phased, and baseline corrected using the TOPSPIN 2.0 software. No pooled quality control (QC) samples, blanks, or system suitability samples were included. All samples were prepared and analysed in a single batch, and the sample run order was not randomised. Metabolites were identified and quantified using Chenomx software NMR suite 7.0™, based on chemical shift, multiplicity, and spectral fitting. TSP was used for chemical shift referencing but no internal standards were used for quantification. The abundance of the metabolites was normalised to the weight of the input material. Differential abundance of metabolites was identified with MetaboAnalyst v.6.0[92].

For liver metabolomics, ~50 mg of liver tissue was homogenised using a FastPrep (MP Biomedicals, Germany) instrument for 40 s at 6 m/s, repeated twice. The homogenisation was done in lysing matrix D tubes (MP Biomedicals), 150 μL cold water, 200 μL ice-cold methanol, and 200 μL ice-cold chloroform. The slurry was incubated at −20 °C for 15 min and centrifuged at 15,000 g for 5 min. The aqueous phase (350 μL) was dried in a centrifugal evaporator and reconstituted in 600 μL of the same NMR buffer used for stool samples. $^1$H NMR spectra acquisition and processing followed the protocol for stool samples, with minor adjustments: each spectrum consisted of 48 scans of 65,536 complex data points. Processing and metabolite quantification were performed as described for stool samples.

For in vitro growth experiments, a modified Postgate C medium was used (basal medium), containing per litre of distilled water: 1 g $NH_4Cl$, 1 g yeast extract, 0.5 g $KH_2PO_4$, 0.3 g sodium citrate, 0.06 g $MgSO_4 \cdot 7H_2O$, 0.004 g $FeSO_4 \cdot 7H_2O$, 4 mL resazurin (0.02% wt./vol.), 0.04 g $CaCl_2 \cdot 2H_2O$, and 0.5 g L-cysteine hydrochloride. The pH was

adjusted to 7.5 ± 0.1 and the media was sterilised by autoclaving. The basal media was supplemented with 10 mL Thauer's vitamin solution[93], 10 mL trace element solution[94], taurine (10 mM) as electron acceptor, and one of the following electron donors: ethanol (40 mM), sodium pyruvate (60 mM), formate (80 mM), sodium lactate (53 mM), or no added electron donor ($H_2$ was present in the gas phase at 5% in all conditions). Cultures were grown under an atmosphere of 90% $N_2$, 5% $CO_2$ and 5% $H_2$.

An overnight-grown *B. wadsworthia* culture in ABB with 10 mM taurine was used to inoculate the medium (1:50 v/v). The optical density at 600 nm ($OD_{600}$) was measured at 1, 18 and 25 hrs for each sample to monitor bacterial growth. Aliquots of 500 μL were taken at these time points for metabolomic analyses. The aliquots were centrifuged at 17,000 g for 5 min, and 400 μl of the spent medium was mixed with 200 μl NMR buffer and transferred to a 5-mm NMR tube. $^1$H-NMR spectra acquisition and processing were conducted as described for stool and liver samples, with 16 spectral scans of 65,536 complex data points.

**Bacterial diversity and abundance in mice stool.** Stool pellets used for $^1$H NMR analysis were also used for cell quantification and DNA extraction. Samples were resuspended in 550 μL 1 X PBS, vortexed, diluted further 1:10 with 1X PBS, and filtered through a nylon cell strainer with a 70 μm pore size (Corning 431751, Fisher Scientific, UK). An aliquot of 1 μL was used for dilutions of 1:1600, 1:3200, and 1:6400. To stain the bacterial cells, 200 μL of dilutions were mixed with 10 μL 100 X SYBR™ Green I nucleic acid gel stain (S7563, ThermoFisher Scientific, UK) and incubated in darkness for 30 min at room temperature. Samples were quantified using 96-well V-bottom plates (3896, Corning Costar, NY, USA) on a Guava easyCyte™ HT flow cytometer (Luminex) using InCyte (v.3.4) software as previously described[95]. The following settings were used: automatic mixing of each well for 10 s at high speed before sampling, excitation by the blue laser (488 nm) and gain controls set to 12.3 for forward scatter, 1.0 for side scatter, 4.0 for green fluorescence (525/30 nm), 3.36 for yellow fluorescence (583/26 nm), and 6.17 for red fluorescence (661/ 15 nm). The final dilution factor was calculated assuming a stool density of 1.06 g/mL. The gating strategy is shown in Supplementary Fig. 4.

The remaining flowthroughs from stool preparations used for $^1$H NMR were centrifuged at 17,000 g for 5 min. DNA was extracted from resulting pellets using the Quick-DNA faecal/soil 96™ Kit (Zymo, UK) following the manufacturer's instructions. Genomic DNA libraries were prepared using the DNA Prep protocol (Illumina DNA prep, (M) Tagmentation, Illumina 20018704) according to the manufacturer's instructions. Each library was sequenced with ~8.7 million 150 bp paired-end reads at the Genewiz sequencing centre (UK). The taxonomic composition of the mice microbiota was estimated using MetaPhlAn4 with the database mpa_vJan21_CHOCOPhlAnSGB_202103[96]. The absolute cell quantification was determined by multiplying the total cell count by the predicted relative abundance (%) of the species, as derived from the metagenomic sequencing data.

**Transcriptomic activity in the caecum and in vitro culture conditions.** A < 30 mg piece from the caecum of six animals per group was dissected and incubated in RNAlater (Thermo Fisher Scientific) at 4 °C overnight to stabilise the RNA. The tissue was processed using the RNeasy Mini kit (Qiagen, UK) according to the manufacturer's instructions. For homogenisation, the material was ground using a FastPrep (MP Biomedicals, Germany) instrument for 40 s at 6 m/s with lysing matrix E tubes (MP Biomedicals). Total RNA was sequenced at Genewiz (Azenta, UK) with an Illumina NovaSeq instrument, using ~20 M 2 × 150 bp reads per sample. rRNA was depleted using NEBNext rRNA depletion kits for human/mouse/rat and bacteria species (E6310 and E7850).

Differential gene expression analyses of *B. wadsworthia* were done as previously described[97]. Briefly, adaptors were removed using bbduk (v.38.06). Clean reads were mapped against the *B. wadsworthia* genome with a minimum identity of 95% and all ambiguous mapping reported. For the analyses of the transcriptome of the SIHUMI bacteria, reference genomes were obtained from BV-BRC. The number of reads mapping to each coding sequence was estimated using featureCounts (v.2.0)[98]. Differentially expressed genes for *B. wadsworthia* were detected using Degust[99] and the Voom/Limma method, with a minimum read count of 3 and a minimum gene CPM of 3 in at least 3 samples, an FDR cut-off of 0.05 and a logFC of 0.585 (>1.5x fold change). After reconstructing the metabolic pathways for strain QI0013 using PathoLogic, the logFC of expression changes in *B. wadsworthia* between gut and in vitro conditions was imported as omics data into Pathway Tools (v.26). Pathways with the highest perturbation, reflecting the extent of pathway expression by calculating the average deviation from zero across all reactions, were identified using Pathway Tools (v.26)[27]. For the SIHUMI transcriptome, the counts were normalised to counts per million (CPM) using EdgeR implemented in Degust[99].

For analysis of host gene expression, we used the rnaflow pipeline with the Mus_musculus.GRCm38.99 reference genome[100]. This pipeline integrated downstream analyses, including pathway analysis using the Piano package[101] to identify directionally disturbed pathways, as well as gene set enrichment analysis using WebGestalt[102]. Some functional enrichments were confirmed using g:Profiler[103].

**Determination of macrophages in the liver and histology.** Livers were fixed using 4% formaldehyde in PBS for histology analysis. Sections were stained with haematoxylin and eosin as previously described[104]. To quantify the macrophages present in the liver, they were isolated as previously described[105]. Isolated macrophages were stained with CD45-APC-Cy7 (BD), CD11b-PE (BD) and F4/80-FITC (Myltenyi) antibodies at the dilutions recommended by the manufacturers (1:200 for BD antibodies and 1:100 for Miltenyi). Flow cytometry analysis was done using BD LSR-Fortessa and analysed using the FlowJo software (v.10.8.1). The gating strategy is shown in Supplementary Fig. 5.

**Transmission electron microscopy of *B. wadsworthia* in culture and caecum material.** For transmission electron microscopy, samples of *B. wadsworthia* were compared as grown with taurine as an electron acceptor in vitro or as isolated from mouse caecum. 50 mL of Postgate C media[97] supplemented with 10 mM taurine was inoculated with a stationary-phase culture of *B. wadsworthia* and grown under anaerobic conditions until late-log phase, pelleted and fixed in 1 mL of fixation buffer (100 mM sodium cacodylate pH 7.2 containing 2.5% glutaraldehyde) at room temperature for 2 h. The fixation buffer was replaced with an additional 1 mL of fixation buffer and the samples were stored at 4 °C overnight. The pellet was washed three times for 10 minutes with 1 mL 100 mM sodium cacodylate pH 7.2 to remove glutaraldehyde. The pellet was incubated with 1% osmium tetroxide for 1 h and washed with 1 mL water for 10 minutes three times to remove extracellular osmium. The pellet was dehydrated with 1 mL 50% and 70% ethanol for 10 minutes each and stored at 4 °C overnight. The pellet was resuspended in 1 mL 90% ethanol and incubated for 10 minutes. The pellet was then dehydrated three times in 100% ethanol incubating for 10 minutes each, then twice in propylene oxide incubating for 15 minutes. For embedding in resin, samples were resuspended in a 1:1 mix of propylene oxide and Agar LV Resin (Agar Scientific) and incubated for 1.5 h. Next, the pellet was infiltrated with Agar LV Resin during centrifugation at 5000 x g for 60 min before being transferred to an embedding capsule and filled remaining with resin. Embedding capsules were spun at 1000 g for 5 minutes to pellet cells to the tip and incubated at 60 °C for 20 h to polymerise.

To examine the internal structures of *B. wadsworthia* in vivo, a mouse caecum that was frozen in liquid nitrogen and stored at −80 °C was thawed on ice and fixed in fixation buffer using a ratio of 10 mL buffer per gram of tissue. The samples were placed on a rotating mixer and fixed for 2 h at room temperature. The buffer was replaced with 10 mL of fresh fixing buffer overnight at 4 °C. Samples were then washed three times for 10 minutes with 10 mL 100 mM sodium cacodylate pH 7.2 to remove glutaraldehyde. The caecums were sliced with a Derby razor blade and the internal contents were resuspended using 0.5 mL sodium cacodylate buffer. The bacteria were isolated using differential centrifugation, first removing large particulates at 1000 g for 2 minutes and then pelleting bacteria at 17,000 g for 5 minutes. The pellets were fixed with 1 mL 1% osmium tetroxide for 1 h and washed with 1 mL water for 10 minutes four times to remove extracellular osmium. The pellet was dehydrated with 1 mL 50% and 70% ethanol for 10 minutes each and stored at 4 °C overnight. Samples were resuspended in 1 mL 90% ethanol and incubated for 10 minutes. The pellet was then dehydrated three times in 100% ethanol incubating for 10 minutes each, then twice in propylene oxide incubating for 15 minutes. For embedding in resin, samples were resuspended in a 1:1 mix of propylene oxide and Agar LV Resin (Agar Scientific) and incubated for 1 h. Next, samples were resuspended in Agar LV Resin incubating for 60 min before being transferred to an embedding capsule and filled remaining with resin. Embedding capsules were spun at 1000 g for 5 minutes to pellet cells to the tip and incubated at 60 °C for 20 h to polymerise.

Ultra-thin sections of 70 nm of embedded samples were obtained with a Leica EM UC7 ultramicrotome using a DiATOME ultra 45 diamond knife and the sections were placed on a glow-discharged 600 mesh copper grid (Agar Scientific). The sections were stained with 4.5% uranyl acetate in 1% acetic acid for 45 minutes, washed with flowing water for 10 seconds, stained with Reynold's Lead citrate for 7 minutes, rinsed with flowing water for 10 seconds and air-dried. The sections were then imaged at 80 kV using a JEOL JEM-1230 transmission-electron microscope (TEM) fitted with a Gatan OneView 4K x 4K camera.

## Reporting summary

Further information on research design is available in the Nature Portfolio Reporting Summary linked to this article.

## Data availability

TraDIS sequencing data generated in this study have been deposited in the NCBI Sequence Read Archive (SRA) under the project PRJNA 1115966 [https://www.ncbi.nlm.nih.gov/bioproject/PRJNA1115966]. Metatranscriptomic sequencing data of the mice caecum are available under project PRJNA1113627 [https://www.ncbi.nlm.nih.gov/bioproject/PRJNA1113627]. The complete genome of *Bilophila wadsworthia* QI0013 was submitted under project PRJNA1085689 [https://www.ncbi.nlm.nih.gov/bioproject/PRJNA1085689]. The plasmid pQtnpATnC5 mPv2.1 is available from Keith Turner. The *E. coli* strain MFDPir can be obtained from the Institut Pasteur's Biological Resources Centre (CRBIP). Source Data underlying all figures and statistical analyses are provided with this paper. Individual-level metadata of the faecal human donor cannot be made available as they are protected by the QIB Colon Model (NCT02653001) ethical approvals. Source data are provided with this paper.

## Code availability

Custom code to create some of the graphs is available at GitHub: github.com/Isayaved/Bilophila archived on Zenodo as: https://doi.org/10.5281/zenodo.15356479 [106].

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

## Acknowledgements

We thank Tianqi Li for assisting with animal experiments, David Baker for metagenomic library preparation, and Sarah Bastkowski for advice on AlbaTradis. We thank Judith Pell and Melinda Mayer for critically commenting on this manuscript. We thank the anonymous human participant who donated a faecal sample. We acknowledge the support of the Biotechnology and Biological Sciences Research Council (BBSRC); this research was funded by the BBSRC Institute Strategic Programme Food Microbiome & Health ISP BB/X011054/1 and its constituent project BBS/E/QU/230001. LS was supported by a BBSRC Discovery Fellowship (BB/Z514445/10).

## Author contributions

L.S. designed the study, conducted experiments, analysed the data and wrote the manuscript. M.Y. and A.K.T. provided technical and intellectual support for TraDIS mutant library creation. A.G. and A.B. handled the germ-free mice. G.L.G. analysed samples for $^1$H NMR. M.M.-G., A.A. and N.B. processed livers for macrophage analysis and histology. G.M.S. provided statistical support. M.D.P. and M.W. did TEM. A.N. critically commented on the manuscript and provided conceptual input. All authors read and approved the manuscript.

## Competing interests

The authors declare no competing interests.

## Ethics Statement

Human stool collection was approved by the Quadram Institute Bioscience Human Research Governance Committee (IFR01/2015) and by the London-Westminster Research Ethics Committee (15/LO/2169). The trial was registered at clinicaltrials.gov (NCT02653001). The participant provided signed informed consent prior to donating samples. The study was conducted in accordance with the Declaration of Helsinki. Germ-free animal experiments were done under the project NB70/8929. All experiments were approved by the Animal Welfare and Ethical Review Body (University of East Anglia). All procedures were carried out following the guidelines of the National Academy of Sciences (National Institutes of Health, publication 86-23, revised 1985) and were performed within the provisions of the Animals (Scientific Procedures) Act 1986.
