## [Transparent Peer Review file · Nature Communications]

Bacterial microcompartments and energy metabolism drive gut colonization by *Bilophila wadsworthia*

Corresponding Author: Dr Lizbeth Sayavedra

Version 0:

Reviewer comments:

Reviewer #1

(Remarks to the Author)

In this manuscript entitled “Energy generation drives gut colonization by *Bilophila wadsworthia*”, Sayavedra and colleagues identified the genes that facilitate gut colonization by *Bilophila wadsworthia*, independently or in conjunction with a simplified humanized microbial consortium (SIHUMI), under a high-fat diet in germ-free mice. By using a combination of genome wide transposon mutagenesis, metatranscriptomics and untargeted metabolomics, the authors revealed the importance of gene clusters related to bacterial microcompartment formation and function, which allows *Bilophila wadsworthia* to metabolize taurine and isethionate, and of a NADH dehydrogenase that might be important for NADH recycling in this bacterium. Moreover, the authors demonstrated that the presence of a microbiota is required for this bacterium to induce intestinal and systemic damage to their host, thus nicely complementing a previous study conducted in specific pathogen-free mice (PMID: 29090023). These results will certainly make an important contribution to the understanding of the interactions between pathobionts and the host microbiota.

Overall evaluation

This manuscript is well written, with a detailed introduction addressing the major topics and an insightful discussion. The abstract nicely summarizes this interesting work. While the overall strategy employed to address the problematic is good and nicely detailed in the material and methods section, this reviewer noted some inconsistencies between the results described in the main text and the data presented in the figures and tables that need to be corrected. Moreover, reorganizing the order by which the figures and some parts of the text are presented would make this interesting story much easier to follow. I hope these comments are useful and congratulate the team on an exciting study.

Minor Comments

- 1) In all the figure legends, please describe what kind of error bars (SD, SEM?) and whether the data are presented with a mean or a median value.
- 2) Discussion:
 - a. Line 363, “this bile acid” should be “these bile acids”
 - b. The discussion section could be improved by comparing the results obtained in the present study, conducted on germ-free mice, with the results published in specific pathogen-free mice (PMID: 29090023). Indeed, the present study confirms that *B. wadsworthia* can have an impact on host health, but also demonstrates that 1) the presence of a microbiota is required for *B. wadsworthia* to become pathogenic, and 2) *B. wadsworthia* has an impact on the composition, and thus maybe the function, of the gut microbiota. This would highlight the importance of the present work.
 - c. The authors could discuss how these findings may pave the way for targeted therapeutic strategies to mitigate the adverse effects of HF diets, as suggested in the introduction.
- 3) The authors indicate, line 132, that 32 genes had lower frequencies in all host tissues. However, supplementary figure 1 seems to indicate that it is not 32, but 34 genes, while I count 35 genes with a lower frequency in all host tissues in supplementary table 1. The authors should confirm how many genes present this characteristic and make this information consistent throughout the manuscript.
- 4) Lines 160-165: Adding the QI0013 ID number would make it easier for the reader to find the genes the authors are referring to into the supplementary table 1.

- 5) Lines 192, 193, 195: By analyzing the supplementary table 1, I cannot figure out how the authors found that “More genes were up or downregulated in the SIHUMI+Bw (2,635 genes) group than the *B. wadsworthia* only group (2,138 genes) (Supplementary Table 1). From these, 161 were upregulated in gut conditions and provided a fitness benefit in SIHUMI+Bw group compared with 101 genes in the *B. wadsworthia* only group.” First, the supplementary table 1 only contains 836 genes and not all the dysregulated genes. Moreover, if positive values in the transcriptome columns refer to genes that are upregulated in the SIHUMI+Bw group or the *B. wadsworthia* only group, it seems like these values should be 321 and 319 respectively. The authors should explain how they obtained these values from this table.
- 6) The authors indicate a pathway perturbation score >1 (line 199) and of 4.05 (line 226). What data are used, and how are these scores calculated?
- 7) Supplementary table 2 should be called out lines 205-207.
- 8) Several QI0013 ID numbers are referred to in the main text but are not present in the supplementary table 1: #3998 (line 230), #823 (line 248), #459 (line 251), #3788 (line 268), #947 (line 269), #1941 (line 269). Therefore, the conclusions drawn from this information are not supported by the results presented in this table.
- 9) Line 256, the authors indicate that stool samples from mice in the SIHUMI+Bw group had a lower abundance of butyrate. However, this SCFA does not appear in supplementary table 2 or anywhere else in the manuscript. The authors should confirm that butyrate is indeed decreased in this group and ensure that the result is presented somewhere.
- 10) Lines 280-288 and supplementary figure 4: the upregulated pathways cited in the main text do not correspond to the ones indicated as significantly altered on the right part of the figure. Either the text or the figure should be updated with the correct information. Moreover, results indicating that alcoholism pathway is decreased in the *B. wadsworthia* only group (as stated line 285-288) do not appear in the supplementary figure 4 or any other figures or table in this manuscript.
- 11) Line 316: The 13 cytokine markers measured in the serum of these mice should be cited, either in the results or in the material and methods section, even if only the ones that are significantly different are presented in figure 4.
- 12) Figure 4C:
 - a. The authors should indicate that the cytokines concentrations were measured in the serum, both in the legend of Y axis and the legend of the figure.
 - b. The authors indicate that the concentration of TNF alpha is significantly different between mice colonized only with the SIHUMI consortium compared to the two other groups, however, the results seem to be highly variable within each group (0-18 pg/mL for the three groups), with a mean of 7-8 pg/mL in all groups. The authors should confirm that the observed differences are indeed significant and indicate what statistical test has been used here.
 - c. IL-1alpha and IFN-gamma seem to be increased in mice receiving the SIHUMI consortium, regardless of the presence of *B. wadsworthia*, and not in SIHUMI+Bw as indicated in the figure title and in the main text (lines 330-333). This suggests that it is not the presence of *B. wadsworthia* as interpreted by the authors, but the presence of the SIHUMI consortium that increases these cytokines.
- 13) Figure 1A presents both the timeline of the experiment and the mouse weight change. I suggest separating this two information by presenting here only the timeline and transferring the graph presenting mouse weight variation to figure 4. Similarly, figure 1D representing the result of a dextran-FITC gut permeability assay would be more suited in figure 4. Thus, figure 1 would show that the protocol used in this study leads to a successful colonization by *Bilophila wadsworthia*, and the figure 4 would show that the presence of a microbiota is deleterious for the host, only when they harbor *B. wadsworthia*.
- 14) Supplementary figure 1 nicely resumes the supplementary table 1, and therefore could be part of the main figure 2 instead.
- 15) Figure 2D nicely present how the gut-liver axis and the microbiota are involved in intestinal colonization by *B. wadsworthia*. By adding information about intestinal and liver inflammation, the author could use this figure as a summary of their study and present it as a last panel of Figure 4.
- 16) In Figure 3, panels B and C should be inverted to follow the order by which they are addressed in the main text (lines 294-299).
- 17) The supplementary figure 4 presents the pathway differentially regulated in Bw compared to SIHUMI+Bw infected mice and reveals an upregulation of pathways associated with immune response rather than a downregulation of pathways producing beneficial metabolites. However, this figure is currently addressed in a section of the manuscript presenting how *B. wadsworthia*, in presence of SIHUMI, leads to a reduction of host beneficial metabolites (lines 278-288). Thus, I suggest that this paragraph should be moved to the part describing how the interaction between *B. wadsworthia* and SIHUMI aggravates liver and systemic inflammation (starting from line 314).

Reviewer #2

(Remarks to the Author)

Using a multi-omics approach including transposon mutagenesis, this study reports the essential genes for gut colonization of *Bilophila wadsworthia* and its health impact on gnotobiotic mice with a defined human microbiota (SIHUMI) under a high fat diet.

The establishment and use of a mutant library of Bw is an important methodological advance for studying this specialized gut bacterium. However, the manuscript generally falls short of adequately explaining many of the observed findings and lacks sufficient novelty as the observations are usually very general, already reported or even contradictory to published studies. Also, the interpretations of the comparative mouse experiment are possibly ill-defined given that the germ-free mouse cohort of SIHUMI+Bw mice is already on a very different weight development trajectory before the actual start of the experiment with the gavage of strains. There is considerable unclarity if the findings on health impact are generally relevant beyond this specific mouse experiment and the single time point multi-omics analysis.

Main points:

The importance of taurine metabolism for colonization of *Bilophila* is an expected finding given the highly specialized energy metabolism (i.e. reliance on sulfite as the actual terminal electron acceptor) of *Bilophila* and other taurine-respiring bacteria in the gut. The importance and role of BMCs for taurine metabolism of *Bilophila wadsworthia* has been previously established and is also not surprising given that the BMC genes are present in the taurine metabolism gene cluster (Burrichter et al. 2021).

The possible involvement of *Bilophila* in ethanol metabolism in the gut is an interesting finding. However, the presented physiological evidence for ethanol metabolism in *Bilophila* is unclear and in parts contradictory. It is mentioned that *Bilophila* produces ethanol in BMCs but also consumes it. Ethanol utilization would be in line with reports of ethanol serving as an electron donor for taurine respiration in some *Bilophila* strains. Furthermore, what is the actual ethanol abundance in the gut? What are levels that are considered harmful?

Speculations about community interactions are not supported by data. The community dynamics and activities have not been analyzed. The focus of the analysis of multi-omic data from the final time point was solely on *Bilophila*. For example, cross feeding with the other members of the community was not experimentally tested or analyzed.

Concerns regarding the mouse experiment: Fig 1a, L314ff, L369ff Germ free mice of the SIHUMI+Bw cohort already showed considerably lower weight compared to the other two germ free mouse cohorts even BEFORE the first gavage of microbes. This puts any inference of the impact of the specific microbiotas on weight development and other aspects of host health in question. It is unclear if differential health impacts observed at the end of the experiments were already evident before the gavage of strains.

Further points:

The title seems a bit too generic as bacterial growth obviously always depends on energy conservation.

Line 41-53 The introduction is not informative enough to provide the basis for the hypothesis that *B. wadsworthia* would alter the microbial community beyond the diet alone.

Line 82ff Is the new *Bilophila wadsworthia* strain representative of the *Bilophila wadsworthia* population in humans? Please provide information on its genetic similarity and presence/absence of genes, including the ones found essential in this study, to other *Bilophila* strains.

Line 92ff. The "mono"-colonization of Bw (with *E. coli*) and strict reliance on taurine is a bit surprising given that there is less free taurine without the presence of bacteria that can deconjugate taurine-conjugated bile acids. It would be interesting to relate Bw colonization to the actual taurine concentrations in this mouse model.

Line 123ff and Fig 1c Some SIHUMI strains seem not to have colonized or only at very low abundance? Is this consistent with previous work with this gnotobiotic model?

Line 125f The reason for the reduced abundance of *B. producta*, *E. ramosum* and *C. butyricum* is not fully evident and could take the individual bacterial physiology into account.

Fig 2 b, Line 152ff The results do not provide sufficient evidence that the microcompartment and Hdr complex are ONLY essential during gut colonization. The fitness of Bw mutants in the in vitro incubations will depend on the medium composition and conditions of incubations However, only one in vitro incubation setup was tested.

Line 176 A justification for this assumption should be given.

Line 177f Significance of transcriptomic and metabolomic differences is not evident from the text.

Line 190f What is a 'stable metabolome' and what is the physiological significance of this finding?

Line 184ff. The explanation for the absence of BMC structures of *Bilophila* cultures grown with taurine is inconsistent with the study of Burrichter et al. that originally identified these BMC structures in *Bilophila* grown in vitro with taurine.

Line 186f What is known about the conditions for induction of BMCs? Is induction dependent on e.g. the concentrations of taurine and isethionate? If yes, what were the concentrations in your experiments?

Line 206f Which SHIHUMI strains are capable of deconjugation of taurine-conjugated bile acids?

Line 209ff Triggering of alternative electron donors is not described in sufficiently organized form, mixing observations, facts, and hypothesis. Absence of hydrogen and presence of formate are mentioned. Relevance of formate for *B. wadsworthia*'s energy metabolism is only based on contradicting information. Namely, one formate dehydrogenase is relevant based on transposon mutation but downregulated in the gut, while FdhABC3 is expressed but information on transposon mutagenesis is missing. The enzymes required for utilization of other possible electron donors of *B. wadsworthia*, such as lactate, were not mentioned. The paragraph could place the observations into the metabolic context of the SIHUMI gut community and provide a more specific conclusion on the microbial interaction and the influence of specific microbial species on this trigger.

Line 255ff This paragraph on short chain fatty acids lacks insight into the mechanisms but purely states observations. It remains unverified if the observed effect on the consumption of SCFA by SIHUMI and Bw is additive or synergistic or has other reasons. It is also to consider that the total cell counts between colonization significantly differs and could impact SCFA dynamics.

Line 261ff This paragraph on ethanol production and consumption lacks insight into the mechanism and its relevance to the topic of gut colonization is not fully clear. The insight that ethanol metabolism is dynamically regulated has little value without explanation of the reasons and consequences.

Line 278ff What is the relevance of the paragraph for gut colonization by *B. wadsworthia*? What is the significance of the impact of *B. wadsworthia* + *E. coli* colonization on the alcoholism pathway? . It is not clear how these observations are connected to a hypothesis or how generalizable these findings are beyond this mouse model.

Line 291 The citation of Rey et al. for inhibition of terminal oxidase is not correct and a "blind citation". The correct citation is Nicholls & Kim 1982. They also described the inhibition kinetics of cytochrome c oxidase by sulfide, it is therefore possible to mention the inhibiting concentration. Also, Rey et al. correctly referred to this publication.

Line 301ff The interpretation of H₂S dynamics is highly speculative as the microbial community dynamics and activities have not been monitored. The conclusion that the sulfur metabolism in the gut is complex is already known and rather general.

Line 334ff This interpretation is not in line with the finding that H₂S levels in SHIHUMI and SIHUMI+Bw mice are not significantly different.

The findings of the study are not discussed in the context of relevant studies regarding the importance of taurine metabolism for gut colonization of taurine-respiring bacteria (<https://www.nature.com/articles/s41467-023-41008-z>) and impact of *Bilophila* on host parameters in the SIHUMI model (<https://www.sciencedirect.com/science/article/pii/S1438422121000230>).

L407 That sulfide is not the main driver of increased gut permeability was already indicated by Rey et al 2013, who could observe that sulfide had no detectable effects on gut barrier integrity of mice with high fat diet.

Minor points:

The description of figures in the legends could be generally improved. It is not always clear what is shown.

Fig 1b Please write Tpa not TPA in accordance with protein nomenclature convention. P1-P3 and their location in the small intestine should be stated here and in the text (line 105). Explanation of the coloring is missing.

Line 89 This sentence is not fully clear. How can a mutant with an insertion of a TN5 every 88 bases survive? Or does it mean there were a total of 40,613 insertions observed in the mutant library, and the average bp distance is just a theoretical value?

Line 96 Fig 1a is referenced but the difference in weight gain between the cohorts, particularly before first gavage, is not mentioned.

Line 104 Please indicate the location of the distinct regions in the small intestine.

Line 110ff It is not fully clear how the absolute cell counts were calculated.

Fig 2a Please indicate in the figure that the blue lines are mutation frequencies. Also 'catabolyc' is misspelled. Please name the genes by exact names, like *isiA* *isiB* ..., not only by numbers

Fig 2b Please describe what the difference between essential (yellow) and the other essentials mean?

Line 128 TraDIS is introduced as the sequencing technique in the introduction and the reference list. Here it is explained as the mutagenesis method, which seems inconsistent as sequencing and transposon mutagenesis are two distinct techniques. Please clarify.

Line 144 Please explain the function of each of these four separate BMCs. It is not fully clear how they are defined. Burcher et al. 2021 should be referenced for the description of the isethionate BMCs. Also, where is it shown that *Bilophila* utilizes ethanolamine?

Line 160 *isiAB* should be *islAB*.

Line 161 Please clarify the meaning of signature enzymes. *Desulfovibrio desulfuricans* DSM642 and *Desulfovibrio alaskensis* G20 do have *IslAB* but no shell proteins, thus BMC are not essential or a signature of this enzyme (Burcher et al.).

Line 165 Please cite the study that showed the *B. wadsworthia* BMCs irrelevance for selection pressure?

Line 202 Explain what "more consistent" means in this context.

FigS3 Please provide a meaningful explanation of the figure. Figure legends should explain what the numbers indicate. "Score plot of metabolites present in faecal water samples" indicates each point represents a metabolite. If this is true, name the metabolite instead of numbering. However, it seems these are sample replicates. In this case, please explain if these are biological replicates from the same time-point. The H1-NMR method for metabolite identification should also be named.

Line 206 Please show the depletion of bile acids.

Line 220 Please indicate the different hydrogenase genes of *B. wadsworthia* and explain if these are individually non-essential or if all hydrogenase genes can be absent for gut colonization.

Line 225ff It is not fully clear what is meant by "significant disruption in formate respiration activity".

Line 230 Is *fdhABC* also essential in the TraDIS data? Is the annotation *fdhC3* correct for the gene?

Line 243 Please state the reason for the increased phosphocholine concentration and explain how the community influences it.

L246ff Is the predicted capability of *Bilophila* to metabolize choline to TMA physiologically confirmed?

Fig3a Please use the reaction equivalents of the reactions shown. Even if it is a schematic representation, it reduces the potential for misinterpretation

Fig3a H⁺ and H₂ are not the same. Please clarify if hydrogen or protons were meant

Fig3b&c As these panels are mentioned together and should be comparable, the y axis should have the same scale and unit. H₂S is depicted as μM/mg and after culling in μmol/mg. Please clarify.

Fig3c All y axes should have the same scale for comparison.

Line 264 notes *B. wadsworthia* as an ethanol degrader, while line 271 mentions *B. wadsworthia* as the source of ethanol. The ethanol dehydrogenase catalyzes both reactions. Please clarify whether *B. wadsworthia* produces or degrades ethanol in the intestine under the given conditions.

L285ff What is the physiological relevance of the increased transcription of the alcoholism pathway in mice?

L299 "dynamic"

Line 301f Please explain which species in the SIHUMI could exacerbate sulfide production.

Fig4a The quality of the images is poor and it is unclear what is visible and what the colors indicate.

Line 314ff Interleukine regulation by sulfide concentration has already been shown. In this section, observations are described, but explanations of these observations, the underlying mechanisms and the role of *B. wadsworthia* are not made sufficiently clear.

Line 349 Please explain why the Hdr complex is essential specifically in the gut and what the consequences of a missing Hdr complex are for colonization.

Line 395 Is the release of sulfide a product of the reaction of bile acid hydrolases or is it only released by further reactions?

Line 554 "fragmented"

Reviewer #3

(Remarks to the Author)

Reviewer #4

(Remarks to the Author)

I co-reviewed this manuscript with one of the reviewers who provided the listed reports. This is part of the Nature Communications initiative to facilitate training in peer review and to provide appropriate recognition for Early Career

Researchers who co-review manuscripts.

Version 1:

Reviewer comments:

Reviewer #1

(Remarks to the Author)

All my comments and suggestions made on the previous version of the manuscript have been addressed. I have no further concerns.

Congratulations to the team for this nice study that make an important contribution to the understanding of the interactions between pathobionts and the host microbiota.

Reviewer #2

(Remarks to the Author)

We thank the authors for their detailed response to our comments and questions. Our concerns regarding the different weight development trajectories of the mouse cohorts were clarified. We acknowledge that revisions of the text and figures have overall improved the manuscript. Additional experiments to address criticisms regarding the hypotheses and relevance of specific physiological properties of *Bilophila* were not performed. The presented results and explanations for the importance and role of BMCs for taurine metabolism and ethanol metabolism in *Bilophila* do not provide sufficient evidence that these findings significantly extend current knowledge.

BMCs:

Given that (i) *Bilophila* is reportedly known to rely on taurine respiration in the gut and (ii) catabolic taurine metabolism genes and BMC genes are part of the same gene cluster and co-expressed during growth on taurine (Burrichter et al. 2021), we maintain, that BMC genes are also essential for growth of *Bilophila* in the gut, is an expected finding. It appears that the microscopic proof for absence of BMCs in *Bilophila* grown in vitro with taurine is based on a single analysis and has not been reproducibly shown. It thus remains unclear if absence of BMCs in vitro is an experimental artefact or true, but maybe linked to a specific growth phase of *Bilophila*.

Metabolism of ethanol and other metabolites:

The hypothetical involvement of *Bilophila* in ethanol metabolism in the gut is now better presented. Also, the measured ethanol concentrations are now reported in the manuscript. The proposed physiological switch of *Bilophila* from use of ethanol as electron donor during e.g. taurine respiration to production of ethanol via fermentation is not supported by physiological experiments. First steps could include growth tests with the isolate. For example, is this strain capable of fermentation? Does it produce ethanol?

Furthermore, ethanol concentrations (and concentrations of other metabolites formate, lactate, pyruvate, choline and tauro-conjugated bile acids) were apparently NOT significantly different between the SIHUMI+BW group, the SIHUMI group and the BW group (Supplementary table 4). These non-significant differences in the concentration of these compounds between the mouse groups were used to hypothesize metabolic changes in *B. wadsworthia* (see below for multiple examples), which appears questionable.

Line 234ff The hypothesis of a diversified energy metabolism is based on formate and lactate concentration that were not significantly different.

Line 275ff The essential function of *fdhAB* is explained by formate concentrations, which was not significantly higher than in the other groups.

Line 286ff The non-significant differences in lactate is mentioned as increased and used to hypothesize a competition for lactate

Line 307ff The ethanol concentration is named to be increased and a diverging ethanol consumption by *B. wadsworthia* and SIHUMI is proposed on this basis. However, the differences are not significant. A shift to fermentative pathway with impact on the ethanol concentration in the gut is therefore questionable.

Line 341f Neither ethanol nor lactate concentrations were different with or without *Clostridium* and *Bacteroides* species. Thus, a contribution to an ethanol production based on these metabolite concentrations is rather speculative. Furthermore, the ethanol concentration is not significantly increased in this group, but was mentioned as increased as a result of lactate conversion.

Line 352 Choline concentration was not significantly different between the groups, but was used to justify a consumption by *B. wadsworthia*

Line 448ff The non-significant ethanol concentration was used to propose an active response to ethanol concentrations that only appears in the *B. wadsworthia* only group

Line 489f The depletion of conjugated bile acids was also observed in half of the mice in the SIHUMI group, and the bile acids concentrations were not significantly different between groups. Based on the measured concentrations, the depletion of conjugated bile acids cannot be attributed solely to *B. wadsworthia*.

Line 498 SIHUMI+Bw group is mentioned to increase ethanol concentrations with negative impact on the liver. However, the differences between the groups were not significant, so it is doubtful that such an effect can be attributed to the ethanol concentration.

Reviewer #3

(Remarks to the Author)

Reviewer #4

(Remarks to the Author)

Version 2:

Reviewer comments:

Reviewer #2

(Remarks to the Author)

We thank the authors for the detailed response and their efforts to resolve our concerns. We acknowledge the improvements and the additional growth experiments and analyses to support the hypothesis of a metabolic diversification during gut colonization by *Bilophila wadsworthia*. However, also the physiological interpretation of the new results seems contradictory and inconclusive.

Metabolism of ethanol and other metabolites: We acknowledge the performance of *B. wadsworthia* growth tests with different electron donors for taurine respiration, revealing ethanol production with formate and pyruvate. However, statistical analysis for significance of these observations was not performed. Also, ethanol and lactate were not utilized as substrates by the pure culture (concentrations of these electron donors did not decrease), although stated otherwise in the text. "Formate and lactate were rapidly consumed, with formate supporting the highest cell density of *B. wadsworthia* (Figure 4)." Ethanol did not decrease and thus it is unclear if it is the source of the produced acetaldehyde. Could acetaldehyde derive from incomplete taurine degradation? The proposed exacerbating effect on the ethanol production by *B. wadsworthia* in interaction with SIHUMI remains therefore to be proven.

It remains unclear whether the availability of electron donors' ethanol, lactate, formate, pyruvate is indeed limited and forces a metabolic diversification. Proposed changes are mainly based on transcriptomic data and are not supported by metabolomic data from faeces. It is questionable whether the measured ethanol concentrations in the liver are sufficient to justify ignoring these metabolites with non-significant differences in the faecal samples. The observed differences of the ethanol concentration in liver also appear to be minimally different and highly depending on a single mouse in the SIHUMI+BW group (based on our statistical re-analysis of the provided data, student T-test). Are these small differences in liver ethanol concentration physiologically relevant? Also, concentrations of two of eight livers seem not to be included in the analysis, raising concerns on the observed difference.

Other comments

L 129 The competition for lactate is not in accordance with the measured physiology of *B. wadsworthia* (Fig. 4) where lactate was not degraded. Other paragraphs should also be adjusted accordingly (e.g. L319)

L 238 Lactate is not consumed during the measured time, the last measured concentration is below the initial one and the intermediate concentration is still within the error of the initial measurement. Furthermore, lactate seems to have a growth inhibiting effect as the growth is lower than in the hydrogen only incubation.

L 240 Except for formate the ethanol accumulation seems not to be significantly increase above the level of the initially measured ethanol concentration, across the incubations.

L 242 The measured ethanol concentrations do not show a distinguishable decrease compared to the initial concentration. As taurine concentration decreases, it remains unclear whether ethanol is indeed consumed and converted to acetaldehyde or if acetaldehyde is a result of the taurine desulfonation reaction.

L326 Only six of the eight livers from the SIHUMI +Bw group are included. The last two ones should also be added.

Reviewer #4

(Remarks to the Author)

Version 3:

Reviewer comments:

Reviewer #5

(Remarks to the Author)

I have been asked to provide feedback on Reviewer #2's comments. After thoroughly examining the manuscript, the response letter, and the additional modifications, I believe the authors have adequately addressed the reviewers' concerns and endorse its publication.

We would like to thank the four anonymous reviewers for their extensive and thoughtful feedback on our study. Please find our point-by-point response below.

REVIEWER COMMENTS

Reviewer #1 (Remarks to the Author):

In this manuscript entitled “Energy generation drives gut colonization by *Bilophila wadsworthia*”, Sayavedra and colleagues identified the genes that facilitate gut colonization by *Bilophila wadsworthia*, independently or in conjunction with a simplified humanized microbial consortium (SIHUMI), under a high-fat diet in germ-free mice. By using a combination of genome wide transposon mutagenesis, metatranscriptomics and untargeted metabolomics, the authors revealed the importance of gene clusters related to bacterial microcompartment formation and function, which allows *Bilophila wadsworthia* to metabolize taurine and isethionate, and of a NADH dehydrogenase that might be important for NADH recycling in this bacterium. Moreover, the authors demonstrated that the presence of a microbiota is required for this bacterium to induce intestinal and systemic damage to their host, thus nicely complementing a previous study conducted in specific pathogen-free mice (PMID: 29090023). These results will certainly make an important contribution to the understanding of the interactions between pathobionts and the host microbiota.

Overall evaluation

This manuscript is well written, with a detailed introduction addressing the major topics and an insightful discussion. The abstract nicely summarizes this interesting work. While the overall strategy employed to address the problematic is good and nicely detailed in the material and methods section, this reviewer noted some inconsistencies between the results described in the main text and the data presented in the figures and tables that need to be corrected. Moreover, reorganizing the order by which the figures and some parts of the text are presented would make this interesting story much easier to follow. I hope these comments are useful and congratulate the team on an exciting study.

Minor Comments

1) In all the figure legends, please describe what kind of error bars (SD, SEM?) and whether the data are presented with a mean or a median value.

Thank you. This has been included

2) Discussion:

a. Line 363, “this bile acid” should be “these bile acids”

Corrected, thank you.

b. The discussion section could be improved by comparing the results obtained in the present study, conducted on germ-free mice, with the results published in specific pathogen-free mice (PMID: 29090023). Indeed, the present study confirms that *B. wadsworthia* can have an impact on host health, but also demonstrates that 1) the presence of a microbiota is required for *B. wadsworthia* to become pathogenic, and 2) *B. wadsworthia* has an impact on the composition, and thus maybe the function, of the gut microbiota. This would highlight the importance of the present work.

Thank you for the excellent suggestion. We want to highlight that the presence of the microbiota was necessary to **exacerbate** *B. wadsworthia*'s impact on the host, as the host's liver was also damaged when *B. wadsworthia*-only colonized the mice, although to a decreased degree.

We have modified the discussion as per your suggestion as follows:

Line 557ff. “These findings are in contrast to a previous study conducted in specific pathogen-free (SPF) mice, where *B. wadsworthia* was sufficient to induce systemic inflammatory responses and hepatosplenomegaly (enlarged liver) without significantly altering the overall gut microbiota composition based on 16S rRNA metataxonomics ¹. In our study, using germ-free mice, *B. wadsworthia* required the presence of additional microbial species to exert its full pathogenic effects under a high-fat diet. In contrast to Feng et al. ¹, mice were gavage with *B. wadsworthia* for three continuous days, followed by twice a week instead of a daily dose for a week. Quantitative profiling of microbiome changes was performed using a combination of flow cytometry cell counts and metagenomics, thus avoiding compositional bias from relative abundance measures ². Moreover, genomic differences between *B. wadsworthia* strains could also impact the outcome of the host. We used a *B. wadsworthia* strain isolated from a healthy individual, while other studies have used strains isolated from a patient with latent autoimmune diabetes ¹, from a sewage plant (DSM 11045), or other patient sources, which could influence *B. wadsworthia*'s virulence.”

We have also highlighted our key results in the “Conclusions” paragraph as follows:

Line 572ff. “Overall, our results suggest that i) under a HF diet that resembles a Western diet, the detrimental effects of *B. wadsworthia* on gut and liver health are exacerbated by interactions with the rest of the microbiome rather than the carriage of high levels of *B. wadsworthia* alone, and ii) *B. wadsworthia* can alter the gut microbial composition, and thus, the metabolic potential of the microbiome, which has implications for the host health.”

c. The authors could discuss how these findings may pave the way for targeted therapeutic strategies to mitigate the adverse effects of HF diets, as suggested in the introduction

We have added this in the concluding paragraph as follows:

Line 578ff. “Given that *B. wadsworthia*'s pathogenicity is amplified through the interaction with the microbiome, targeted interventions that modify the gut microbial composition, such as prebiotics, probiotics, and diet modifications, could potentially counteract the impact of *B. wadsworthia* when people are exposed to a high-fat Western diet. For example, a combination of inosine and probiotics including *Lacticaseibacillus rhamnosus* ³, *B. producta*, *C. butyricum* and *T. ramosa* could potentially ameliorate the impact of *B. wadsworthia* on the host health. However, future studies are required to evaluate the impact of *B. wadsworthia* under different defined microbial consortia, representative of the human microbiome from different geographic locations and under different diets.”

3) The authors indicate, line 132, that 32 genes had lower frequencies in all host tissues. However, supplementary figure 1 seems to indicate that it is not 32, but 34 genes, while I count 35 genes with a lower frequency in all host tissues in supplementary table 1. The authors should confirm how many genes present this characteristic and make this information consistent throughout the manuscript.

Thank you. It should be 34 throughout the manuscript. The text has been corrected accordingly. Supplementary Table 1 matches this information; these are the genes that have a logFC <0 in all conditions.

4) Lines 160-165: Adding the QI0013 ID number would make it easier for the reader to find the genes the authors are referring to into the supplementary table 1.

Thanks for the suggestion. We have included this information with the final locus tag given by NCBI as follows:

Line 159. “Genes encoding the isethionate sulfite-lyase and its activating enzyme, *islAB* (WCP94_002039-2040),...”

5) Lines 192, 193, 195: By analyzing the supplementary table 1, I cannot figure out how the authors found that “More genes were up or downregulated in the SIHUMI+Bw (2,635 genes) group than the *B. wadsworthia* only group (2,138 genes) (Supplementary Table 1). From these, 161 were upregulated in gut conditions and provided a fitness benefit in SIHUMI+Bw group compared with 101 genes in the *B. wadsworthia* only group.” First, the supplementary table 1 only contains 836 genes and not all the dysregulated genes. Moreover, if positive values in the transcriptome columns refer to genes that are upregulated in the SIHUMI+Bw group or the *B. wadsworthia* only group, it seems like these values should be 321 and 319 respectively. The authors should explain how they obtained these values from this table.

An additional supplementary table with all the transcriptomic information is now provided (Supplementary Table 3). The numbers are now slightly different as we have changed some parameters as detailed in the methods as follows:

“Differentially expressed genes were detected using Degust⁴ and the Voom/Limma method, with a minimum read count of 3 and a minimum gene CPM of 3 in at least 3 samples, an FDR cut-off of 0.05 and a logFC of 0.585 (>1.5x fold change).”

The RNA-seq data results can be visualised interactively under:

<https://degust.erc.monash.edu/degust/compare.html?code=b204766988adfed5afd7d6f12d71d87e#/>

To estimate the number of genes that were upregulated in gut conditions and provided a fitness benefit, we considered those genes that had a logFC >0.585 (equivalent to 1.5X expression) for RNA (more expressed in gut compared to *in vitro*), and logFC ≤1 (equivalent to 2X for the number of mutants present in the gut compared to the culture used for the gavage of the mice.

We have included in the manuscript the minimum fold change considered from the TraDIS results as follows: “The minimum logFC was 1, equivalent to 2X change in mutant frequency.”

The original counts, and the R scripts to calculate in detail the number of genes mentioned in the manuscript have been included under the following GitHub repository:

<https://github.com/lsayaved/Bilophila/>

6) The authors indicate a pathway perturbation score >1 (line 199) and of 4.05 (line 226). What data are used, and how are these scores calculated?

The pathway perturbation score (PPS) quantifies the overall expression of a pathway by calculating the average deviation from zero (following log transformation) across all reactions within the pathway.

We have included further details in the methods as follows:

Line 827ff. “After reconstructing the metabolic pathways for strain QI0013 using PathoLogic, the logFC of expression changes in *B. wadsworthia* between gut and *in vitro* conditions was imported as omics data into Pathway Tools (v.26).”

7) Supplementary table 2 should be called out lines 205-207.
Done (now Supplementary Table 3)

8) Several QI0013 ID numbers are referred to in the main text but are not present in the supplementary table 1: #3998 (line 230), #823 (line 248), #459 (line 251), #3788 (line 268), #947 (line 269), #1941 (line 269). Therefore, the conclusions drawn from this information are not supported by the results presented in this table.

As mentioned before, all the transcriptomic data is now provided as Supplementary Table 3 and has been referred to accordingly.

9) Line 256, the authors indicate that stool samples from mice in the SIHUMI+Bw group had a lower abundance of butyrate. However, this SCFA does not appear in supplementary table 2 or anywhere else in the manuscript. The authors should confirm that butyrate is indeed decreased in this group and ensure that the result is presented somewhere.

Thank you, we have updated Supplementary Table 4, which now includes the raw data for all the metabolites, detected with $^1\text{H-NMR}$. Figure 3b also shows the metabolites that are discussed in this manuscript.

10) Lines 280-288 and supplementary figure 4: the upregulated pathways cited in the main text do not correspond to the ones indicated as significantly altered on the right part of the figure. Either the text or the figure should be updated with the correct information. Moreover, results indicating that alcoholism pathway is decreased in the *B. wadsworthia* only group (as stated line 285-288) do not appear in the supplementary figure 4 or any other figures or table in this manuscript.

Thank you for pointing this out. We now provide Supplementary Table 6, which includes the differentially expressed host genes detected using DESeq2, with the enrichment analysis from WebGestalt. The text has been updated as follows:

Line 434ff. “Gene expression analysis of caecum transcriptome datasets revealed distinct host responses to colonization by *B. wadsworthia* alone versus the SIHUMI+Bw consortium. In the SIHUMI+Bw group, mice exhibited upregulation of pathways associated with immune responses, circadian rhythm, RNA processing, plasma membrane processes, lipid binding, and steroid and cholesterol metabolism (Supplementary Figure 3 and Supplementary Table 6).”

Line 440ff. “In contrast, mice colonized by *B. wadsworthia* alone showed upregulation of pathways related to nucleosome assembly, chromatin structural components, and protein refolding. Notably, the “alcoholism” pathway (mmu05034) was enriched in the *B. wadsworthia*-only group, which includes 24 genes linked to histone cluster 1 and the proto-oncogene AP-1 transcription factor subunit *fosB*, implicated in responses to addictive and compulsive behaviours⁵ (Supplementary Table 6, based on functional enrichment using WebGestalt). The caecum mice cells also overexpressed the acetaldehyde dehydrogenases *aldh1a1*, *aldh1a7*, and *aldh18a1*, genes that encode enzymes for the detoxification of aldehyde substrates into carboxylic acids (Supplementary Table 6). Considering that ethanol concentration was higher in the SIHUMI+Bw compared to the *B. wadsworthia*-only group, and the differential host response, the mice with the *B. wadsworthia*-only group actively responded to ethanol and acetaldehyde, possibly preventing their accumulation, while the SIHUMI+Bw group had a more active immune

response. The high levels of H₂S in the gut and liver of these animals could have contributed to the activation of the “alcoholism” pathway, since H₂S has been linked with cell apoptosis, DNA damage, histone modification and alterations in DNA methylation^{6,7}. This highlights how the microbiome interactions with *B. wadsworthia* can influence the host in driving liver and systemic inflammation, suggesting a link between gut microbiota-produced ethanol and metabolic comorbidities.”

11) Line 316: The 13 cytokine markers measured in the serum of these mice should be cited, either in the results or in the material and methods section, even if only the ones that are significantly different are presented in figure 4.

This has been included as follows:

Line 695-696. “The cytokines measured included IL-23, IL-1 α , IFN- γ , TNF- α , CCL2 (MCP-1), IL-12p70, IL-1 β , IL-10, IL-6, IL-27, IL-17A, IFN- β , and GM-CSF.”

12) Figure 4C: a. The authors should indicate that the cytokines concentrations were measured in the serum, both in the legend of Y axis and the legend of the figure.

We have included this information in the Y axis and the figure legend

b. The authors indicate that the concentration of TNF alpha is significantly different between mice colonized only with the SIHUMI consortium compared to the two other groups, however, the results seem to be highly variable within each group (0-18 pg/mL for the three groups), with a mean of 7-8 pg/mL in all groups. The authors should confirm that the observed differences are indeed significant and indicate what statistical test has been used here.

Thank you for pointing this out. We have now averaged the values of the technical replicates before the statistical analysis and included in our generalised linear mixed model (GLMM) the baseline weight as a co-variate. As a result, the differences previously observed for IL-1a and IFN-G are no longer significant. We have included in the source data all data points, the GLMM model, confidence intervals and p-values. The figure legend was also modified to include the analysis.

c. IL-1alpha and IFN-gamma seem to be increased in mice receiving the SIHUMI consortium, regardless of the presence of *B. wadsworthia*, and not in SIHUMI+Bw as indicated in the figure title and in the main text (lines 330-333). This suggests that it is not the presence of *B. wadsworthia* as interpreted by the authors, but the presence of the SIHUMI consortium that increases these cytokines.

Thank you. After re-analysing IL-1a and IFN-G as mentioned above, these are no longer significant differences between groups.

Line 427. “These findings align with previous research indicating the capacity of H₂S to suppress the expression of some pro-inflammatory cytokines, including IL-17A, GM-CSF and TNF- α , potentially conferring protection against acute gastrointestinal lesions⁸⁻¹⁰. However, in this context, H₂S may have an overall detrimental effect by hampering the macrophage repair response to liver parenchymal damage.”

13) Figure 1A presents both the timeline of the experiment and the mouse weight change. I suggest

separating this two information by presenting here only the timeline and transferring the graph presenting mouse weight variation to figure 4. Similarly, figure 1D representing the result of a dextran-FITC gut permeability assay would be more suited in figure 4. Thus, figure 1 would show that the protocol used in this study leads to a successful colonization by *Bilophila wadsworthia*, and the figure 4 would show that the presence of a microbiota is deleterious for the host, only when they harbor *B. wadsworthia*.

Excellent suggestion, we have incorporated the suggested changes

14) Supplementary figure 1 nicely resumes the supplementary table 1, and therefore could be part of the main figure 2 instead.

We have made this change

15) Figure 2D nicely present how the gut-liver axis and the microbiota are involved in intestinal colonization by *B. wadsworthia*. By adding information about intestinal and liver inflammation, the author could use this figure as a summary of their study and present it as a last panel of Figure 4.

Thanks for the suggestion. We have moved panel 2D to Figure 5 due to space constrictions. The figure includes information about gut barrier damage, as well as the exchange of potential metabolites that could cause liver inflammation.

16) In Figure 3, panels B and C should be inverted to follow the order by which they are addressed in the main text (lines 294-299).

Done

17) The supplementary figure 4 presents the pathway differentially regulated in Bw compared to SIHUMI+Bw infected mice and reveals an upregulation of pathways associated with immune response rather than a downregulation of pathways producing beneficial metabolites. However, this figure is currently addressed in a section of the manuscript presenting how *B. wadsworthia*, in presence of SIHUMI, leads to a reduction of host beneficial metabolites (lines 278-288). Thus, I suggest that this paragraph should be moved to the part describing how the interaction between *B. wadsworthia* and SIHUMI aggravates liver and systemic inflammation (starting from line 314). Thanks for the suggestion. We have moved the paragraph to the subsection "*B. wadsworthia* and SIHUMI interactions aggravates liver and systemic inflammation", Line 404ff.

Reviewer #2 (Remarks to the Author):

Using a multi-omics approach including transposon mutagenesis, this study reports the essential genes for gut colonization of *Bilophila wadsworthia* and its health impact on gnotobiotic mice with a defined human microbiota (SIHUMI) under a high fat diet.

The establishment and use of a mutant library of Bw is an important methodological advance for studying this specialized gut bacterium. However, the manuscript generally falls short of adequately explaining many of the observed findings and lacks sufficient novelty as the observations are usually very general, already reported or even contradictory to published studies. Also, the interpretations of the comparative mouse experiment are possibly ill-defined given that the germ-free mouse cohort of SIHUMI+Bw mice is already on a very different weight development

trajectory before the actual start of the experiment with the gavage of strains. There is considerable unclarity if the findings on health impact are generally relevant beyond this specific mouse experiment and the single time point multi-omics analysis.

Thank you for your thoughtful feedback and for highlighting the importance of the mutant library. We agree that our study represents a significant step forward in the study of *B. wadsworthia* and its interactions within the gut microbiome. Please see our responses to your specific points below.

Main points:

1. The importance of taurine metabolism for colonization of *Bilophila* is an expected finding given the highly specialized energy metabolism (i.e. reliance on sulfite as the actual terminal electron acceptor) of *Bilophila* and other taurine-respiring bacteria in the gut. The importance and role of BMCs for taurine metabolism of *Bilophila wadsworthia* has been previously established and is also not surprising given that the BMC genes are present in the taurine metabolism gene cluster (Burrichter et al. 2021).

Thank you for your feedback. While we agree that our results are supported by previous studies, including the work by Burrichter et al (2021), our findings reveal a previously unreported role of BMCs in *B. wadsworthia* colonization within the gut. Specifically, the essentiality and overexpression of the BMC gene cluster *in vivo* were not anticipated and had not been demonstrated before experimentally.

This result is particularly surprising given that, *in vitro*, *B. wadsworthia* grown in anaerobe basal broth (ABB) supplemented with taurine does not require the BMC gene cluster for growth (e.g. there were more *B. wadsworthia* mutants for the BMC gene cluster). The fact that this cluster becomes critical in the gut indicates a unique, context-dependent role for BMCs that extends beyond what was previously assumed. This observation challenges the expectation that taurine metabolism would confer the same fitness advantage in both environments.

To put this in context, *B. wadsworthia* grows minimally in ABB without taurine, which is why we supplemented the media with 10 mM taurine for bacterial gavage. *In vitro*, *B. wadsworthia* can use taurine intermediates such as sulfite, which does not need microcompartment formation¹¹. However, our data suggest that *in vivo* conditions impose additional pressures or requirements that make BMCs indispensable for efficient colonization and survival.

Figure 1. **Growth (OD₆₀₀) of *B. wadsworthia* in ABB media after 48 hrs.** *B. wadsworthia* growth was compared using anaerobe basal broth (ABB), ABB supplemented with 4 mM sulfite or ABB supplemented with 10 mM taurine. Each box plot represents the median and interquartile range of the distribution of 7 culture replicates. Results of mixed linear model analyses are shown where * = $p \leq 0.01$. Modified from Davies et al., *Bacteroides thetaiotaomicron* enhances H₂S production in *Bilophila wadsworthia*. Submitted.

We believe this highlights the novelty of our findings and provides new insights into the role of BMCs in gut colonization by *B. wadsworthia*.

2. The possible involvement of *Bilophila* in ethanol metabolism in the gut is an interesting finding. However, the presented physiological evidence for ethanol metabolism in *Bilophila* is unclear and in parts contradictory. It is mentioned that *Bilophila* produces ethanol in BMCs but also consumes it. Ethanol utilization would be in line with reports of ethanol serving as an electron donor for taurine respiration in some *Bilophila* strains. Furthermore, what is the actual ethanol abundance in the gut? What are levels that are considered harmful?

Thanks for your feedback and pointing out this was not clear. We have expanded the section on ethanol to include further analysis on the different *adh* genes of *B. wadsworthia*, those that are expressed by the SIHUMI consortia when together with *B. wadsworthia*, and the host transcriptome:

Line 307ff. “Ethanol concentrations were slightly higher in the SIHUMI+Bw group (4.3 ± 3.1 mM), compared to the *B. wadsworthia*-only group (1.7 ± 1.7 mM), and the SIHUMI group (3.6 ± 3.5 mM, $p > 0.06$) (Figure 3b). Ethanol can be used as an electron donor by *B. wadsworthia*, which would explain the lower abundance of ethanol in the *B. wadsworthia*-only group. However, ethanol could also be produced during fermentative metabolism, similar to the sulfate-reducing bacterium *N. vulgaris*^{12,13}. The Flx-Hdr complex, previously shown to be involved in ethanol production in *N. vulgaris* during pyruvate fermentation¹², may play a similar role in *B. wadsworthia*. Indeed, pyruvate was slightly increased in the SIHUMI+Bw compared to the SIHUMI group (Figure 3b and Supplementary Table 4). Moreover, the *B. wadsworthia* alcohol dehydrogenase (*adh*) gene WCP94_001710 showed increased fitness in the small intestine of the SIHUMI+Bw group (Supplementary Table 1), suggesting that at least part of the *B. wadsworthia* population may shift towards

fermentative metabolism when in competition with the SIHUMI consortia. Ethanol could then be converted to toxic acetaldehyde not only by an alcohol dehydrogenase of *B. wadsworthia*, but also by those encoded by different members of the gut microbiota ¹⁴. Acetaldehyde is known to disrupt tight junctions, thereby compromising the intestinal cell barrier and affecting the host ¹⁴.

Transcriptomic analysis revealed significant perturbations in the ethanol metabolic pathways encoded by *B. wadsworthia*, with distinct changes observed in the expression of alcohol dehydrogenases under gut conditions compared with *in vitro* conditions (in the top 5 most perturbed pathways, based on Pathways Tools). Of the nine reversible alcohol dehydrogenases encoded by *B. wadsworthia*, the aldehyde/alcohol dehydrogenase *adhE* (WCP94_001941) was 2.7x more expressed in the SIHUMI+Bw group compared to the *B. wadsworthia*-only group. AdhE is a bifunctional enzyme that converts acetyl-CoA to ethanol in two steps, playing a key role in fermentation across a range of organisms including *Klebsiella pneumoniae*, *E. coli*, and *Blautia schinkii* ^{15,16}. The *adh* whose essentiality *in vivo* means it is likely linked to the isethionate microcompartment was not differentially expressed in the caecum, but its higher fitness in the small intestine with the SIHUMI+Bw group compared to other tissues suggests that lower microbiome biomass and host signalling may influence its expression. Thus, we hypothesize that at least part of the *B. wadsworthia* population shifts to fermentation when together with the SIHUMI consortia, while by itself, it could be using ethanol as an additional energy source (Figure 3a)."

Line 448ff. (Host) "Considering that ethanol concentration was higher in the SIHUMI+Bw compared to the *B. wadsworthia*-only group, and the differential host response, the mice with the *B. wadsworthia*-only group actively responded to ethanol and acetaldehyde, possibly preventing their accumulation, while the SIHUMI+Bw group had a more active immune response."

3. Speculations about community interactions are not supported by data. The community dynamics and activities have not been analyzed. The focus of the analysis of multi-omic data from the final time point was solely on *Bilophila*. For example, cross feeding with the other members of the community was not experimentally tested or analyzed.

This is an excellent point. While cross-feeding with other members of the microbiome is an interesting area for further research, it is outside the scope of this study. However, the cross-feeding between *B. thetaiotaomicron* and *B. wadsworthia* has been investigated in a separate study, which is currently under review (Davies et al., under review).

We acknowledge that interactions with the SIHUMI consortia appear to drive the overall impact on the host. Hence, in addition to our community profiling analysis from stool samples, we have included transcriptomic analyses of the SIHUMI consortia (Supplementary Table 5). This analysis revealed the following:

- i) The gene *adhE*, which is critical for ethanol production during fermentation, was expressed by some members of the SIHUMI consortia.

Line 338ff. "Within the SIHUMI microbial consortia, *Blautia producta* expressed *adhE* at high levels in the caecum, followed by *T. ramosa*, *C. butyricum*, and *L. plantarum* (Supplementary Table 5)."

- ii) *L. plantarum* was present in the transcriptome.

Line 549ff. “Although *Lactiplantibacillus plantarum* was not detected in stool samples using metagenomics, transcriptomic analysis of caecal material revealed reads mapping to its reference genome (Supplementary Table 5). This suggests that this bacterium was present but below the detection limit with the metagenomic sequencing depth.”

iii) As predicted, *B. thetaiotaomicron* actively expressed genes involved in bile hydrolysis (choloylglycine hydrolase). Additionally, *B. producta* and *C. butyricum* also actively expressed these genes.

Line 221ff. “By using this gene as a query, we found that *C. butyricum*, *L. plantarum*, *A. caccae*, *B. producta*, and *B. longum* also encoded bile salt hydrolases (BSH, EC 3.5.1.24). Based on the transcriptome analysis of the caecum, *B. thetaiotaomicron*, followed by *B. producta* and *C. butyricum*, exhibited the most active bile salt hydrolase expression (Supplementary Table 5).”

4. Concerns regarding the mouse experiment: Fig 1a, L314ff, L369ff Germ free mice of the SIHUMI+Bw cohort already showed considerably lower weight compared to the other two germ free mouse cohorts even BEFORE the first gavage of microbes. This puts any inference of the impact of the specific microbiotas on weight development and other aspects of host health in question. It is unclear if differential health impacts observed at the end of the experiments were already evident before the gavage of strains.

We thank the reviewer for their feedback.

The mice were randomly allocated to cages, and assays were conducted with mice in a random order, with the assessor blind to the treatment status of the mouse. The experiment was repeated twice. To address the issue of potential weight differences in mice before the start of the experiment, we have included in our linear model the baseline weight (the weight before the first treatment, in grams, as a covariate in our statistical analyses for the H₂S concentrations and the inflammatory cytokines. The baseline weight per mouse was not a significant contributor to the effects observed in any of the models. You can also observe in the following graph that the weight of the mice that were later treated with the bacterial mix SIHUMI+Bw was not lower compared to the other groups.

Figure 2. Weight (g) of mice during the acclimatisation period to the high-fat diet. During this period, the mice were germ-free. "Group" refers to the gavage treatment assigned to the mice after acclimatisation to the diet.

Figure 1a previously showed mean \pm standard error. To avoid any potential confusion, we have modified Figure 4a to represent the weight (g), and 4b the weight change (%) from the day before the mice received the first oral gavage. The full timeline of the weight change (%) is shown below:

Figure 3. Percentage weight change in mice following high-fat diet acclimation, with the timeline of bacterial gavage and sample collection. Weight change was calculated relative to body weight on the last day of acclimation. Solid line represents the average weight per group.

Following the "Animal Research: Reporting of *In Vivo* Experiments (ARRIVE)" guidelines, we now include as source data the weight of each mouse across the experiment (Source data for figure 4a).

Further points:

The title seems a bit too generic as bacterial growth obviously always depends on energy conservation.

Thank you. We have revised the title to: “Bacterial Microcompartments and Energy Metabolism Drive Gut Colonization by *Bilophila wadsworthia*” to better reflect the specific findings of our study.

Additionally, we would like to highlight that our conclusions regarding energy metabolism are drawn in comparison to cultures growing under “ideal” *in vitro* conditions. This underscores the fact that energy metabolism remains highly relevant to gut colonization in more complex, competitive *in vivo* environments.

Line 41-53 The introduction is not informative enough to provide the basis for the hypothesis that *B. wadsworthia* would alter the microbial community beyond the diet alone.

We have expanded the introduction as follows:

Line 47ff. “Sulfidogenic bacteria, such as *B. wadsworthia*, have been shown to confer protection against pathogens like *Klebsiella pneumoniae* and *Citrobacter rodentium*^{17,18}. However, it remains unclear how the presence of *B. wadsworthia* affects the commensal microbiome under HF dietary conditions. Thus, we sought to determine whether the commensal microbiota was significantly altered in response to *B. wadsworthia* colonization.”

Line 82ff Is the new *Bilophila wadsworthia* strain representative of the *Bilophila wadsworthia* population in humans? Please provide information on its genetic similarity and presence/absence of genes, including the ones found essential in this study, to other *Bilophila* strains.

Excellent question. We have included a phylogenomic tree with the average nucleotide identity (ANI) and included the information in the manuscript as follows:

Results

Line 80ff. “Strain QI0013 shared >96% ANI with other *B. wadsworthia* strains, including 3.1.6 and ATCC 49260, and formed a subclade with *B. wadsworthia* 4.1.30 (>98.5% ANI) (Figure 1a). Most *B. wadsworthia* genomes and metagenome-assembled genomes (MAGs) were recovered from human samples, as previously reported from 16S rRNA amplicon sequencing¹⁷.”

Line 195ff. “Next, we compared whether the genes that had higher fitness in all tissues were shared with other *B. wadsworthia* strains. The *hdrCBA-flxDCBA* and the isethionate microcompartment gene cluster gut were conserved among representative *B. wadsworthia* strains 3.1.6, ATCC 49260, and 4.1.30 (Supplementary Table 2). Taken together, our results highlight the functional importance of bacterial microcompartments and energy conservation for gut colonization by *B. wadsworthia* species.”

Methods

Line 611ff. “A maximum-likelihood phylogenomic tree was constructed using BV-BRC with RaxML (v.8.2.12), based on the alignment of 100 single-copy genes using MAFFT¹⁹⁻²¹. The tree included *Bilophila* spp. genomes with >90% completeness. ANI values were calculated using fastANI (v.1.3.4)²². Protein sequences of QI0013 were compared against the reference strains 4.1.30, ATCC 49260, and 3.1.6 BV-BRC using BV-BRC.”

Line 92ff. The “mono”-colonization of Bw (with *E. coli*) and strict reliance on taurine is a bit

surprising given that there is less free taurine without the presence of bacteria that can deconjugate taurine-conjugated bile acids. It would be interesting to relate Bw colonization to the actual taurine concentrations in this mouse model.

In our study, tauro-conjugated bile acids were observed only in the group of mice gavaged with SIHUMI but were absent in both the SIHUMI + Bw and Bw groups. This finding suggests that *B. wadsworthia* (in association with *E. coli*) may utilize these bile acids (Figure 3b). Consequently, our data suggest that the reduction in tauro-conjugated bile acids positively correlates with the presence of *B. wadsworthia*.

The suggestion to relate Bw colonization to taurine concentrations is excellent but it will not be possible to confirm this since taurine was below the detection limit when using H¹NMR. Moreover, we do not assert that *B. wadsworthia* exhibits a strict reliance on taurine. It is possible that the formation of microcompartments is triggered by intermediates of taurine degradation, such as isethionate, which can be obtained through the diet.

Line 123ff and Fig 1c Some SIHUMI strains seem not to have colonized or only at very low abundance? Is this consistent with previous work with this gnotobiotic model?

The SIHUMI model, established by Becker et al.,²³ used rats as a model. We followed the same principle of gavaging the mice on three consecutive days with ~10⁸ cells. The study by Becker et al.²³ reported that *C. butyricum* was increased by 1.25 log₁₀ with a high-fat diet, but this species could not be detected in conventional Sprague Dawley rats, suggesting that the colonization of some of the SIHUMI species might be different depending on the rat/mouse line.

We did not detect *Lactiplantibacillus plantarum* in the metagenome, but the transcriptome did show transcripts for *Lactiplantibacillus plantarum* at a low level. Becker et al.²³ and Burkhardt, et al.²⁴ estimated the concentration of *L. plantarum* by plating, suggesting that low levels could not be quantified using qPCR. The inherent limitation of using metagenomics for metataxonomics applies in our study, including potential biases during the DNA extraction, especially against Gram-positive bacteria.

We have included this information in the Discussion as follows:

Line 549ff. “Although *Lactiplantibacillus plantarum* was not detected in stool samples using metagenomics, transcriptomic analysis of caecal material revealed reads mapping to its reference genome (Supplementary Table 5). This suggests that this bacterium was present but below the detection limit with the metagenomic sequencing depth.”

Line 125f The reason for the reduced abundance of *B. producta*, *E. ramosum* and *C. butyricum* is not fully evident and could. take the individual bacterial physiology into account.

Thank you for the suggestion. We have included a brief description as follows:

Line 126ff. “This suggests that *B. wadsworthia* had a negative impact on these taxa, potentially due to the competition for hydrogen and lactate.”

We also bring up the reduced abundance of these bacteria when mentioning that SCFA had a lower abundance in the SIHUMI+Bw group compared to the others as follows:

Line 303ff. “The reduced levels of SCFA may have resulted from the competition between the SIHUMI consortia and *B. wadsworthia* for lactate, which actively expressed lactate dehydrogenases, as well as the decreased abundance of the SCFA-producing bacteria *B. producta* and *C. butyricum* (Supplementary Table 5, Figure 1d).”

The competition for electron donors is discussed more extensively under “Competition with the microbiome triggers the use of alternative electron donors in *B. wadsworthia*”, Line 228ff.

Fig 2 b, Line 152ff The results do not provide sufficient evidence that the microcompartment and Hdr complex are ONLY essential during gut colonization. The fitness of Bw mutants in the in vitro incubations will depend on the medium composition and conditions of incubations However, only one in vitro incubation setup was tested.

Thank you for raising this point. To clarify, we do not claim that microcompartments and the Hdr complex are only essential during gut colonization. Instead, we propose that microcompartments are essential for the *in vivo* colonization. In our study, we compared the number of reads corresponding to each transposon in the mutant library grown under the same conditions as the bacterial gavage (the input pool) with the number of reads in the mouse tissues (the output pool). While additional *in vitro* conditions could potentially yield different fitness results, such comparisons would not specifically demonstrate that these genes are exclusively essential in the gut environment. In future studies, we will investigate the impact of different microbiome communities and dietary effects on *B. wadsworthia*.

Line 176 A justification for this assumption should be given.

Thanks for pointing this out. We have rephrased the paragraph to combine it with information from the discussion as follows:

L. 163ff. “The redox balance and internal NAD⁺ recycling from NADH within the isethionate microcompartments of *B. wadsworthia* remains elusive, despite previous suggestions implicating RnfC as an electron sink ¹¹. We identified a gene cluster encoding a heterodisulfide reductase linked with a flavin oxidoreductase (*hdrCBA-flxDCBA*, WCP94_001527-1532) with increased fitness in the gut (Figure 2c). In *Nitratridessulfovibrio vulgaris* Hildenborough, this cluster facilitates NADH oxidation during growth on ethanol and sulfate, leading to acetaldehyde production; or, to a lesser extent, operates in reverse during fermentation, reducing NAD⁺ and producing ethanol ¹². *N. vulgaris* encodes an alcohol dehydrogenase capable of reversibly oxidising ethanol, yielding NADH, which is then channelled to HdrABC for bifurcation to a ferredoxin and DsrC ¹². *B. wadsworthia* encodes an iron-containing alcohol dehydrogenase (*adh*) that also provided a fitness advantage in the gut, although it was not part of the *hdrCBA-flxDCBA* gene operon (WCP94_001710). In the absence of a NAD-dependent alcohol dehydrogenase in the operon of the isethionate microcompartment, this alcohol dehydrogenase may reduce acetaldehyde to ethanol using NADH, similar to that in *N. vulgaris*. Given the consistent essentiality and co-expression of *hdrABC-flxABCD* with the isethionate microcompartment gene cluster, we propose that this flavin-based electron bifurcation system plays a role in NADH recycling in *B. wadsworthia* (Figure 2c).”

Line 177ff Significance of transcriptomic and metabolomic differences is not evident from the text. We have expanded upon the relevance of transcriptomic and metabolomic analyses in several sections of the manuscript, and we hope that these additions clarify their significance. Please see lines 228ff

Line 190f What is a ‘stable metabolome’ and what is the physiological significance of this finding? We have analysed five additional metabolomic profiles (5 mice with the SIHUMI gavage). Now all mice are included and this showed that the most metabolome of the *B. wadsworthia*-only group was the most variable. We have reworded the subheading for clarity and expanded our results as follows:

Line 213ff. “The metabolite profile of faecal water from mice in the SIHUMI+Bw group showed a distinct metabolic profile with less variation between biological replicates compared to the *B. wadsworthia*-only group, as indicated by the tighter clustering in the principal component analysis (PCA) of ¹H-NMR metabolites from the same samples analysed for taxonomic profiling (Supplementary Figure 2). As expected, tauro-conjugated bile acids were depleted in stool samples from mice in the *B. wadsworthia*-only or SIHUMI+Bw groups, likely due to the utilisation of taurine derived from the deconjugation of conjugated bile acids (Figure 3b, Supplementary Table 4). Among the SIHUMI consortia, *B. thetaiotaomicron* has a bile salt hydrolase that can specifically deconjugate tauro-β-muricholic acid (BT_2086)²⁵. By using this gene as a query, we found that *C. butyricum*, *L. plantarum*, *A. caccae*, *B. producta*, and *B. longum* also encoded bile salt hydrolases (BSH, EC 3.5.1.24). Based on the metatranscriptome analysis of the caecum, *B. thetaiotaomicron*, followed by *B. producta* and *C. butyricum*, exhibited the most active bile salt hydrolase expression (Supplementary Table 5). The next subsections will discuss the key results from the integrated TraDIS, transcriptomic and metabolomic analyses.”

Line 184ff. The explanation for the absence of BMC structures of *Bilophila* cultures grown with taurine is inconsistent with the study of Burrichter et al. that originally identified these BMC structures in *Bilophila* grown *in vitro* with taurine.

Correct. We now acknowledge this in the results section as follows:

Line 191ff. “Our findings differ from those of Burrichter et al.¹¹, who observed microcompartment formation without clustering in the *B. wadsworthia* strain 3.1.6 when cultured in a hydrogen-carbonate-buffered minimal media supplemented with 20 mM taurine.”

Line 186f What is known about the conditions for induction of BMCs? Is induction dependent on e.g. the concentrations of taurine and isethionate? If yes, what were the concentrations in your experiments?

The previous study from Burrichter et al. did not demonstrate that the BMC cluster is specifically overexpressed with the addition of taurine. Currently, there is a lack of understanding regarding the conditions that induce this cluster in *Bilophila* or how it's regulated. Therefore, we find that the TEM imaging comparison of *in vitro* to *in vivo* is a compelling case for showing that BMCs form *in vivo* but not *in vitro* when grown in Postgate C supplemented with 10mM taurine, taken together with the change in fitness of the isethionate-utilization microcompartment gene cluster by *B. wadsworthia* grown on Anaerobe Basal Media supplemented with taurine 10 mM.

Line 206f Which SHIHUMI strains are capable of deconjugation of taurine-conjugated bile acids? This information has been included as follows:

Line 220ff. “Among the SIHUMI consortia, *B. thetaiotaomicron* has a bile salt hydrolase that can specifically deconjugate tauro-β-muricholic acid (BT_2086)²⁵. By using this gene as a query, we found that *C. butyricum*, *L. plantarum*, *A. caccae*, *B. producta*, and *B. longum* also encoded bile salt hydrolases (BSH, EC 3.5.1.24). Based on the metatranscriptome analysis of the caecum, *B. thetaiotaomicron*, followed by *B. producta* and *C. butyricum*, exhibited the most active bile salt hydrolase expression (Supplementary Table 5).”

Line 209ff Triggering of alternative electron donors is not described in sufficiently organized form,

mixing observations, facts, and hypothesis. Absence of hydrogen and presence of formate are mentioned. Relevance of formate for *B. wadsworthia*'s energy metabolism is only based on contradicting information. Namely, one formate dehydrogenase is relevant based on transposon mutation but downregulated in the gut, while FdhABC3 is expressed but information on transposon mutagenesis is missing. The enzymes required for utilization of other possible electron donors of *B. wadsworthia*, such as lactate, were not mentioned. The paragraph could place the observations into the metabolic context of the SIHUMI gut community and provide a more specific conclusion on the microbial interaction and the influence of specific microbial species on this trigger.

We have rephrased and expanded this section as follows:

Line 230ff. “*B. wadsworthia* can use a variety of electron donors that are the result of the fermentative process of bacteria that degrade complex polysaccharides, including hydrogen and the short-chain fatty acids, formate, lactate, and pyruvate^{13,26}. The H¹NMR metabolome of stool samples revealed subtle differences in the abundance of some of the electron donors of *B. wadsworthia* when together with the SIHUMI consortia. Formate and lactate were higher in the *B. wadsworthia*-only group compared to the other two groups (see Figure 3a-b and Supplementary Table 4). Thus, we hypothesised that in the presence of the SIHUMI consortia, *B. wadsworthia* diversifies its use of energy sources in response to competition²⁷, for example, for hydrogen.

Hydrogen is a high-energy reductant generated by colonic fermenters efficiently utilized by *B. wadsworthia*²⁶ and other gut bacteria such as the acetogen *Blautia producta*. *B. wadsworthia* encodes a soluble [FeFe] hydrogenase (WCP94_003875) and four [NiFe] uptake hydrogenases (WCP94_003605, WCP94_003816, WCP94_001421, WCP94_001823), highlighting hydrogen's role as an important energy source. Although hydrogen has been predicted to contribute to *B. wadsworthia* virulence, as it does for other gut pathogens²⁶, none of the hydrogenases had an increased fitness benefit during gut colonization (Supplementary Table 1). However, the [NiFe] hydrogenase genes *hyaAB* (WCP94_003604-3605) were essential in the initial pool of the mutant library, which precluded their direct testing in the gut environment. Interestingly, transcriptomic data showed reduced expression of this [NiFe] hydrogenase *hyaABC* in the SIHUMI+Bw group compared with the *B. wadsworthia* alone group (Supplementary Table 3), supporting that *B. wadsworthia* might broaden its electron donor usage when co-inhabiting the gut with the SIHUMI community.

Formate and hydrogen play a crucial role in electron transfer between bacterial species, with formate intracellular cycling being used for energy conservation²⁸. To use formate, bacteria rely on formate dehydrogenases, enzymes that catalyse the reversible reaction transforming formate to CO₂²⁸. Pathway analysis of gene expression perturbations in *B. wadsworthia* suggested a significant disruption in formate respiration activity, being among the top 10 perturbed pathways based on the pathway perturbation score calculated by Pathway Tools when comparing the transcriptome of *in vitro* vs. *in vivo* conditions²⁹.

Three formate dehydrogenases showed significant changes in expression: *fdhAB* (WCP94_000777-778), *fdhABC₃* (WCP94_003998-4000), and *fdol* (WCP94_000049). In the gut, *fdhAB* was under-expressed, while *fdhABC₃* and *fdol* were overexpressed compared to *in vitro* conditions (see Supplementary Table 3). Notably, *fdol* was expressed 7.6x more in the *B. wadsworthia*-only group compared to the SIHUMI+Bw group. The genes encoding *fdhABC₃* and *fdol* were essential in the input mutant pool, and could thus not be tested for fitness differences. However, *fdhAB* showed a higher fitness in the *B. wadsworthia*-only group (Supplementary Table 1).

The formate dehydrogenases *fdhAB* and *fdhABC₃* are the main *fdh* enzymes present when *Nitratidesulfovibrio vulgaris* is grown with formate, lactate and hydrogen; *fdhAB* is

upregulated during growth with hydrogen, while *fdhABC₃* is upregulated during growth with formate²⁸. When the main electron acceptor of *N. vulgaris* is depleted (sulfate), *fdhAB* can work in reverse reducing CO₂ to formate, providing a proton and electron sink, leading to formate accumulation²⁸. Given that the lack of microbial competition allowed *B. wadsworthia* to proliferate (Figure 1c), it is likely that the source of its electron acceptor (e.g., taurine-conjugated bile acids) was depleted early during colonization. This would explain why *fdhAB* was essential for colonization in the *B. wadsworthia*-only group, where formate concentrations were highest. As formate accumulated, we propose that *B. wadsworthia* began using formate as an electron donor. The elevated formate concentration in this group could have triggered the expression of *fdol*, similar to in *E. coli* which uses this enzyme for formate respiration³⁰.

Lactate, a metabolic by-product of *L. plantarum*, *Bacteroides* spp. and *Bifidobacterium* spp., can be used as an electron donor with lactate dehydrogenases. Lactate is typically found in low concentrations in healthy human faecal samples (<5 mmol L⁻¹)³¹, but high concentrations have been linked with short bowel syndrome and ulcerative colitis^{32,33}. Based on the transcriptome, *B. wadsworthia*, *B. producta*, *B. thetaiotaomicron*, and *A. caccae* actively expressed L-lactate dehydrogenases (Supplementary Table 5). The increased lactate in the *B. wadsworthia*-only group could be attributed to the absence of other microbes competing for the substrate. Lactate dehydrogenase genes (WCP94_002036-2035) had higher fitness in all the tissues of the SIHUMI+Bw group compared to the input mutant pool, as well as in the small intestine of the *B. wadsworthia*-only group. Although lactate dehydrogenase was not differentially expressed, an L-lactate permease (WCP94_002306) was more highly expressed in the gut. The higher fitness of the lactate dehydrogenase in the SIHUMI+Bw groups agrees with the need for higher diversity in the usage of electron donors when the microbiome is present. ”

Line 255ff This paragraph on short chain fatty acids lacks insight into the mechanisms but purely states observations. It remains unverified if the observed effect on the consumption of SCFA by SIHUMI and Bw is additive or synergistic or has other reasons. It is also to consider that the total cell counts between colonization significantly differs and could impact SCFA dynamics.

Interesting point. The metabolites are first normalized by the weight of the stool, so the total cell counts would not impact the results. Moreover, there wasn't a significant difference in the total cell counts between SIHUMI and SIHUMI+Bw, suggesting that the lower abundance of SCFA between these two groups is not impacted by bacterial biomass, but rather by composition.

We have included the requested information as follows:

Line 300ff. “Stool samples from mice in the SIHUMI+Bw group had a lower abundance of acetate, propionate, and butyrate, as well as the branched-chain fatty acid isovalerate than the other two groups (Figure 3b, Supplementary Table 4). The reduced levels of SCFA may have resulted from the competition between the SIHUMI consortia and *B. wadsworthia* for lactate, which actively expressed lactate dehydrogenases, as well as the decreased abundance of the SCFA-producing bacteria *B. producta* and *C. butyricum* (Supplementary Table 5, Figure 1d).”

Line 261ff This paragraph on ethanol production and consumption lacks insight into the mechanism and its relevance to the topic of gut colonization is not fully clear. The insight that ethanol metabolism is dynamically regulated has little value without explanation of the reasons and consequences.

The ethanol section was changed as described above. Our discussion has also been modified to put into context the ethanol with MASLD as follows:

Line 481ff. “In this study, ethanol metabolism emerged as an important aspect of *B. wadsworthia* physiology, particularly when interacting with the SIHUMI microbiome (Figure 5). The involvement of *B. wadsworthia* in ethanol production is relevant given that ethanol can decrease bile acid secretion and synthesis, as well as modulating macrophage activation^{34,35}. Moreover, recent studies have linked endogenous ethanol production by the gut microbiome to the pathogenesis of metabolic dysfunction-associated steatotic liver disease (MASLD) and its most severe form, metabolic dysfunction-associated steatohepatitis (MASH)³⁶. Under a milk-derived HF diet, caecal bile acid levels - including tauro-conjugated bile acids- were significantly increased in previous studies^{37,38}. However, *B. wadsworthia*'s presence led to the depletion of these bile acids³⁷. Our analysis of the host transcriptome showed overexpression of genes related to alcohol metabolism in the presence of *B. wadsworthia* compared to SHUMI+Bw. In humans, alcohol metabolism is upregulated in obese individuals suffering from metabolic MASLD^{36,39}. Moreover, ethanol concentration in stool samples from individuals with MASH has been found to be significantly higher than in control individuals (2 ± 2 mM vs. 0.95 ± 0.6 mM)⁴⁰, highlighting the potential role of gut-derived ethanol in liver pathology. MASLD is linked to metabolic comorbidities, including obesity, type 2 diabetes, hyperlipidemia, hypertension, and metabolic syndrome⁴¹.”

Line 278ff What is the relevance of the paragraph for gut colonization by *B. wadsworthia*? What is the significance of the impact of *B. wadsworthia* + *E. coli* colonization on the alcoholism pathway? . It is not clear how these observations are connected to a hypothesis or how generalizable these findings are beyond this mouse model.

As suggested by Reviewer 1, we have moved this section to “*B. wadsworthia* and SIHUMI interactions aggravate liver and systemic inflammation”.

We have expanded this section as follows:

Line 440ff. “In contrast, mice colonized by *B. wadsworthia* alone showed upregulation of pathways related to nucleosome assembly, chromatin structural components, and protein refolding. Notably, the “alcoholism” pathway (mmu05034) was enriched in the *B. wadsworthia*-only group, which includes 24 genes linked to histone cluster 1 and the proto-oncogene AP-1 transcription factor subunit *fosB*, implicated in responses to addictive and compulsive behaviours⁵ (Supplementary Table 6, based on functional enrichment using WebGestalt). The caecum mice cells also overexpressed the acetaldehyde dehydrogenases *aldh1a1*, *aldh1a7*, and *aldh18a1*, genes that encode enzymes for the detoxification of aldehyde substrates into carboxylic acids (Supplementary Table 6). Considering that ethanol concentration was higher in the SIHUMI+Bw compared to the *B. wadsworthia*-only group, and the differential host response, the mice with the *B. wadsworthia*-only group actively responded to ethanol and acetaldehyde, possibly preventing their accumulation, while the SIHUMI+Bw group had a more active immune response. The high levels of H₂S in the gut and liver of these animals could have contributed to the activation of the “alcoholism” pathway, since H₂S has been linked with cell apoptosis, DNA damage, histone modification and alterations in DNA methylation^{6,7}. This highlights how the microbiome interactions with *B. wadsworthia* can influence the host in driving liver and systemic inflammation, suggesting a link between gut microbiota-produced ethanol and metabolic comorbidities.”

Line 291 The citation of Rey at al. for inhibition of terminal oxidase is not correct and a “blind citation”. The correct citation is Nicholls & Kim 1982. They also described the inhibition kinetics

of cytochrome c oxidase by sulfide, it is therefore possible to mention the inhibiting concentration. Also, Rey et al. correctly referred to this publication.

Thanks for pointing this out. Rey et al. identified an inhibitory concentration range for cytochrome c oxidase in the millimolar (mM) range for H₂S, a detail not included in the study by Nicholls & Kim (1982). Nicholls & Kim reported that inhibition required more than 1 mol of sulfide per mol of cytochrome c oxidase (CcO) for full activity suppression. However, a more recent study by Leschelle, et al.⁴² demonstrated that 1 mM H₂S effectively inhibited the respiratory chain in colonocyte cells. We have incorporated both references into our revised statement.

Line 301ff The interpretation of H₂S dynamics is highly speculative as the microbial community dynamics and activities have not been monitored. The conclusion that the sulfur metabolism in the gut is complex is already known and rather general.

The abundance of *B. wadsworthia* correlates with the abundance of H₂S (Figure 1d and Figure 3c&d), and we monitored the H₂S concentration over time in mice stool samples. We have removed that conclusion of the paragraph and have added a reference of a follow-up study that showed that *B. thetaiotamicron* can trigger H₂S production by this specific *B. wadsworthia* strain.

Line 334ff This interpretation is not in line with the finding that H₂S levels in SHIHUMI and SIHUMI+Bw mice are not significantly different.

H₂S had consistently higher concentrations in the small intestine, colon and liver in the SIHUMI+Bw group compared to the SIHUMI group. We have recalculated the stats of the H₂S in different tissues using the log scale to address the data's high skewness, particularly due to the high H₂S levels in the Bw group, and to account for the wide range of values across different orders of magnitude. This showed that there was indeed a significant difference in the colon and liver between the SIHUMI and SIHUMI+Bw groups.

The findings of the study are not discussed in the context of relevant studies regarding the importance of taurine metabolism for gut colonization of taurine-respiring bacteria

(<https://www.nature.com/articles/s41467-023-41008-z>)

Thanks for pointing this study out. We discuss our results in the context of this study as follows: Line 474ff. "Taurine and isethionate are predominantly derived from the host's diet, with high concentrations in meat and seafood, though they are also found in algae and plants^{43,44}. In contrast to *Taurinivorans muris*, the main taurine-utilizing bacterium in the murine gut, *B. wadsworthia* can metabolize not only taurine but also isethionate and sulfite¹⁷. This broader metabolic range likely confers a competitive advantage to *B. wadsworthia* in the human gut, allowing it to occupy ecological niches where other bacteria may be limited by substrate availability."

And impact of Bilophila on host parameters in the SIHUMI model

(<https://www.sciencedirect.com/science/article/pii/S1438422121000230>).

We have compared our results to those from Burkhardt, et al. as follows:

Line 519ff. "Previous experiments used the SIHUMI and a different *B. wadsworthia* strain (DSM 11045) in gnotobiotic Il-10^{-/-} mice fed sulfoquinovose or taurocholate, and showed an initial drop in body weight (2-3%), followed by eventual recovery²⁴. Burkhardt, et al.²⁴ also reported a slight increase in gut permeability in mice gavaged with SIHUMI+Bw and fed taurocholate compared to controls (~0.75 µg/mL FITC dextran). In our study, the combination of a high-fat diet with SIHUMI+Bw resulted in higher gut permeability (1.2±0.3

µg/mL FITC dextran), suggesting that the interaction between *B. wadsworthia* and the high-fat diet exacerbates the detrimental impact on the host. This highlights the significance of the diet-microbiome interaction in modulating the pathophysiological effects of *B. wadsworthia*.”

L407 That sulfide is not the main driver of increased gut permeability was already indicated by Rey et al 2013, who could observe that sulfide had no detectable effects on gut barrier integrity of mice with high fat diet.

The study by Rey et al., 2013 showed a concentration of ~0.85 mM in caecal contents. This is comparable with the concentrations that we quantified in the colon for the SIHUMI+Bw group. However, our *B. wadsworthia*-only group, reached concentrations of 2.3 mM. We have included the information in our discussion as follows:

Line 529ff. “The severe effects on the host from the SIHUMI+Bw contrast with the findings in the *Bw*-only group, despite this mouse group having the highest *B. wadsworthia* cell numbers and the highest H₂S concentrations in the liver, gut, and stool at the last collection point before culling. The role of H₂S in health has been suggested to be concentration-dependent, with beneficial effects at lower doses, and detrimental at high concentrations ⁴⁵. Rey, et al. ⁴⁶ showed that H₂ S produced by *D. piger* under a high-fat diet (~0.85 mM in caecal contents) did not significantly impact gut permeability ⁴⁶. In our *B. wadsworthia*-only group, H₂ S levels reached 2.3±1.2 mM, but this still was not the primary cause of the increased permeability. However, these mice had the lowest concentration of ethanol from the groups testes, which could have prevented the exacerbated damage on the host. This difference highlights the importance of microbial interactions in exacerbating the pathogenic potential of *B. wadsworthia*.”

Minor points:

The description of figures in the legends could be generally improved. It is not always clear what is shown.

We have expanded the figure legends throughout the manuscript and provided source data.

Fig 1b Please write Tpa not TPA in accordance with protein nomenclature convention. P1-P3 and their location in the small intestine should be stated here and in the text (line 105). Explanation of the coloring is missing.

Thanks for pointing this out. We have adapted Figure 1. Additionally, we have changed the naming of the parts of the small intestine to duodenum, jejunum and ileum throughout the manuscript.

Line 89 This sentence is not fully clear. How can a mutant with an insertion of a TN5 every 88 bases survive? Or does it mean there were a total of 40,613 insertions observed in the mutant library, and the average bp distance is just a theoretical value?

The insertions are spread among the bacterial population that forms the mutant library. It is not a theoretical value, but rather an average. We have revised the sentence as follows:

Line 88ff. “The resulting collection of mutants, also referred to as the mutant library, had a total of 40,613 insertions. This corresponds to an average distance of one mutation every 88 bases spread among the mutant population. Only 3,722 out of 4,582 identified genes had transposon insertions, allowing these genes to be assayed.”

Line 96 Fig 1a is referenced but the difference in weight gain between the cohorts, particularly before first gavage, is not mentioned.

There was no significant or evident difference in weight gain between the cohorts before the first gavage as clarified above in response to point 4.

Line 104 Please indicate the location of the distinct regions in the small intestine.

Done

Line 110ff It is not fully clear how the absolute cell counts were calculated.

We have included a citation to the study from Vandeputte et al. (2017), which proposed the use of Quantitative Microbiome Profiling. The text was modified as follows:

Line 113ff. “We achieved this using Quantitative Microbiome Profiling (QMP)² by combining the metagenomic taxonomic profiling, and cell quantification using flow cytometry.”

Additional information was also included in the methods:

Line 799. “The gating strategy is shown in Supplementary Figure 4.”

Line 807ff. “The absolute cell quantification was determined by multiplying the total cell count by the predicted relative abundance (%) of the species, as derived from the metagenomic sequencing data.”

Fig 2a Please indicate in the figure that the blue lines are mutation frequencies. Also ‘catabolyc’ is misspelled. Please name the genes by exact names, like *islA* *islB* ..., not only by numbers
Corrected and modified Figure to include mutation frequency and the names of the genes for which a short name could be assigned.

Fig 2b Please describe what the difference between essential (yellow) and the other essentials mean?

This means that the gene was essential in the *in vitro* conditions used for making the mutant library (Anaerobe Basal Broth supplemented with taurine) and so could not be assayed as no mutants survived to be put into gavage. We have modified the text to “Higher fitness in .. compared to culture”

Line 128 TraDIS is introduced as the sequencing technique in the introduction and the reference list. Here it is explained as the mutagenesis method, which seems inconsistent as sequencing and transposon mutagenesis are two distinct techniques. Please clarify.

TraDIS stands for “Transposon directed insertion-site sequencing”, but it also refers to the mutagenesis method. We have further explained the TraDIS technique in the introduction as follows:

Line 57ff. “This technique uses the transposase enzyme to randomly insert Tn5 mini-transposons into the bacterial genome, creating a library of mutants with disrupted gene function or altered genes⁴⁷. After extracting and fragmenting the DNA, the transposon-flanking regions are selectively amplified using PCR and prepared for sequencing. The resulting sequencing data reveals the locations of transposon insertions, quantifies insertion frequencies and identifies genes essential for bacterial survival or important for specific growth conditions.”

Line 144 Please explain the function of each of these four separate BMCs. It is not fully clear how they are defined. Burchter et al. 2021 should be referenced for the description of the isethionate

BMCs. Also, where is it shown that *Bilophila* utilizes ethanolamine?

We have included the information as follows:

Line 144ff. “The genome of *B. wadsworthia* QI0013 encoded four gene clusters containing the polyhedral-body-like protein EutS, which is characteristic of BMCs. Two of these clusters encode putative ethanolamine ammonia-lyases (*eutBC*; WCP94_004157-4158 and WCP94_003786-3787), suggesting a role in ethanolamine utilization. Indeed, ethanolamine was depleted in the mice groups that had *B. wadsworthia* (Figure 3b). One cluster has previously been linked to isethionate utilization ¹¹, while the function of the fourth BMC cluster remains to be discovered. Through BMC activity, potentially toxic acetaldehyde is produced as a metabolic intermediate. The isethionate cleavage microcompartment is used to metabolise the by-product of taurine respiration through isethionate, which produces sulfite and the central carbon intermediate acetate.”

Line 160 *isiAB* should be *islAB*.

Corrected. Thank you.

Line 161 Please clarify the meaning of signature enzymes. *Desulfovibrio desulfuricans* DSM642 and *Desulfovibrio alaskensis* G20 do have *IslAB* but no shell proteins, thus BMC are not essential or a signature of this enzyme (Burrichter et al.).

Correct. Thanks for pointing this out. We have rephrased our statement as follows:

Line 155ff. “Genes encoding the isethionate sulfite-lyase and its activating enzyme, *islAB* (WCP94_002039-2040), which are key catabolic enzymes of the taurine and isethionate- inducible gene cluster that also includes genes for bacterial microcompartment shell proteins ¹¹, were essential under all conditions (Figure 2b).”

Line 165 Please cite the study that showed the *B. wadsworthia* BMCs irrelevance for selection pressure?

We have added the citation as follows:

Line 159ff. “Surprisingly, two genes encoding the microcompartment shell were essential for gut colonisation, which suggests a strong selection pressure for the maintenance of this microcompartment in *B. wadsworthia*, in contrast to a previous study in *Salmonella* Typhimurium ⁴⁸.”

Line 202 Explain what “more consistent” means in this context.

We have reworded this section as follows:

Line 213ff. “The metabolite profile of faecal water from mice in the SIHUMI+Bw group showed a distinct metabolic profile with less variation between biological replicates compared to the *B. wadsworthia*-only group, as indicated by the tighter clustering in the principal component analysis (PCA) of ¹H-NMR metabolites from the same samples analysed for taxonomic profiling (Supplementary Figure 2).”

FigS3 Please provide a meaningful explanation of the figure. Figure legends should explain what the numbers indicate. “Score plot of metabolites present in faecal water samples” indicates each point represents a metabolite. If this is true, name the metabolite instead of numbering. However, it seems these are sample replicates. In this case, please explain if these are biological replicates from the same time-point. The H1-NMR method for metabolite identification should also be named.

We have expanded the description of the figure as follows: “**Supplementary Figure 1. Score plot of the metabolome quantified using H¹-NMR from faecal water samples.** Each point on the plot represents a biological replicate collected from the same time point (refer to Figure 1a for context on experimental design). The score plot was generated using MetaboAnalyst v.5.0 based on

Principal Component Analysis (PCA), showing the variance between biological replicates in terms of their overall metabolite profiles. The clustering of points reflects the metabolic similarity between replicates. Statistical significance was assessed using PERMANOVA (Permutational Multivariate Analysis of Variance) with 999 permutations. F-value: 9.7383; R-squared: 0.44798; and p-value: 0.001, indicating significant differences between the groups.”

Line 206 Please show the depletion of bile acids.

Included as follows:

Line 217ff. “As expected, tauro-conjugated bile acids were depleted in stool samples from mice in the *B. wadsworthia*-only or SIHUMI+Bw groups, likely due to the utilisation of taurine derived from the deconjugation of conjugated bile acids (Figure 3b, Supplementary Table 4).”

Line 220 Please indicate the different hydrogenase genes of *B. wadsworthia* and explain if these are individually non-essential or if all hydrogenase genes can be absent for gut colonization.

We have included this information as follows:

Line 240ff. “*B. wadsworthia* encodes a soluble [FeFe] hydrogenase (WCP94_003875) and four [NiFe] uptake hydrogenases (WCP94_003605, WCP94_003816, WCP94_001421, WCP94_001823), highlighting hydrogen’s role as an important energy source. Although hydrogen has been predicted to contribute to *B. wadsworthia* virulence, as it does for other gut pathogens²⁶, none of the hydrogenases had an increased fitness benefit during gut colonization (Supplementary Table 1). However, the [NiFe] hydrogenase genes *hyaAB* (WCP94_003604-3605) were essential in the initial pool of the mutant library, which precluded their direct testing in the gut environment. Interestingly, transcriptomic data showed reduced expression of this [NiFe] hydrogenase *hyaABC* in the SIHUMI+Bw group compared with the *B. wadsworthia* alone group (Supplementary Table 3), supporting that *B. wadsworthia* might broaden its electron donor usage when co-inhabiting the gut with the SIHUMI community.”

Line 225ff It is not fully clear what is meant by “significant disruption in formate respiration activity”.

We have rephrased this with the following:

Line 256ff. “Pathway analysis of gene expression perturbations in *B. wadsworthia* suggested a significant disruption in formate respiration activity, being among the top 10 perturbed pathways based on the pathway perturbation score calculated by Pathway Tools when comparing the transcriptome of *in vitro* vs. *in vivo* conditions²⁹.”

Line 230 Is *fdhABC* also essential in the TraDIS data? Is the annotation *fdhC3* correct for the gene? The notation of *fdhABC₃* was used by DaSilva et al.,²⁸ as this formate dehydrogenase includes a cytochrome c (3)-like subunit.

The genes were essential in the input pool, so this information has been included as follows:

Line 264ff. “The genes encoding *fdhABC₃* and *fdol* were essential in the input mutant pool, and could thus not be tested for fitness differences. However, *fdhAB* showed a higher fitness in the *B. wadsworthia*-only group (Supplementary Table 1).”

Line 243 Please state the reason for the increased phosphocholine concentration and explain how the community influences it.

We apologise for a lack of clarity on this point. After including the five additional metabolomic profiles from the missing SIHUMI animals, our analysis showed that SIHUMI and Bw have similar levels of phosphocholine, while the SIHUMI+Bw group had a reduced level. This suggests that the reduced levels were caused by the interaction of the SIHUMI consortia with *B. wadsworthia*.

We have included the following information:

Line 347ff. “Phosphocholine was significantly lower in the SIHUMI+Bw group compared to the other two groups (Figure 3b). The secretion of phosphocholine in hepatocytes is primarily regulated by the bile acids cholic acid and deoxycholic acid ⁴⁹. Phosphocholine is a precursor of phosphatidylcholine, the main phospholipid found in bile. Phosphatidylcholine can be then hydrolyzed by some members of the gut microbiome, including *E. coli*, to choline ⁵⁰. Choline was more abundant in the *B. wadsworthia*-only group compared with the other two groups (Figure 3b and Supplementary Table 4). Choline is important in humans, contributing to cell membrane function, methyl transfer, neurotransmission and liver health, by preventing accumulation of lipids in the liver ⁵¹. Some members of the gut microbiome can convert choline to trimethylamine (TMA), which is then converted to the disease-associated trimethylamine-N-oxide (TMAO) in the liver ⁵⁰. It has been suggested that TMAO increases the risk of fatty liver disease by decreasing the bile acid pool size ⁵².”

L246ff Is the predicted capability of *Bilophila* to metabolize choline to TMA physiologically confirmed?

Not directly. It has been shown that faecal slurry has the activity and *B. wadsworthia* encodes the genes to metabolize choline. We have toned down our statement and added the references as follows:

Line 360ff. “*B. wadsworthia* has the genetic potential to metabolize choline to produce TMA via the action of the choline-TMA lyase *cutC* and its activating enzyme *cutD* (WCP94_000823-824) ^{53,54}.”

We are currently investigating this aspect of *B. wadsworthia* metabolism and the results will be published elsewhere.

Fig3a Please use the reaction equivalents of the reactions shown. Even if it is a schematic representation, it reduces the potential for misinterpretation

Thanks. We have modified the figure to include the reaction equivalents and the predicted interactions with some of the SIHUMI consortia.

Fig3a H⁺ and H₂ are not the same. Please clarify if hydrogen or protons were meant

We meant hydrogen (H₂), the figure has been modified. Thank you for pointing this out.

Fig3b&c As these panels are mentioned together and should be comparable, the y axis should have the same scale and unit. H₂S is depicted as μM/mg and after culling in μmol/mg. Please clarify.

Fig3c All y axes should have the same scale for comparison.

Thanks for your feedback. We used different methods for the quantification (methylene blue assay for stool samples and H₂S microsensor for organs). We have revised our calculations and adapted them all to μM, with a zoom-in panel for the liver as they have lower concentrations of H₂S, as expected. The Source data for the figure includes the details of the statistical analysis.

Line 264 notes *B. wadsworthias* as an ethanol degrader, while line 271 mentions *B. wadsworthia* as the source of ethanol. The ethanol dehydrogenase catalyzes both reactions. Please clarify whether *B. wadsworthia* produces or degrades ethanol in the intestine under the given conditions.

We propose that *B. wadsworthia* can use ethanol as electron donor but due to the competition for electron donors and acceptors, *B. wadsworthia* is shifting to fermentative metabolism, with ethanol as a by-product.

L285ff What is the physiological relevance of the increased transcription of the alcoholism pathway

in mice?

We have expanded our discussion as follows:

Line 490ff. “Our analysis of the host transcriptome showed overexpression of genes related to alcohol metabolism in the presence of *B. wadsworthia* compared to SIHUMI+Bw. In humans, alcohol metabolism is upregulated in obese individuals suffering from metabolic MASLD^{36,39}. Moreover, ethanol concentration in stool samples from individuals with MASH has been found to be significantly higher than in control individuals (2 ± 2 mM vs. 0.95 ± 0.6 mM)⁴⁰, highlighting the potential role of gut-derived ethanol in liver pathology. MASLD is linked to metabolic comorbidities, including obesity, type 2 diabetes, hyperlipidemia, hypertension, and metabolic syndrome⁴¹.

In our study, using mice fed a high-fat diet, the slightly higher ethanol concentration in the SIHUMI+Bw group compared to *B. wadsworthia*, combined with the depletion of SCFA compared to the other two groups, may have contributed to the negative impact on liver health, macrophage response, as well as the gut permeability. The depletion of inosine in this group could further amplify these harmful outcomes, as inosine has been shown to play a protective role in alcohol-induced liver injury in mice⁵⁵. The combined effects of increased ethanol and H₂S levels, along with reduced inosine, may have promoted hepatocyte cell death, thereby triggering macrophage infiltration and impairing their proinflammatory function under acute ethanol exposure in the livers of SIHUMI+Bw. Future research should further investigate the dietary conditions that promote the fermentation activity by *B. wadsworthia* in different gut microbiome contexts.”

L299 “dynamic”

Changed

Line 301f Please explain which species in the SIHUMI could exacerbate sulfide production.

This is an excellent point. We have a follow-up study investigating this under review, but we have included the information as follows:

Line 393ff. “Indeed, *B. thetaiotaomicron* can trigger *B. wadsworthia* to produce more H₂S when in co-culture⁵⁶.”

Fig4a The quality of the images is poor and it is unclear what is visible and what the colors indicate. Thanks for pointing this out. We have included new H&E microscopic images with improved quality in the new Fig 4C. To improve the interpretation of these images, we have included the following description in the text:

Line 412ff. “Hematoxylin and eosin staining of liver tissue from mice gavaged with either SIHUMI+Bw or *B. wadsworthia* showed increased parenchymal damage, evidenced by areas of necrosis where dark nuclei have been lost and entire cells are lost or destroyed (Figure 4d, white dashed area), which were particularly profuse in the SIHUMI+Bw group.”

The figure legend has been extended to add “Liver histology stained with H&E of representative mice. All sections were stained on the same day. White dashed area: areas of necrosis.”

Line 314ff Interleukine regulation by sulfide concentration has already been shown. In this section, observations are described, but explanations of these observations, the underlying mechanisms and the role of *B. wadsworthia* are not made sufficiently clear.

This section has been restructured as suggested by Reviewer 1. We also acknowledge that cytokine expression is affected by H₂S as follows:

Line 427ff. “These findings align with previous research indicating the capacity of H₂S to suppress the expression of some pro-inflammatory cytokines, including IL-17A, GM-CSF and TNF- α , potentially conferring protection against acute gastrointestinal lesions⁸⁻¹⁰.”

Line 349 Please explain why the Hdr complex is essential specifically in the gut and what the consequences of a missing Hdr complex are for colonization.

This complex is relevant for energy conservation when respiring taurine and isethionate. We have changed this paragraph as follows:

Line 465ff. “Our findings revealed novel insights into the genetic determinants that facilitate adaptation in *B. wadsworthia* and its function within the gut environment. In the gut, *B. wadsworthia* showed increased fitness in genes involved in taurine and isethionate respiration, microcompartment assembly and energy conservation through the HdrABC-FixABCD protein complex. The expression of these genes was higher in the gut compared with *in vitro* conditions, suggesting that *B. wadsworthia* thrives in the gut by utilizing the organosulfur compounds taurine and isethionate. This metabolic adaptability resembles the strategy used by the gut pathogen *Salmonella*, which requires not only virulence factors to colonise the gut, but also genes that provide metabolic flexibility for energy conservation⁴⁸.”

Line 395 Is the release of sulfide a product of the reaction of bile acid hydrolases or is it only released by further reactions?

It is not a direct by-product, but rather released during sulfite respiration. We have adapted the text as follows:

Line 393ff. “The availability of taurine for *B. wadsworthia* could have initially increased H₂S production in this group before numbers were decreased by competition with other bacteria⁵⁷ (Figures 1d and 3b)⁵⁷.”

Line 554 "fragmented"

Thank you. Tagmented is the correct word. Tagmentation is the initial step in library prep where unfragmented DNA is cleaved and tagged for analysis.

- 1 Feng, Z. *et al.* A human stool-derived *Bilophila wadsworthia* strain caused systemic inflammation in specific-pathogen-free mice. *Gut Pathog* **9**, 59 (2017). <https://doi.org:10.1186/s13099-017-0208-7>
- 2 Vandeputte, D. *et al.* Quantitative microbiome profiling links gut community variation to microbial load. *Nature* **551**, 507-511 (2017). <https://doi.org:10.1038/nature24460>
- 3 Natividad, J. M. *et al.* *Bilophila wadsworthia* aggravates high fat diet induced metabolic dysfunctions in mice. *Nature Communications* **9**, 2802 (2018). <https://doi.org:10.1038/s41467-018-05249-7>
- 4 Degust: interactive RNA-seq analysis (2015).
- 5 Nestler, E. J. *et al.* Δ FosB: a sustained molecular switch for addiction. *Proceedings of the National Academy of Sciences* **98** (2001-9-25). <https://doi.org:10.1073/pnas.191352698>
- 6 Kelley, J. L. *et al.* Epigenetic inheritance of DNA methylation changes in fish living in hydrogen sulfide-rich springs. *Proceedings of the National Academy of Sciences* **118** (2021). <https://doi.org:10.1073/pnas.2014929118>

- 7 Wang, Y., Yu, R., Wu, L. & Yang, G. Hydrogen sulfide signaling in regulation of cell behaviors. *Nitric Oxide* **103** (2020). <https://doi.org/10.1016/j.niox.2020.07.002>
- 8 Magierowski, M. *et al.* Cross-talk between hydrogen sulfide and carbon monoxide in the mechanism of experimental gastric ulcers healing, regulation of gastric blood flow and accompanying inflammation. *Biochemical Pharmacology* **149**, 131-142 (2018).
- 9 Ma, S. *et al.* Exogenous hydrogen sulfide ameliorates diabetes-associated cognitive decline by regulating the mitochondria-mediated apoptotic pathway and IL-23/IL-17 expression in db/db mice. *Cellular Physiology and Biochemistry* **41**, 1838-1850 (2017).
- 10 Pozzi, G. *et al.* Buffering adaptive immunity by hydrogen sulfide. *Cells* **11**, 325 (2022).
- 11 Burrichter, A. G. *et al.* Bacterial microcompartments for isethionate desulfonation in the taurine-degrading human-gut bacterium *Bilophila wadsworthia*. *BMC Microbiology* **21** (2021). <https://doi.org/10.1186/s12866-021-02386-w>
- 12 Ramos, A. R. *et al.* The FlxABCD-HdrABC proteins correspond to a novel NADH dehydrogenase/heterodisulfide reductase widespread in anaerobic bacteria and involved in ethanol metabolism in *Desulfovibrio vulgaris* Hildenborough. *Environmental Microbiology* **17**, 2288-2305 (2015). <https://doi.org/10.1111/1462-2920.12689>
- 13 Laue, H., Denger, K. & Cook, A. M. Taurine reduction in anaerobic respiration of *Bilophila wadsworthia* RZATAU. *Appl Environ Microbiol* **63**, 2016-2021 (1997). <https://doi.org/10.1128/aem.63.5.2016-2021.1997>
- 14 Ferrier, L. *et al.* Impairment of the intestinal barrier by ethanol Involves enteric microflora and mast cell activation in rodents. *The American Journal of Pathology* **168** (2006). <https://doi.org/10.2353/ajpath.2006.050617>
- 15 Oh, B.-R. *et al.* The role of aldehyde/alcohol dehydrogenase (AdhE) in ethanol production from glycerol by *Klebsiella pneumoniae*. *Journal of Industrial Microbiology and Biotechnology* **40**, 227-233 (2013). <https://doi.org/10.1007/s10295-012-1224-8>
- 16 Trischler, R., Poehlein, A., Daniel, R. & Müller, V. Ethanologenesis from glycerol by the gut acetogen *Blautia schinkii*. *Environmental Microbiology* **25**, 3577-3591 (2023).
- 17 Ye, H. *et al.* Ecophysiology and interactions of a taurine-respiring bacterium in the mouse gut. *Nature Communications* **14**, 5533 (2023). <https://doi.org/10.1038/s41467-023-41008-z>
- 18 Stacy, A. *et al.* Infection trains the host for microbiota-enhanced resistance to pathogens. *Cell* **184**, 615-627. e617 (2021).
- 19 Olson, R. D. *et al.* Introducing the Bacterial and Viral Bioinformatics Resource Center (BV-BRC): a resource combining PATRIC, IRD and ViPR. *Nucleic Acids Research* **51**, D678-d689 (2023). <https://doi.org/10.1093/nar/gkac1003>
- 20 Stamatakis, A. RAxML version 8: a tool for phylogenetic analysis and post-analysis of large phylogenies. *Bioinformatics* **30**, 1312-1313 (2014). <https://doi.org/10.1093/bioinformatics/btu033>
- 21 Katoh, K., Misawa, K., Kuma, K.-i. & Miyata, T. MAFFT: A novel method for rapid multiple sequence alignment based on fast Fourier transform. *Nucleic Acids Research* **30**, 3059-3066 (2002). <https://doi.org/10.1093/nar/gkf436>
- 22 Jain, C., Rodriguez-R, L. M., Phillippy, A. M., Konstantinidis, K. T. & Aluru, S. High throughput ANI analysis of 90K prokaryotic genomes reveals clear species

- boundaries. *Nature Communications* **9** (2018). <https://doi.org/10.1038/s41467-018-07641-9>
- 23 Becker, N., Kunath, J., Loh, G. & Blaut, M. Human intestinal microbiota: characterization of a simplified and stable gnotobiotic rat model. *Gut Microbes* **2**, 25-33 (2011).
- 24 Burkhardt, W., Rausch, T., Klopffleisch, R., Blaut, M. & Braune, A. Impact of dietary sulfolipid-derived sulfoquinovose on gut microbiota composition and inflammatory status of colitis-prone interleukin-10-deficient mice. *International Journal of Medical Microbiology* **311**, 151494 (2021). <https://doi.org/https://doi.org/10.1016/j.ijmm.2021.151494>
- 25 Yao, L. *et al.* A selective gut bacterial bile salt hydrolase alters host metabolism. *Elife* **7** (2018). <https://doi.org/10.7554/eLife.37182>
- 26 da Silva, S. M., Venceslau, S. S., Fernandes, C. L., Valente, F. M. & Pereira, I. A. Hydrogen as an energy source for the human pathogen *Bilophila wadsworthia*. *Antonie Van Leeuwenhoek* **93**, 381-390 (2008). <https://doi.org/10.1007/s10482-007-9215-x>
- 27 Ansorge, R. *et al.* Functional diversity enables multiple symbiont strains to coexist in deep-sea mussels. *Nature microbiology* **4**, 2487-2497 (2019).
- 28 Silva, S. M. d. *et al.* Function of formate dehydrogenases in *Desulfovibrio vulgaris* Hildenborough energy metabolism. *Microbiology* **159** (2013). <https://doi.org/10.1099/mic.0.067868-0>
- 29 Karp, P. D. *et al.* Pathway Tools version 23.0 update: software for pathway/genome informatics and systems biology. *Briefings in Bioinformatics* **22**, 109-126 (2021). <https://doi.org/10.1093/bib/bbz104>
- 30 Abaibou, H., Pommier, J., Benoit, S., Giordano, G. & Mandrand-Berthelot, M. A. Expression and characterization of the *Escherichia coli fdo* locus and a possible physiological role for aerobic formate dehydrogenase. *Journal of Bacteriology* **177**, 7141-7149 (1995). <https://doi.org/doi:10.1128/jb.177.24.7141-7149.1995>
- 31 Belenguer, A. *et al.* Rates of production and utilization of lactate by microbial communities from the human colon. *FEMS microbiology ecology* **77**, 107-119 (2011).
- 32 Muñoz-Tamayo, R. *et al.* Kinetic modelling of lactate utilization and butyrate production by key human colonic bacterial species. *FEMS Microbiology Ecology* **76**, 615-624 (2011). <https://doi.org/10.1111/j.1574-6941.2011.01085.x>
- 33 Hove, H., Nordgaard-Andersen, I. & Mortensen, P. B. Faecal DL-lactate concentration in 100 gastrointestinal patients. *Scandinavian journal of gastroenterology* **29**, 255-259 (1994).
- 34 Monroe, P., Vlahcevic, Z. & Swell, L. Effects of acute and chronic ethanol intake on bile acid metabolism. *Alcoholism: Clinical and Experimental Research* **5**, 92-100 (1981).
- 35 Karavitis, J., Murdoch, E. L., Gomez, C. R., Ramirez, L. & Kovacs, E. J. Acute ethanol exposure attenuates pattern recognition receptor activated macrophage functions. *J Interferon Cytokine Res* **28**, 413-422 (2008). <https://doi.org/10.1089/jir.2007.0111>
- 36 Zhu, L. X. *et al.* Characterization of Gut Microbiomes in Nonalcoholic Steatohepatitis (NASH) Patients: A Connection Between Endogenous Alcohol and NASH. *Hepatology* **57**, 601-609 (2013). <https://doi.org/10.1002/hep.26093>
- 37 Devkota, S. *et al.* Dietary-fat-induced taurocholic acid promotes pathobiont expansion and colitis in Il10^{-/-} mice. *Nature* **487**, 104-108 (2012). <https://doi.org/10.1038/nature11225>

- 38 Zheng, X. *et al.* Bile acid is a significant host factor shaping the gut microbiome of diet-induced obese mice. *BMC Biology* **15** (2017). <https://doi.org/10.1186/s12915-017-0462-7>
- 39 Baker, S. S., Baker, R. D., Liu, W., Nowak, N. J. & Zhu, L. Role of alcohol metabolism in non-alcoholic steatohepatitis. *PLoS ONE* **5**, e9570 (2010). <https://doi.org/10.1371/journal.pone.0009570>
- 40 Mbaye, B. *et al.* Increased fecal ethanol and enriched ethanol-producing gut bacteria *Limosilactobacillus fermentum*, *Enterocloster boltea*, *Mediterraneibacter gnavus* and *Streptococcus mutans* in nonalcoholic steatohepatitis. *Front Cell Infect Microbiol* **13**, 1279354 (2023). <https://doi.org/10.3389/fcimb.2023.1279354>
- 41 Younossi, Z. M. *et al.* Global epidemiology of nonalcoholic fatty liver disease—Meta-analytic assessment of prevalence, incidence, and outcomes. *Hepatology* **64**, 73-84 (2016). <https://doi.org/10.1002/hep.28431>
- 42 Leschelle, X. *et al.* Adaptative metabolic response of human colonic epithelial cells to the adverse effects of the luminal compound sulfide. *Biochimica et Biophysica Acta (BBA)-General Subjects* **1725**, 201-212 (2005).
- 43 Styp von Rekowski, K., Denger, K. & Cook, A. M. Isethionate as a product from taurine during nitrogen-limited growth of *Klebsiella oxytoca* TauN1. *Archives of Microbiology* **183**, 325-330 (2005). <https://doi.org/10.1007/s00203-005-0776-7>
- 44 McCusker, S., Buff, P. R., Yu, Z. & Fascetti, A. J. Amino acid content of selected plant, algae and insect species: a search for alternative protein sources for use in pet foods. *J Nutr Sci* **3**, e39 (2014). <https://doi.org/10.1017/jns.2014.33>
- 45 Beaumont, M. *et al.* Detrimental effects for colonocytes of an increased exposure to luminal hydrogen sulfide: The adaptive response. *Free Radical Biology and Medicine* **93**, 155-164 (2016). [https://doi.org:https://doi.org/10.1016/j.freeradbiomed.2016.01.028](https://doi.org/https://doi.org/10.1016/j.freeradbiomed.2016.01.028)
- 46 Rey, F. E. *et al.* Metabolic niche of a prominent sulfate-reducing human gut bacterium. *Proceedings of the National Academy of Sciences* **110**, 13582-13587 (2013).
- 47 Langridge, G. C. *et al.* Simultaneous assay of every *Salmonella* Typhi gene using one million transposon mutants. *Genome Res* **19**, 2308-2316 (2009). <https://doi.org/10.1101/gr.097097.109>
- 48 Chaudhuri, R. R. *et al.* Comprehensive assignment of roles for *Salmonella* typhimurium genes in intestinal colonization of food-producing animals. *PLoS genetics* **9**, e1003456 (2013).
- 49 Claus, S. P. *et al.* Systemic multicompartmental effects of the gut microbiome on mouse metabolic phenotypes. *Molecular Systems Biology* **4**, 219 (2008). [https://doi.org:https://doi.org/10.1038/msb.2008.56](https://doi.org/https://doi.org/10.1038/msb.2008.56)
- 50 Chittim, C. L., Martínez del Campo, A. & Balskus, E. P. Gut bacterial phospholipase Ds support disease-associated metabolism by generating choline. *Nature Microbiology* **4**, 155-163 (2019). <https://doi.org/10.1038/s41564-018-0294-4>
- 51 Zeisel, S. H. A brief history of choline. *Annals of Nutrition and Metabolism* **61**, 254-258 (2012). <https://doi.org/10.1159/000343120>
- 52 Koeth, R. A. *et al.* Intestinal microbiota metabolism of L-carnitine, a nutrient in red meat, promotes atherosclerosis. *Nature medicine* **19**, 576-585 (2013).
- 53 Kivenson, V. & Giovannoni, S. J. An expanded genetic code enables trimethylamine metabolism in human gut bacteria. *mSystems* **5** (2020). <https://doi.org/10.1128/mSystems.00413-20>

- 54 Craciun, S. & Balskus, E. P. Microbial conversion of choline to trimethylamine requires a glycy radical enzyme. *Proceedings of the National Academy of Sciences* **109**, 21307-21312 (2012).
- 55 Pruett, S. B. & Fan, R. Ethanol inhibits LPS-induced signaling and modulates cytokine production in peritoneal macrophages in vivo in a model for binge drinking. *BMC Immunology* **10**, 49 (2009). <https://doi.org/10.1186/1471-2172-10-49>
- 56 Davies, J., Mayer, M. J., Juge, N., Narbad, A. & Sayavedra, L. *Bacteroides thetaiotaomicron* enhances H₂S production in *Bilophila wadsworthia*. *bioRxiv*, 2024.2010.2014.618174 (2024). <https://doi.org/10.1101/2024.10.14.618174>
- 57 Jung, J. H., Kim, S.-E., Suk, K. T. & Kim, D. J. Gut microbiota-modulating agents in alcoholic liver disease: links between host metabolism and gut microbiota. *Frontiers in Medicine* **9** (2022). <https://doi.org/10.3389/fmed.2022.913842>

We thank reviewer 2 for their thorough feedback.

REVIEWER COMMENTS

Reviewer #2 (Remarks to the Author):

We thank the authors for the detailed response and their efforts to resolve our concerns. We acknowledge the improvements and the additional growth experiments and analyses to support the hypothesis of a metabolic diversification during gut colonization by *Bilophila wadsworthia*. However, also the physiological interpretation of the new results seems contradictory and inconclusive.

Metabolism of ethanol and other metabolites: We acknowledge the performance of *B. wadsworthia* growth tests with different electron donors for taurine respiration, revealing ethanol production with formate and pyruvate. However, statistical analysis for significance of these observations was not performed.

Thank you for highlighting this. We did not originally include a formal statistical analysis because other studies reporting substrate utilisation in this journal (e.g., Ye et al., 2023) also did not provide such statistics. However, in response to your feedback, we have now included the statistical analysis with the description in the figure legend as follows:

“A one-way anova was conducted in R for each incubation setup (e.g. the provided electron donor) and for each metabolite of interest with time as the predictor variable. To test for differences between each time point and the reference time (Time = 1 hour), we applied Dunnett's test via the `glht` function. * <0.05 ; ** <0.01 ; *** <0.001 .”

Also, ethanol and lactate were not utilized as substrates by the pure culture (concentrations of these electron donors did not decrease), although stated otherwise in the text. “Formate and lactate were rapidly consumed, with formate supporting the highest cell density of *B. wadsworthia* (Figure 4).“

Apologies; it should have read “formate and pyruvate,” and this has been corrected. The change in ethanol concentration is likely not detected because ethanol is present in excess. Given that 0.5 mM acetaldehyde is produced, an equivalent amount of ethanol would be expected to be consumed, but this falls outside the detection range of ^1H NMR.

Ethanol did not decrease and thus it is unclear if it is the source of the produced acetaldehyde. Could acetaldehyde derive from incomplete taurine degradation?

This is unlikely. If this would be the case, then you would expect to see acetaldehyde accumulation with the other electron donors when coupled with taurine. Moreover, the direct activity of alcohol dehydrogenases can convert ethanol to acetaldehyde in one step, making it the most likely source of acetaldehyde.

Finally, the metabolome was done on supernatant samples and not the cell pellets, which means that the acetaldehyde would be most likely excreted and not contained within the cell (and within microcompartments, which is what would be expected from the taurine metabolism).

We have clarified that the ^1H NMR quantification was done on the supernatant as follow (Line 269ff):

“To investigate the metabolic flexibility of *B. wadsworthia*, we examined its growth *in vitro* to characterise the utilisation of some of these electron donors and measure the metabolite changes in the supernatant using ^1H NMR.”

And clarify the most likely source of acetaldehyde when ethanol is provided as electron donor as follows (Line 276ff):

“When ethanol was provided as the primary energy source, acetaldehyde accumulated progressively in the supernatant, suggesting the induction of ethanol metabolism. The absence of intracellular retention indicates that acetaldehyde was not sequestered within microcompartments, but rather excreted into the surrounding environment.”

The proposed exacerbating effect on the ethanol production by *B. wadsworthia* in interaction with SIHUMI remains therefore to be proven.

Our growth experiments with different electron donors demonstrate that *B. wadsworthia* can produce ethanol depending on the available electron donor. Furthermore, we have shown that acetaldehyde, a direct metabolite of alcohol dehydrogenase activity, accumulates when *B. wadsworthia* is grown on ethanol. These findings provide clear evidence of *B. wadsworthia*'s capacity for ethanol metabolism and support the slightly increased ethanol levels *in vivo*.

It remains unclear whether the availability of electron donors' ethanol, lactate, formate, pyruvate is indeed limited and forces a metabolic diversification.

Our data demonstrate that the presence of various electron donors, such as ethanol, lactate, formate, and pyruvate, promotes *B. wadsworthia*'s growth at different rates. This suggests that the availability of these energy sources is an important factor in driving metabolic diversification. This study provides a foundation for future research exploring how microbial metabolic adaptation is shaped by electron donor availability in complex host-associated ecosystems.

Proposed changes are mainly based on transcriptomic data and are not supported by metabolomic data from faeces.

To clarify, our conclusions are supported by multiple complementary datasets, including gene fitness differences, microbial transcriptomics, and the host's transcriptomic response. The advantage of considering TraDIS fitness differences over other techniques is that it provides insight into the local microenvironment encountered by the bacteria during colonisation. Additionally, our data suggests that ethanol levels in the liver were, on average, higher in the SIHUMI+*B. wadsworthia* group, providing metabolomic evidence that aligns with our findings. These results indicate that systemic metabolic changes occur beyond what is detected in faecal metabolomics alone.

It is questionable whether the measured ethanol concentrations in the liver are sufficient to justify ignoring these metabolites with non-significant differences in the faecal samples. We are not disregarding the metabolite concentrations in stool samples. Rather, we identified a trend in stool and extended our analysis to the liver, where ethanol is known to have physiological effects. Given the well-documented systemic impact of ethanol metabolism, liver measurements provide relevant and complementary insights beyond faecal metabolomics alone.

The observed differences of the ethanol concentration in liver also appear to be minimally different and highly depending on a single mouse in the SIHUMI+BW group (based on our statistical re-analysis of the provided data, student T-test). Are these small differences in liver ethanol concentration physiologically relevant?

We acknowledge that depending on the statistical test, the results might differ. However, they still support the hypothesis that microbial fermentation can influence ethanol levels on the host, with potential implications on hepatic metabolism and gut barrier integrity.

However, we have added the following statement in the discussion:

“However, since ethanol can be rapidly converted to acetaldehyde and subsequently to acetate, it remains unclear to what extent *B. wadsworthia* contributes to a significant ethanol accumulation. Future studies should examine how ethanol consumption and production fluctuate under different microbiome compositions and dietary conditions to better delineate its role.”

Also, concentrations of two of eight livers seem not to be included in the analysis, raising concerns on the observed difference.

Two mice from the SIHUMI+*B. wadsworthia* group died before the end of the experiment, and therefore, liver material was not available for testing (this is mentioned in Line 772, and has also been incorporated into the Figure 5 legend).

Other comments

L 129 The competition for lactate is not in accordance with the measured physiology of *B. wadsworthia* (Fig. 4) where lactate was not degraded. Other paragraphs should also be adjusted accordingly (e.g. L319)

Lactate was significantly increased when *B. wadsworthia* was grown with pyruvate, which suggests that lactate can also be produced. Lactate was provided in excess and the variation in the measurements was high, likely due to the production and consumption of this metabolite. However, we agree that we do not have enough evidence to show the lactate degradation. Accordingly, we have updated the following sections:

Line 128-129: “This suggests that *B. wadsworthia* had a negative impact on these taxa, potentially due to competition for hydrogen, formate or pyruvate.”

Line 329ff: “Lactate did not play a major role in promoting *B. wadsworthia*’s growth (Figure 4), and, at a concentration of 53 mM, even inhibited its growth. The higher fitness of the lactate dehydrogenase in the SIHUMI+Bw groups might have helped mitigate the growth-limiting effects of lactate.”

Line 353ff: “Although *B. wadsworthia* may not have directly utilised lactate as an energy source, its lactate dehydrogenase activity could have contributed to lactate depletion. Additionally, the decreased abundance of the SCFA-producing bacteria *B. producta* and *C. butyricum* may have further influenced SCFA levels (Supplementary Table 5, Figure 1d).”

Figure 3: We have adapted panel a) to show that the activity of lactate dehydrogenases is reversible.

L 238 Lactate is not consumed during the measured time, the last measured concentration is below the initial one and the intermediate concentration is still within the error of the initial measurement.

We have rephrased to (Line 240ff): “Formate and pyruvate were rapidly consumed, with formate supporting the highest cell density of *B. wadsworthia* (Figure 4).”

Furthermore, lactate seems to have a growth inhibiting effect as the growth is lower than in the hydrogen only incubation.

We have included this in the results as follows (Line 247ff):

“Unexpectedly, lactate slowed the growth of *B. wadsworthia*, and during growth on pyruvate, lactate concentrations significantly increased over time.”

L 240 Except for formate the ethanol accumulation seems not to be significantly increase above the level of the initially measured ethanol concentration, across the incubations.

Thank you for pointing this out. We re-examined the data and confirmed that, with formate and pyruvate as electron donors, ethanol levels increased **significantly** compared to the first time point. While lactate showed a slight upward trend, it did not reach statistical significance ($p = 0.06$), so we removed it from the text.

This has been changed as follows (Line 242):

“Over time, a significant accumulation of ethanol was observed during growth on formate and pyruvate as electron donors.”

L 242 The measured ethanol concentrations do not show a distinguishable decrease compared to the initial concentration. As taurine concentration decreases, it remains unclear whether ethanol is indeed consumed and converted to acetaldehyde or if acetaldehyde is a result of the taurine desulfonation reaction.

We have clarified the text as follows: “When ethanol was provided as the primary energy source, acetaldehyde accumulated progressively in the supernatant, suggesting the induction of ethanol metabolism. The absence of intracellular retention suggests that acetaldehyde was not sequestered within microcompartments, and therefore was not the result of taurine desulfonation.”

L326 Only six of the eight livers from the SIHUMI +Bw group are included. The last two ones should also be added.

We only had six livers for the SIHUMI+Bw group since the other two mice died before the end of the experiment, and the organs were not harvested. This is described in Line 739: “two mice of SIHUMI+Bw group died on day 40” and is consistent with all other analyses. We have now included in the SourceData file the number of animals “Groups: SIHUMI+Bw (N=6), SIHUMI (N=10), Bw (N=9)”. For clarity, this is now also included in the legend of Figure 5.

We thank reviewer 2 for their thorough feedback.

REVIEWER COMMENTS

Reviewer #2 (Remarks to the Author):

We thank the authors for the detailed response and their efforts to resolve our concerns. We acknowledge the improvements and the additional growth experiments and analyses to support the hypothesis of a metabolic diversification during gut colonization by *Bilophila wadsworthia*. However, also the physiological interpretation of the new results seems contradictory and inconclusive.

Metabolism of ethanol and other metabolites: We acknowledge the performance of *B. wadsworthia* growth tests with different electron donors for taurine respiration, revealing ethanol production with formate and pyruvate. However, statistical analysis for significance of these observations was not performed.

Thank you for highlighting this. We did not originally include a formal statistical analysis because other studies reporting substrate utilisation in this journal (e.g., Ye et al., 2023) also did not provide such statistics. However, in response to your feedback, we have now included the statistical analysis with the description in the figure legend as follows:

“A one-way anova was conducted in R for each incubation setup (e.g. the provided electron donor) and for each metabolite of interest with time as the predictor variable. To test for differences between each time point and the reference time (Time = 1 hour), we applied Dunnett's test via the `glht` function. * <0.05 ; ** <0.01 ; *** <0.001 .”

Also, ethanol and lactate were not utilized as substrates by the pure culture (concentrations of these electron donors did not decrease), although stated otherwise in the text. “Formate and lactate were rapidly consumed, with formate supporting the highest cell density of *B. wadsworthia* (Figure 4).“

Apologies; it should have read “formate and pyruvate,” and this has been corrected. The change in ethanol concentration is likely not detected because ethanol is present in excess. Given that 0.5 mM acetaldehyde is produced, an equivalent amount of ethanol would be expected to be consumed, but this falls outside the detection range of ^1H NMR.

Ethanol did not decrease and thus it is unclear if it is the source of the produced acetaldehyde. Could acetaldehyde derive from incomplete taurine degradation?

This is unlikely. If this would be the case, then you would expect to see acetaldehyde accumulation with the other electron donors when coupled with taurine. Moreover, the direct activity of alcohol dehydrogenases can convert ethanol to acetaldehyde in one step, making it the most likely source of acetaldehyde.

Finally, the metabolome was done on supernatant samples and not the cell pellets, which means that the acetaldehyde would be most likely excreted and not contained within the cell (and within microcompartments, which is what would be expected from the taurine metabolism).

We have clarified that the ^1H NMR quantification was done on the supernatant as follow (Line 269ff):

“To investigate the metabolic flexibility of *B. wadsworthia*, we examined its growth *in vitro* to characterise the utilisation of some of these electron donors and measure the metabolite changes in the supernatant using ^1H NMR.”

And clarify the most likely source of acetaldehyde when ethanol is provided as electron donor as follows (Line 276ff):

“When ethanol was provided as the primary energy source, acetaldehyde accumulated progressively in the supernatant, suggesting the induction of ethanol metabolism. The absence of intracellular retention indicates that acetaldehyde was not sequestered within microcompartments, but rather excreted into the surrounding environment.”

The proposed exacerbating effect on the ethanol production by *B. wadsworthia* in interaction with SIHUMI remains therefore to be proven.

Our growth experiments with different electron donors demonstrate that *B. wadsworthia* can produce ethanol depending on the available electron donor. Furthermore, we have shown that acetaldehyde, a direct metabolite of alcohol dehydrogenase activity, accumulates when *B. wadsworthia* is grown on ethanol. These findings provide clear evidence of *B. wadsworthia*'s capacity for ethanol metabolism and support the slightly increased ethanol levels *in vivo*.

It remains unclear whether the availability of electron donors' ethanol, lactate, formate, pyruvate is indeed limited and forces a metabolic diversification.

Our data demonstrate that the presence of various electron donors, such as ethanol, lactate, formate, and pyruvate, promotes *B. wadsworthia*'s growth at different rates. This suggests that the availability of these energy sources is an important factor in driving metabolic diversification. This study provides a foundation for future research exploring how microbial metabolic adaptation is shaped by electron donor availability in complex host-associated ecosystems.

Proposed changes are mainly based on transcriptomic data and are not supported by metabolomic data from faeces.

To clarify, our conclusions are supported by multiple complementary datasets, including gene fitness differences, microbial transcriptomics, and the host's transcriptomic response. The advantage of considering TraDIS fitness differences over other techniques is that it provides insight into the local microenvironment encountered by the bacteria during colonisation. Additionally, our data suggests that ethanol levels in the liver were, on average, higher in the SIHUMI+*B. wadsworthia* group, providing metabolomic evidence that aligns with our findings. These results indicate that systemic metabolic changes occur beyond what is detected in faecal metabolomics alone.

It is questionable whether the measured ethanol concentrations in the liver are sufficient to justify ignoring these metabolites with non-significant differences in the faecal samples. We are not disregarding the metabolite concentrations in stool samples. Rather, we identified a trend in stool and extended our analysis to the liver, where ethanol is known to have physiological effects. Given the well-documented systemic impact of ethanol metabolism, liver measurements provide relevant and complementary insights beyond faecal metabolomics alone.

The observed differences of the ethanol concentration in liver also appear to be minimally different and highly depending on a single mouse in the SIHUMI+BW group (based on our statistical re-analysis of the provided data, student T-test). Are these small differences in liver ethanol concentration physiologically relevant?

We acknowledge that depending on the statistical test, the results might differ. However, they still support the hypothesis that microbial fermentation can influence ethanol levels on the host, with potential implications on hepatic metabolism and gut barrier integrity.

However, we have added the following statement in the discussion:

“However, since ethanol can be rapidly converted to acetaldehyde and subsequently to acetate, it remains unclear to what extent *B. wadsworthia* contributes to a significant ethanol accumulation. Future studies should examine how ethanol consumption and production fluctuate under different microbiome compositions and dietary conditions to better delineate its role.”

Also, concentrations of two of eight livers seem not to be included in the analysis, raising concerns on the observed difference.

Two mice from the SIHUMI+*B. wadsworthia* group died before the end of the experiment, and therefore, liver material was not available for testing (this is mentioned in Line 772, and has also been incorporated into the Figure 5 legend).

Other comments

L 129 The competition for lactate is not in accordance with the measured physiology of *B. wadsworthia* (Fig. 4) where lactate was not degraded. Other paragraphs should also be adjusted accordingly (e.g. L319)

Lactate was significantly increased when *B. wadsworthia* was grown with pyruvate, which suggests that lactate can also be produced. Lactate was provided in excess and the variation in the measurements was high, likely due to the production and consumption of this metabolite. However, we agree that we do not have enough evidence to show the lactate degradation. Accordingly, we have updated the following sections:

Line 128-129: “This suggests that *B. wadsworthia* had a negative impact on these taxa, potentially due to competition for hydrogen, formate or pyruvate.”

Line 329ff: “Lactate did not play a major role in promoting *B. wadsworthia*’s growth (Figure 4), and, at a concentration of 53 mM, even inhibited its growth. The higher fitness of the lactate dehydrogenase in the SIHUMI+Bw groups might have helped mitigate the growth-limiting effects of lactate.”

Line 353ff: “Although *B. wadsworthia* may not have directly utilised lactate as an energy source, its lactate dehydrogenase activity could have contributed to lactate depletion. Additionally, the decreased abundance of the SCFA-producing bacteria *B. producta* and *C. butyricum* may have further influenced SCFA levels (Supplementary Table 5, Figure 1d).”

Figure 3: We have adapted panel a) to show that the activity of lactate dehydrogenases is reversible.

L 238 Lactate is not consumed during the measured time, the last measured concentration is below the initial one and the intermediate concentration is still within the error of the initial measurement.

We have rephrased to (Line 240ff): “Formate and pyruvate were rapidly consumed, with formate supporting the highest cell density of *B. wadsworthia* (Figure 4).”

Furthermore, lactate seems to have a growth inhibiting effect as the growth is lower than in the hydrogen only incubation.

We have included this in the results as follows (Line 247ff):

“Unexpectedly, lactate slowed the growth of *B. wadsworthia*, and during growth on pyruvate, lactate concentrations significantly increased over time.”

L 240 Except for formate the ethanol accumulation seems not to be significantly increase above the level of the initially measured ethanol concentration, across the incubations.

Thank you for pointing this out. We re-examined the data and confirmed that, with formate and pyruvate as electron donors, ethanol levels increased **significantly** compared to the first time point. While lactate showed a slight upward trend, it did not reach statistical significance ($p = 0.06$), so we removed it from the text.

This has been changed as follows (Line 242):

“Over time, a significant accumulation of ethanol was observed during growth on formate and pyruvate as electron donors.”

L 242 The measured ethanol concentrations do not show a distinguishable decrease compared to the initial concentration. As taurine concentration decreases, it remains unclear whether ethanol is indeed consumed and converted to acetaldehyde or if acetaldehyde is a result of the taurine desulfonation reaction.

We have clarified the text as follows: “When ethanol was provided as the primary energy source, acetaldehyde accumulated progressively in the supernatant, suggesting the induction of ethanol metabolism. The absence of intracellular retention suggests that acetaldehyde was not sequestered within microcompartments, and therefore was not the result of taurine desulfonation.”

L326 Only six of the eight livers from the SIHUMI +Bw group are included. The last two ones should also be added.

We only had six livers for the SIHUMI+Bw group since the other two mice died before the end of the experiment, and the organs were not harvested. This is described in Line 739: “two mice of SIHUMI+Bw group died on day 40” and is consistent with all other analyses. We have now included in the SourceData file the number of animals “Groups: SIHUMI+Bw (N=6), SIHUMI (N=10), Bw (N=9)”. For clarity, this is now also included in the legend of Figure 5.